# Fully Distributed, Flexible Compositional Visual Representations via Soft Tensor Products

**Bethia Sun**[*] **Maurice Pagnucco** **Yang Song**

School of Computer Science and Engineering
UNSW Sydney

## Abstract

Since the inception of the classicalist vs. connectionist debate, it has been argued that the ability to systematically combine symbol-like entities into compositional representations is crucial for human intelligence. In connectionist systems, the field of disentanglement has gained prominence for its ability to produce explicitly compositional representations; however, it relies on a fundamentally *symbolic, concatenative* representation of compositional structure that clashes with the *continuous, distributed* foundations of deep learning. To resolve this tension, we extend Smolensky's Tensor Product Representation (TPR) and introduce *Soft TPR*, a representational form that encodes compositional structure in an inherently *distributed, flexible* manner, along with *Soft TPR Autoencoder*, a theoretically-principled architecture designed specifically to learn Soft TPRs. Comprehensive evaluations in the visual representation learning domain demonstrate that the Soft TPR framework consistently outperforms conventional disentanglement alternatives – achieving state-of-the-art disentanglement, boosting representation learner convergence, and delivering superior sample efficiency and low-sample regime performance in downstream tasks. These findings highlight the promise of a *distributed* and *flexible* approach to representing compositional structure by potentially enhancing alignment with the core principles of deep learning over the conventional symbolic approach.

## 1 Introduction

Compositional structure, capturing the property of being decomposable into a set of constituent parts, permeates numerous aspects of our world – from the recursive application of syntax in language, to the parsing of richly complex visual scenes into their constituent parts. Given this ubiquity, it is natural to seek deep learning representations that also embody compositional structure. Indeed, empirical evidence demonstrates a multitude of benefits conferred by explicitly compositional representations, including increased interpretability [10, 12], reduced sample complexity [30, 33], increased fairness [20, 25, 41], and improved performance in out-of-distribution generalisation [33, 48, 50].

We consider the following, intuitive notion of compositional representations. A representation of compositionally-structured data is a *compositional representation* if its structure explicitly reflects the constituency structure of the represented data [49]. In the visual representation learning domain, data is clearly *compositionally-structured*, as images can be decomposed into a set of constituent *factors of variation* (FoVs), e.g., {magenta floor, orange wall, aqua object colour, oblong object shape} for the image in Figure 1.

---

[*]Correspondence to: bethia.sun@unsw.edu.au
Code is available at: https://github.com/gomb0c/soft_tpr/

38th Conference on Neural Information Processing Systems (NeurIPS 2024).

A widely explored framework for learning explicitly compositional representations is that of *disentanglement*. We adopt the conventional [5, 25, 33, 43], intuitive definition of a *disentangled representation*, which states a representation, $\psi(x)$, is disentangled if each data constituent (FoV) can be mapped onto a distinct dimension or contiguous group of dimensions in $\psi(x)$, effectively establishing a 1-1 correspondence between FoVs and distinct representational *parts* [43]. Framed in this way, it is clear that disentangled representations explicitly encode the data's constituency structure and are thus, compositional representations by nature. The majority of state-of-the-art disentanglement approaches use a variational autoencoder backbone, and rely on weak supervision [13, 22, 31, 33, 35], or a penalisation of the aggregate posterior $\int q(z|x)p(x)dx$ [10, 14, 17, 24, 26, 32] to promote disentanglement. More recent approaches depart from the restrictive assumptions of a variational framework, and instead use standard autoencoding [47], or energy-function based optimisation [36], with additional inductive biases to encourage disentanglement. Despite their methodological diversity, disentanglement methods share a unifying principle: by enforcing the 1-1 correspondence between FoVs and distinct "slots" in the representation, they produce compositional representations as a dimension-wise *concatenation* of scalar-valued or vector-valued FoV *tokens* [10, 13, 14, 17, 22, 24, 26, 31, 32, 33, 35, 36, 47], as illustrated in Figure 1a. This design utilises a fundamentally localist encoding scheme, where discrete parts of the representation are exclusively dedicated to encoding *single* FoVs, paralleling symbolic representations, in which discrete representational slots are occupied by individual symbols – e.g., the symbols {'c', 'a', 't'} in the word 'cat'. We theorise that this inherently *symbolic* method, offering a *localist* encoding of compositional structure, may be misaligned with the continuous, distributed nature of deep learning models for the following reasons (see A.3 for further details):

1. **Gradient Flow and Learning**: Symbolic compositional representations utilise a localist encoding approach, assigning each FoV to a unique, non-overlapping subset of the dimensions of the representation. This modular separation may introduce practical misalignments with gradient-based optimisation. By restricting each FoV to its own slot, gradients for one FoV are confined to a set of designated dimensions, limiting the smooth propagation of gradients globally across *all* dimensions of the representation. Consequently, updates to a single FoV provide minimal feedback to other FoVs, hindering the model's ability to jointly refine interdependent components. Furthermore, the localist encoding – where each FoV occupies a small, disjoint subset of the total dimensions – can induce abrupt, discontinuous shifts in the representation when transitioning between FoV updates, potentially complicating convergence.

2. **Representational Expressivity**: The localist encoding inherent in symbolic compositional structures, where each FoV is assigned a distinct subset of the overall dimensions, may constrain practical expressiveness. Specifically, confining each of the $n$ FoVs to a subset of size $n/d$ within a $d$-dimensional representation causes each FoV to underutilise the full capacity of the $d$-dimensional space, potentially limiting the richness and flexibility of the learned representations.

3. **Robustness to Noise**: By confining each FoV to a small, non-overlapping subset of dimensions, symbolic compositional representations become highly vulnerable to dimension-wise noise. Even slight perturbations in a single dimension can significantly impair the representation of the corresponding FoV, as there is no overlapping or redundant encoding to mitigate such disruptions.

Critically, we hypothesise that this potential incompatibility between symbolic, localist representations and the distributed, continuous nature of deep learning results in suboptimal behaviour in models that learn or use these representations. To overcome these limitations, we are motivated to pursue an inherently *distributed* approach to representing compositional structure. Instead of concatenating discrete slots (*FoV tokens*) dimension-wise to form the compositional representation, an inherently *distributed* approach *continuously combines* densely encoded FoVs within a unified vector space. This design allows information from each FoV to be *smoothly interwoven* throughout all dimensions of the representation, as illustrated in Figure 1b, potentially resulting in smoother gradient flow, enhanced expressivity, and heightened robustness to noise. By reconciling the demand for representations that are explicitly compositional with the *continuous, distributed* nature of deep learning, distributed compositional representations offer a compelling alternative to traditional symbolic, slot-based paradigms.

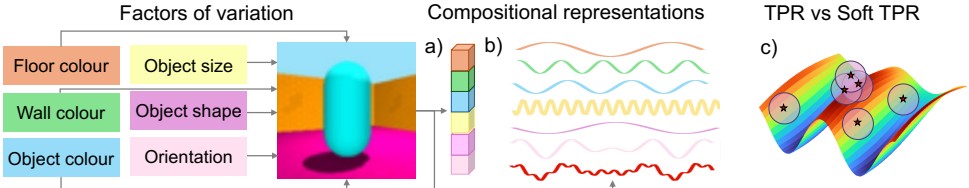

Figure 1: (a) Disentangled representations can be conceptualised as a *concatenation* of FoV tokens (coloured blocks), enforcing a *symbolic, string-like* compositional structure, where each FoV is allocated to a discrete *slot* in the representation. We instead, consider a *distributed* representation of compositional structure, (b), where information from densely encoded FoV (first 6 waves) are *continuously combined* together to form the representation, $\psi(x)$ (in red), effectively distributing the information from *multiple FoVs* into a *single* dimension of $\psi(x)$. (c) Only a subset of points (stars) in the underlying representational space (rainbow manifold) satisfy the TPR specification. The Soft TPR relaxes this, capturing larger, *continuous regions* of the underlying representational space (the translucent circles), while approximately preserving the TPR's key properties.

Pioneered by Smolensky, the Tensor Product Representation [3] is a specific representational form that encodes compositional structure in an inherently *distributed* manner. At the crux of it, TPRs are formed by *continuously blending* representational components together into the overall representation, in a manner analogous to superposing multiple waves together to produce a complex waveform, as illustrated in Figure 1b. For a representation to qualify as a TPR, it must adhere to a highly specific mathematical form, which confers upon the TPR valuable structural properties (elaborated on in Section 3.2), but also imposes two limitations (see B.1 for further details). First, as depicted by the stars in Figure 1c, only a discrete subset of points in the underlying representational space, $\tilde{V}$, satisfies the stringent mathematical criteria to qualify as TPRs. Consequently, to learn TPRs, representation learners must map from the data manifold onto this *discrete subset*, which constitutes a highly constrained and inherently challenging learning task. Second, the TPR specification enforces a strict, algebraic definition of compositional structure, limiting the TPR's ability to flexibly represent ambiguous, real-world data which is often *quasi*-compositional, only *approximately* adhering to some rigid, formal definition. Historically, these limitations have confined TPR learning to formal domains characterised by explicit, algebraic structure – as evidenced by the near exclusive deployment of TPRs in language [19, 23, 28, 34, 52] – and, to contexts where strong supervision from highly structured downstream tasks is available to steer the representation learning process [23, 28, 38, 51]. To negate these drawbacks and extend distributed compositional representations to weakly supervised, non-formal domains, we propose *Soft TPR*, which can be thought of as a *continuous relaxation* of the traditional TPR, as illlustrated by the translucent circular regions in Figure 1c. At its core, the Soft TPR is designed to promote representational flexibility and ease of learning while simultaneously preserving the structural integrity of the traditional TPR. We additionally introduce *Soft TPR Autoencoder*, a theoretically-principled *weakly-supervised* architecture for learning Soft TPRs, used to operationalise the Soft TPR framework in the visual representation learning domain.

Our main contributions are threefold: i) We propose a novel compositional representation learning framework, introducing the *distributed, flexible* Soft TPR compositional form, alongside a dedicated, weakly-supervised architecture, *Soft TPR Autoencoder*, for learning this form. ii) Our framework is the first to learn *distributed compositional representations* in the non-formal, less explicitly algebraic domain of *vision*. iii) We empirically affirm the far-reaching benefits produced by the Soft TPR framework, demonstrating that Soft TPRs achieve state-of-the-art disentanglement, accelerate representation learner convergence, and provide downstream models with enhanced sample efficiency and superior low-sample regime performance.

## 2 Related Work

**Disentanglement**: In aiming to produce explicitly compositional representations without strong supervision, our work shares the same objective as disentangled representation learning. Prior to the highly influential work of [25] which proved the impossibility of learning disentangled representations without supervision or other inductive biases, disentangled representations were learnt in a completely unsupervised fashion [8, 10, 14, 17, 24, 26, 37]. Our use of weak supervision is inspired by the work [13, 22, 31, 33, 35] relating to this highly influential impossibility result. In particular, we leverage the type of weak supervision termed 'match pairing' [35], where pairs, $(x, x')$, differing in values for a subset of known FoVs are presented to the model, to incentivise disentanglement. Our work,

however, fundamentally diverges from all disentanglement work we are aware of, by adopting an inherently *distributed* representation of compositional structure, which contrasts with the inherently *symbolic*, localist representations of compositional structure characterising existing work.

**TPR-based Work**: Existing TPR-based approaches generate distributed representations of compositional structure by producing an element with the explicit mathematical form of a TPR. To learn this highly specific form, these approaches rely on the algebraic characterisation of compositionality present in formal domains, such as mathematics [38], or language [19, 23, 28, 34, 52] in addition to strong supervision signals from highly structured downstream tasks, such as part-of-speech tagging [23], and answering structured language [28, 51] or mathematics questions [38]. In contrast, *Soft TPR* eases these stringent constraints by offering a continuously relaxed specification of compositional structure. This allows our approach to extend distributed representations of compositional structure to the orthogonal and less algebraically structured domain of *visual* data, while also reducing reliance on annotated data by instead using weak supervision to learn this relaxed representational form.

## 3 Preliminaries

### 3.1 A Formal Framework for Compositional Representations

We adopt a generalised, non-generative version of the definition of *compositional representations* from [49]. Let data $x \in X$ be *compositionally-structured* if there exists a decomposition function $\beta : X \to A_1 \times \ldots \times A_n$ decomposing $x$ into constituent parts (FoVs), i.e. $\beta(x) = \{a_1, \ldots, a_n\}$, where $a_i \in A_i$. A mapping $\psi : X \to V_F$ then produces a *compositional representation* if $\psi(x) = C(\psi_1(a_1), \ldots, \psi_n(a_n))$, where each $\psi_i : A_i \to V_i$ is a *component function* that independently embeds a part $a_i$ (i.e., a particular FoV) into a vector space, and $C : V_1 \times \ldots \times V_n \to V_F$ is a *composition function* that combines these embedded parts to form the overall representation. This construction enforces a faithful correspondence between the constituency structure of the data $\{a_i\}$ and that of the representation, $\{\psi_i(a_i)\}$ (provided $C$ is invertible).

We formalise a *symbolic* compositional representation, $\psi_s(x)$, as one in which $C$ is given by a concatenation operation. Concretely, $\psi_s(x) = \left(\psi_1(a_1)^T, \ldots, \psi_n(a_n)^T\right)^T$. Indeed, the disentanglement approaches of [8, 10, 13, 14, 17, 22, 24, 26, 31, 32, 33, 35, 36, 37, 47] all fit this framework, concatenating scalar or vector-valued FoV tokens together to produce the representation. While this design may be fundamentally at odds with the *continuous, distributed* nature of deep learning (as detailed in Section 1), it provides one clear benefit: the embedded FoVs, $\{\psi_i(a_i)\}$ are trivially recoverable by partitioning $\psi_s(x)$.

In many scenarios, readily recovering the data constituents (i.e., FoVs), $\{a_i\}$, from the compositional representation $\psi(x)$ is essential for practical utility. Here, we assume each $\psi_i$ is invertible, so direct recoverability of the data constituents, $\{a_i\}$, from the representation $\psi(x)$ is guaranteed if the embedded FoVs, $\{\psi_i(a_i)\}$, remain structurally separable in the representation, $\psi(x)$. Motivated by this requirement, we aim to construct compositional representations that: 1) achieve an inherently *distributed* encoding of compositional structure by *smoothly interweaving* FoVs throughout *all* dimensions of $\psi(x)$, and 2) preserve the direct recoverability of the embedded FoVs.

### 3.2 The TPR Framework

Smolensky's Tensor Product Representations (TPR) [3] provides one compelling realisation of an inherently *distributed* encoding of compositional structure that, under certain conditions, preserves the *direct recoverability* of the embedded FoVs. We briefly review key aspects here, with additional details and formal proofs deferred to Appendix A. The TPR framework conceptualises a compositionally-structured object, $x$, as comprising a number, $N_R$, of roles, where each role $r \in R$ is bound to a corresponding filler $f \in F$, with this binding being denoted by $f/r$. Thus, compositionally-structured objects are decomposeable into a set of role-filler bindings, $\beta(x) = \{f_i/r_i\}_{i=1}^{N_R}$. In natural language, where the TPR is most commonly deployed [19, 28, 34, 52], roles may correspond to grammatical categories (e.g., noun) and fillers specific words (e.g., cat). Translating this role-filler formalism to the visual domain, we reinterpret roles as FoV *types* (e.g., floor colour), and fillers as FoV *values* (e.g., blue). Using this role-filler formalism to decompose the Shapes3D image in Figure 1, we have $\beta(x) = \{$magenta/floor colour, orange/wall colour, aqua/object colour, large/object size, oblong/object shape$\}$.

To construct a TPR, the roles and fillers for each binding in $x$ are independently embedded via role and filler embedding functions, $\xi_R : R \to V_R, \xi_F : F \to V_F$ respectively. The binding $f/r$ is then encoded by taking a tensor product (denoted by $\otimes$) over the embedded role, $\xi_R(r)$, and embedded filler, $\xi_F(f)$. Summing over the embeddings for all role-filler bindings in $x$ yields the TPR, $\psi_{tpr}(x)$:

$$\psi_{tpr}(x) := \sum_i \xi_F(f_{m(i)}) \otimes \xi_R(r_i), \tag{1}$$

where $m : \{1, \ldots, N_R\} \to \{1, \ldots, N_F\}$ is a matching function[2] associating each of the $N_R$ roles to the filler it binds to in the decomposition of $x$ (i.e., $\beta(x) = \{f_{m(i)}/r_i\}$).

We now situate the TPR within the formal framework of Section 3.1, by observing that it defines each representational component, $\psi_i(a_i)$ as an embedded role-filler binding, $\xi_F(f_{m(i)}) \otimes \xi_R(r_i)$, and adopts *ordinary vector space addition* as its composition function, $C$. Unlike the *concatenation* in symbolic, localist schemes, the TPR *additively superposes* all embedded FoVs in the same underlying vector space, producing an inherently *distributed* representation of compositional structure in which information from *multiple* role-filler bindings can be smoothly interwoven into the *same* dimensions.[3]

A natural question arises from defining $C$ as ordinary vector space addition: can we recover each representational component – the embedded role-filler binding, $\xi_i(f_{m(i)}) \otimes \xi_R(r_i)$ – from the TPR, which is their sum? Remarkably, despite the non-injectivity of the summation operation, the TPR's specific algebraic structure enables faithful recoverability of all representational components from the TPR, through a process referred to as *unbinding* (see A.2 for more details). More concretely, provided that the role embedding vectors $\{\xi_R(r_i)\}$ are linearly independent and known, the embedded filler bound to the $i$-th role can be *unbound* from the representation, $\psi_{tpr}(x)$, by taking a (tensor) inner product between $\psi_{tpr}(x)$ and the $i$-th *unbinding* vector, $u_i$:

$$\psi_{tpr}(x)u_i = \left( \sum_i \xi_F(f_{m(i)}) \otimes \xi_R(r_i) \right) u_i = \xi_F(f_{m(i)}), \tag{2}$$

where $u_i$ is a vector corresponding to the $i$-th column of the (left) inverse of the matrix obtained by taking all (linearly independent) role embeddings as columns. Repeating this *unbinding* procedure using each of the $N_R$ unbinding vectors recovers each of the embedded fillers bound to the $N_R$ roles, and thus, the entire set of embedded role filler bindings, $\{\xi_F(f_{m(i)}) \otimes \xi_R(r_i)\}$. Hence, provided that the linear independence of the roles holds, the TPR offers an inherently *distributed* representation of compositional structure that nonetheless retains the chief benefit of *symbolic* approaches – direct recoverability of each representational component.

## 4 Methods

### 4.1 Soft TPR: An Extension to the TPR Framework

While Tensor Product Representations offer a robust framework for encoding compositional structure in a distributed manner, their rigid representational form imposes practical limitations. The core limitation lies in TPR's strict algebraic specification of compositional structure as a set of discrete bindings, each of which comprise a single role and a single filler. This specification precludes the representation of more ambiguous, quasi-compositional structures that cannot be neatly decomposed in this way. For instance, even in the highly compositional domain of natural language, there exist phenomena such as idiomatic expressions that resist straightforward decomposition into role-filler pairs because their meanings cannot be directly inferred from their composite parts. Such examples illustrate that real-world compositional structures may require more flexible representation than what is offered by traditional TPRs. Consequently, TPRs may struggle to model more nuanced forms of compositional structure that characterise less structured, non-formal data domains, such as vision. Additionally, TPRs present a challenging learning task. Their representational form $\psi_{tpr}(x) := \sum_i \xi_F(f_{m(i)}) \otimes \xi_R(r_i)$ is only satisfied by a *discrete subset* of points within the underlying representational space, $V_F \otimes V_R$, as illustrated in Figure 1c. This constraint forces representation learners to map data to a highly restricted, discrete set of points, making learning inherently arduous.

---

[2]We omit the dependence of $m$ on $x$ for notational clarity.

[3]See Appendix A.4 for further discussion on the distributed nature of the TPR.

To overcome these shortcomings, we introduce *Soft TPR*, a continuous relaxation of the traditional TPR specification. Soft TPRs allow any point in the underlying representational space $V_F \otimes V_R$ that *approximately* conforms to the TPR specification. This removes the need for representations to perfectly adhere to the rigid structural specification of the TPR, instead permitting them to *flexibly approximate* this rigid form. This flexibility provides two main advantages: **1. Enhanced Representational Flexibility**: Soft TPR facilitates the representation of *quasi*-compositional structures that do not perfectly adhere to the traditional TPR's structural specification. This includes scenarios where bindings are approximate rather than exact, or where multiple fillers influence a single role. **2. Improved Ease of Learning**: The relaxed representational specification allows representation learners to map data to broader regions of the representational space (as illustrated by the translucent circular regions in Figure 1c). This may facilitate more efficient learning, as there are more possible mappings from the data to the required representational form.

Formally, to define the Soft TPR, we consider the vector space underlying TPRs, $V_F \otimes V_F$, produced by an arbitrary role embedding function $\xi_R : R \to V_R$ and an arbitrary filler embedding function $\xi_F : F \to V_F$. A *Soft TPR* is defined as any element $z \in V_F \otimes V_R$ in this underlying representational space that is sufficiently close to a traditional TPR, $\psi_{tpr}$[4], as measured by a chosen distance metric (we take the Frobenius norm). Denoting the Frobenius norm by $||\cdot||_F$, Soft TPR satisfies $||z - \psi_{tpr}||_F < \epsilon$, where $\epsilon$ is some small, scalar-valued positive quantity. This condition ensures that the Soft TPR approximately encodes the TPR-based compositional structure of $\psi_{tpr}$, while allowing for slight deviations which capture a more flexible, relaxed notion of compositionality that cannot be reduced to a set of role-filler bindings (see B.2).

Despite this relaxation, Soft TPRs retain the critical advantage of traditional TPRs: the ability to recover constituent role-filler bindings through the unbinding operation, albeit approximately. Specifically, when unbinding a Soft TPR $z$, the operation yields soft filler embeddings $\tilde{f}_i$ that closely approximate the true filler embeddings $\xi_F(f_{m(i)})$. Formally, for a Soft TPR $z$, performing the unbinding operation for the $i$-th role yields:

$$zu_i \approx \psi_{tpr}u_i = \left( \sum_i \xi_F(f_{m(i)}) \otimes \xi_R(r_i) \right) u_i = \xi_F(f_{m(i)}), \tag{3}$$

$$= \xi_F(f_{m(i)}) + \epsilon_i =: \tilde{f}_i, \tag{4}$$

where $\epsilon_i = zu_i - \xi_F(f_{m(i)})$ represents the approximation error. This approximate recoverability ensures that Soft TPRs *implicitly encode* a precise, algebraically-expressible form of compositional structure, even when their representational form *deviates* from the exact TPR specification.

### 4.2 Soft TPR Autoencoder: Learning Distributed and Flexible Compositional Representations

We define our vector spaces of interest over the reals as $V_F := \mathbb{R}^{D_F}$ and $V_R := \mathbb{R}^{D_R}$ where $D_F, D_R$ denote the dimensionality of the filler and role embedding spaces. The pivotal insight underlying our method is that a *Soft TPR* can be effectively realised by ensuring that the encoder produces representations that are amenable to quantisation into explicit TPRs. Specifically, since a Soft TPR is any arbitrary element from the vector space $\mathbb{R}^{D_F \cdot D_R}$[5] sufficiently close to an explicit TPR, any $(D_F \cdot D_R)$-dimensional vector produced by an encoder in a standard autoencoding framework can be treated as a Soft TPR *candidate*. This realisation suggests that a conventional autoencoding framework only requires modest modifications to generate Soft TPRs.

Our Soft TPR Autoencoder comprises three main components: a standard encoder, $E$, the TPR decoder, and a standard decoder, $D$. The encoder output, $z$, serves as the Soft TPR. The overarching intuition guiding our architecture is to ensure that $z$ can be effectively quantised or decoded into an explicit TPR, thereby enforcing the Soft TPR property, $||z - \psi_{tpr}||_2 < \epsilon$.

**Representational Form**: To ensure that the encoder produces elements with the *form* of a Soft TPRs, we introduce a mechanism that penalises the Euclidean distance $||z - \psi_{tpr}^*||_2$ between the encoder

---

[4]We occasionally omit the dependence on $x$ for notational clarity.

[5]Due to isomorphism of vector spaces $\mathbb{R}^{D_F} \otimes \mathbb{R}^{D_R} \cong \mathbb{R}^{D_F \cdot D_R}$, we henceforth use vectors from $\mathbb{R}^{D_F \cdot D_R}$ in place of rank-2 tensors from $\mathbb{R}^{D_F} \otimes \mathbb{R}^{D_R}$, and the Euclidean norm instead of the Frobenius norm to align the Soft TPR framework more seamlessly with the autoencoding framework.

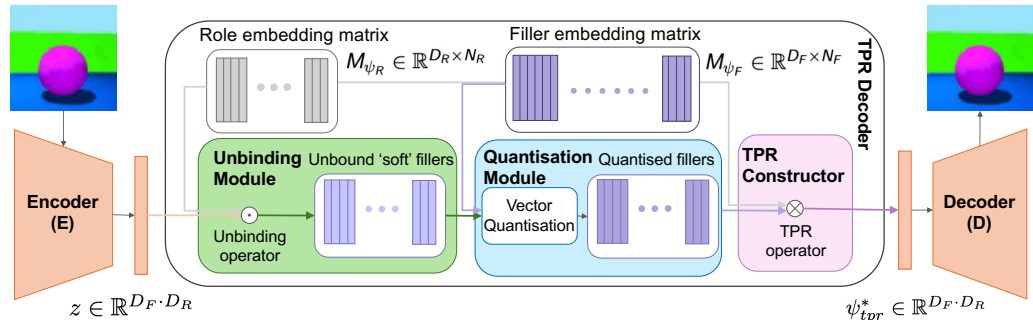

Figure 2: Diagram illustrating the Soft TPR Autoencoder. We encourage the encoder $E$'s output, $z$, to have the form of a Soft TPR by penalising its distance with the greedily defined, explicit TPR, $\psi_{tpr}^*$ of Equation 5 that $z$ best approximates. $\psi_{tpr}^*$ is recovered using a 3 step process performed by our TPR decoder (center rectangle): 1) unbinding, 2) quantisation, and 3) TPR construction. The decoder, $D$, reconstructs the input image using $\psi_{tpr}^*$.

output, $z$, and some explicit TPR, $\psi_{tpr}^*$, that $z$ best approximates. To obtain $\psi_{tpr}^*$ needed to compute the loss, we derive an explicit analytical form for $\psi_{tpr}^*$ and construct elements satisfying this analytical form using a role embedding matrix, $M_{\xi_R}$ containing $N_R$ $D_R$-dimensional role embedding vectors, $\{\xi_R(r_i)\}$, and a filler embedding matrix, $M_{\xi_F}$ containing $N_F$ $D_F$-dimensional filler embedding vectors, $\{\xi_F(f_i)\}$. To define an explicit analytical form for $\psi_{tpr}^*$, we use a greedily optimal selection of filler embeddings based on their proximity to the soft filler embeddings extracted from $z$:

$$\psi_{tpr}^* := \sum_i \xi_F(f_{m(i)}) \otimes \xi_R(r_i), \text{where } m(i) := \arg\min_j ||\tilde{f}_k - \xi_F(f_j)||_2, \text{ and } \tilde{f}_k := z u_i. \quad (5)$$

That is, we define $\psi_{tpr}^*$ as the TPR constructed from explicit filler embeddings $\xi_F(f_j)$ with the smallest Euclidean distance to the *soft* filler embeddings $\tilde{f}_k$ of $z$. To construct elements satisfying (5), we use a three-step process carried out by the novel TPR decoder we introduce, visible in Figure 2: **1) Unbinding**: Utilising a randomly-initialised dense role embedding matrix $M_{\xi_R}$, the unbinding module extracts soft filler embeddings $\{\tilde{f}_i\}$ by performing the TPR *unbinding* operation on $z$. We elaborate on our theoretically-informed reason for this design, how the unbinding vectors are obtained, and for not backpropagating gradient to $M_{\xi_R}$ in B.4.1. **2) Quantisation**: The quantisation module, containing a learnable filler embedding matrix $M_{\xi_F}$, employs the VQ-VAE vector quantisation algorithm [11] to both 1) learn the explicit filler embeddings, and 2) quantise the soft filler embeddings $\{\tilde{f}_1, \ldots, \tilde{f}_{N_R}\}$ produced by the unbinding module into the explicit filler embeddings $\{\xi_F(f_1), \ldots, \xi_F(f_{N_R})\}$ with the smallest Euclidean distances. **3) TPR Construction**: The quantised filler embeddings $\{\xi_F(f_{m(i)})\}$ are then bound to their respective roles using the tensor product, and all embedded bindings are summed together to produce the explicit TPR $\psi_{tpr}^*$ with the form of Eq 5. $\psi_{tpr}^*$ is subsequently used by the decoder, $D$, to reconstruct the input image. The overall unsupervised loss, $\mathcal{L}_u$, comprises three components, where s denotes the stop-gradient operator, $\beta$ is a hyperparameter controlling the commitment loss in VQ-VAE, and $\mathcal{L}_r$ is a suitable image-based reconstruction loss (we use $L_2$):

$$\mathcal{L}_u := \underbrace{||z - \psi_{tpr}^*||_2^2}_{\text{Soft TPR form penalty}} + \underbrace{\mathcal{L}_r(x, D(\psi_{tpr}^*))}_{\text{reconstruction loss}} + \underbrace{\sum_i \frac{1}{N_R}\left(||s[\xi_F(f_{m(i)})] - \tilde{f}_i||_2^2 + \beta||\xi_F(f_{m(i)}) - s[\tilde{f}_i]||_2^2\right)}_{\text{VQ-VAE quantisation loss}}.$$

$$(6)$$

**Representational Content**: While the unsupervised loss $\mathcal{L}_u$ ensures that $z$ maintains the Soft TPR form by being close to an explicit TPR, it does not guarantee that the content of $z$ accurately reflects the true role-filler semantics of the data. To address this, we introduce a weakly supervised loss that aligns $z$ with the ground-truth semantics of the image. We employ a match-pairing context similar to [13, 22, 33, 35], where image pairs $(x, x')$ share the same role-filler bindings for all but one role, $r_i$. The identity of $r_i$ is known, but not any of the specific fillers or bindings. Our intuition is that, for $z$ to accurately reflect the semantics of the image, the Euclidean distance between the quantised fillers of $x$ and $x'$ bound to role $r_i$ should be maximal, relative to the distances between the filler embeddings pairs for all other roles $r_j, j \neq i$. To encourage this, we apply the cross entropy loss corresponding to the 3rd term in Eq 7, where $\Delta q$ denotes the $N_R$-dimensional vector with each dimension $(\Delta q)_k$

populated by the Euclidean distance between the quantised fillers of $x$ and $x'$ for role $r_k$, and $l$ denotes the one-hot vector of dimension $N_R$ with the index for $r_i$ set to 1. Additionally, like [47], we incorporate a reconstruction loss (the 2nd term of Eq 7) to enforce consistency when swapping the quantised filler embeddings for role $r_i$. Specifically, we construct new TPRs by swapping the quantised fillers for $r_i$ between $x$ and $x'$ to generate $\psi^s_{tpr}(x)$ and $\psi^s_{tpr}(x')$, and ensure that the decoder can accurately reconstruct the corresponding swapped images from these swapped TPRs.

Combining these components, our final loss, $\mathcal{L}$, is a weighted sum over the unsupervised and weakly supervised loss components, where $\lambda_1$ and $\lambda_2$ are hyperparameters:

$$\mathcal{L} := \mathcal{L}_u + \lambda_1 \left( \frac{1}{2}\mathcal{L}_r(x, D(\psi^s_{tpr}(x'))) + \frac{1}{2}\mathcal{L}_r(x', D(\psi^s_{tpr}(x))) \right) + \lambda_2 \text{CE}(\Delta q, l). \tag{7}$$

## 5 Results

To evaluate our Soft TPR framework, we perform evaluation along three dimensions that are standard in the compositional representation learning literature [25, 30, 33, 42]: **1) Compositional Structure / Disentanglement**: To what extent do Soft TPR representations capture *explicitly compositional structure*? **2) Representation Learner Convergence**: How *efficiently* can representation learners acquire Soft TPR's *distributed, flexible* compositional structure? **3) Downstream Model Performance**: Does Soft TPR's *distributed, flexible* compositional form offer tangible benefits for *downstream models* utilising compositional representations?

We benchmark against a suite of weakly supervised disentanglement baselines: Ada-GVAE [33], GVAE [22], ML-VAE [13], SlowVAE [39], and the GAN-based model of [35], which we henceforth refer to as 'Shu'. These models produce symbolic compositional representations corresponding to a concatenation of *scalar-valued* FoV tokens. Similar to our approach, Ada-GVAE, GVAE, ML-VAE, SlowVAE, and Shu are trained with paired samples $(x, x')$ sharing values for all but a subset of FoVs types (roles), $I$, with ML-VAE, GVAE, SlowVAE, and Shu assuming access to $I$, matching our model's level of supervision. To ensure a fair comparison, we modify Ada-GVAE (method detailed in Appendix C.2.2) for more direct comparability, denoting our modification by Ada-GVAE-k. Additionally, we benchmark against 2 baselines producing symbolic *vector-tokened* compositional representations: COMET [36], and Visual Concept Tokeniser (VCT) [47], although these are fully unsupervised and not directly comparable to our method. We train five instances of each representation learning model using five random seeds for 200,000 iterations across all datasets, and report averaged results (an unabridged suite of all results is contained in the Appendix).

### 5.1 Compositional Structure / Disentanglement

To demonstrate that Soft TPRs inherently capture the algebraic compositional structure required for effective disentanglement, we first quantise each Soft TPR, $z$, into its corresponding explicit TPR, $\psi^*_{tpr}$ using the TPR decoder (note that once the model is trained, quantisation is fully deterministic). This quantisation process transforms the implicit compositional information encoded within Soft TPRs into an explicit algebraic form suitable for evaluation. We then apply standard disentanglement metrics across benchmark datasets including Cars3D [4], MPI3D [21], and Shapes3D [24]. Table 1 illustrates that Soft TPR consistently outperforms symbolic compositional baselines across all datasets, achieving state-of-the-art results with notable DCI metric improvements of 29% on Cars3D and 74% on the more challenging MPI3D dataset. To ensure these gains are attributable to Soft TPR framework rather than merely a marginal increase in model capacity (Soft TPR Autoencoder adds between 1,600-13,568 model parameters to a standard (variational) autoencoder), we conduct control experiments by equalising parameter counts in applicable baselines. Consistent with findings from prior work [40], we observe no performance improvements in relevant symbolic baselines when parameter counts are increased (see C.2.4).

### 5.2 Representation Learner Convergence Rate

To explore how the Soft TPR framework influences the convergence rate of representation learners, we analyse representations produced at varying stages of representation learner training (100, 1,000, 10,000, 100,000 and 200,000 iterations), and evaluate both their **1) Disentanglement**, and **2) Downstream Utility**. We quantify downstream utility of learned representations by examining the

performance of downstream models on two tasks common in the disentanglement literature [25, 30, 33, 42]: a classification-based task assessing the model's ability to perform abstract visual reasoning [30] and a regression task involving the prediction of continuous FoV values for the disentanglement datasets [42]. Our findings indicate that the Soft TPR framework generally achieves faster disentanglement convergence, particularly on Cars3D and MPI3D datasets (see Appendix C.4.1). Moreover, Soft TPR consistently accelerates the learning of useful representations for *both* downstream tasks. The downstream performance improvements are particularly pronounced in the low iteration regime – for instance, at only 100 iterations of representation learner training, as shown in Tables 2 and 3. For fair comparison, we embed baseline representations of both higher and lower dimensionality into the same vector space as Soft TPR, which we denote by †, and select the best-performing variant (original or dimensionality-matched) for each baseline (details in Appendix C).

Table 1: FactorVAE and DCI scores. Additional results in Section C.3.3

| Models | Cars3D | | Shapes3D | | MPI3D | |
|---|---|---|---|---|---|---|
| | FactorVAE score | DCI score | FactorVAE score | DCI score | FactorVAE score | DCI score |
| | Symbolic scalar-tokened compositional representations | | | | | |
| Slow-VAE | $0.902 \pm 0.035$ | $0.509 \pm 0.027$ | $0.950 \pm 0.032$ | $0.850 \pm 0.047$ | $0.455 \pm 0.083$ | $0.355 \pm 0.027$ |
| Ada-GVAE-k | $0.947 \pm 0.064$ | *$0.664 \pm 0.167$* | *$0.973 \pm 0.006$* | **$0.963 \pm 0.077$** | $0.496 \pm 0.095$ | $0.343 \pm 0.040$ |
| GVAE | $0.877 \pm 0.081$ | $0.262 \pm 0.095$ | $0.921 \pm 0.040$ | $0.842 \pm 0.040$ | $0.378 \pm 0.024$ | $0.245 \pm 0.074$ |
| ML-VAE | $0.870 \pm 0.052$ | $0.216 \pm 0.063$ | $0.835 \pm 0.111$ | $0.739 \pm 0.115$ | $0.390 \pm 0.026$ | $0.251 \pm 0.029$ |
| Shu | $0.573 \pm 0.062$ | $0.032 \pm 0.014$ | $0.265 \pm 0.043$ | $0.017 \pm 0.006$ | $0.287 \pm 0.034$ | $0.033 \pm 0.008$ |
| | Symbolic vector-tokened compositional representations | | | | | |
| VCT | *$0.966 \pm 0.029$* | $0.382 \pm 0.080$ | $0.957 \pm 0.043$ | $0.884 \pm 0.013$ | *$0.689 \pm 0.035$* | *$0.475 \pm 0.005$* |
| COMET | $0.339 \pm 0.008$ | $0.024 \pm 0.026$ | $0.168 \pm 0.005$ | $0.002 \pm 0.000$ | $0.145 \pm 0.024$ | $0.005 \pm 0.001$ |
| | Fully continuous compositional representations | | | | | |
| Ours | **$0.999 \pm 0.001$** | **$0.863 \pm 0.027$** | **$0.984 \pm 0.012$** | *$0.926 \pm 0.028$* | **$0.949 \pm 0.032$** | **$0.828 \pm 0.015$** |

Table 2: FoV regression $R^2$ scores (100 iterations of representation learner training).

| Models | Cars3D | Shapes3D | MPI3D |
|---|---|---|---|
| | Symbolic scalar-tokened | | |
| Slow-VAE | $0.233 \pm 0.048$ | $0.600 \pm 0.048$ | *$0.557 \pm 0.012$* |
| Ada-GVAE-k | $0.307 \pm 0.084$ | $0.684 \pm 0.059$ | $0.519 \pm 0.023$ |
| GVAE | $0.319 \pm 0.073$ | $0.565 \pm 0.034$ | $0.511 \pm 0.037$ |
| ML-VAE | $0.317 \pm 0.058$ | $0.551 \pm 0.059$ | $0.504 \pm 0.016$ |
| Shu† | $0.012 \pm 0.007$ | $0.461 \pm 0.075$ | $0.299 \pm 0.040$ |
| | Symbolic vector-tokened | | |
| VCT | $0.080 \pm 0.001$ | *$0.886 \pm 0.033$* | $0.316 \pm 0.016$ |
| COMET† | *$0.484 \pm 0.477$* | $0.474 \pm 0.062$ | $0.240 \pm 0.010$ |
| | Fully continuous | | |
| Ours | **$0.531 \pm 0.054$** | **$0.981 \pm 0.003$** | **$0.732 \pm 0.012$** |

Table 3: Abstract visual reasoning accuracy (100 iterations of representation learner training).

| Models | Abstract visual reasoning dataset |
|---|---|
| | Symbolic scalar-tokened |
| Slow-VAE† | $0.552 \pm 0.035$ |
| Ada-GVAE-k† | *$0.631 \pm 0.037$* |
| GVAE† | $0.554 \pm 0.031$ |
| ML-VAE† | $0.550 \pm 0.025$ |
| Shu | $0.208 \pm 0.052$ |
| | Symbolic vector-tokened |
| VCT | $0.440 \pm 0.033$ |
| COMET† | $0.348 \pm 0.069$ |
| | Fully continuous |
| Ours | **$0.804 \pm 0.016$** |

## 5.3 Downstream Models

To evaluate whether the *distributed* and *flexible* encoding of compositional structure offered by the Soft TPR benefits downstream models, we examine both: **1) Sample Efficiency** and **2) Raw Performance in Low Sample Regimes**, using the previously mentioned abstract visual reasoning and FoV regression tasks. In line with [25], we quantify sample efficiency with a ratio-based metric comparing downstream model performance using restricted sample sizes (100, 250, 500, 1,000, and 10,000) against performance when trained using all samples. As illustrated in Table 4, Soft TPR substantially outperforms baseline models in sample efficiency, especially in the most restrictive case involving *only 100* samples, achieving a 93% improvement. Additionally, Soft TPR produces substantial raw performance increases in the low sample regime, as evidenced by the 138% and 168% improvements in Table 4 for the low-sample regimes of 100 and 200 samples respectively, and the 30% improvement in Table 5.

## 5.4 Ablation Studies

To disentangle the contributions of Soft TPR's foundational components, we conduct ablation studies focusing on two critical properties that are hypothesised to influence the Soft TPR's learning of compositional representations: **1) Relaxed Representational Constraints** and **2) Distributed Encoding**, denoted RRC and DE respectively in Table 7. For **Relaxed Representational Constraints**, we significantly increase the weighting of the form penalty $||z - \psi^*_{tpr}||_2$ in the Soft TPR loss to enforce rigid representational constraints on the Soft TPR Autoencoder, effectively forcing it to

Table 4: Downstream FoV $R^2$ scores (odd columns) and sample efficiencies (even columns) on the MPI3D dataset.

| Models | 100 samples | 100 samples/all | 250 samples | 250 samples/all |
|---|---|---|---|---|
| | Symbolic scalar-tokened compositional representations | | | |
| Slow-VAE | $0.127 \pm 0.050$ | $0.130 \pm 0.051$ | $0.152 \pm 0.011$ | $0.155 \pm 0.011$ |
| Ada-GVAE-k | $0.206 \pm 0.031$ | $0.270 \pm 0.037$ | $0.213 \pm 0.023$ | $0.279 \pm 0.026$ |
| GVAE | $0.181 \pm 0.030$ | $0.234 \pm 0.035$ | $0.217 \pm 0.023$ | $0.282 \pm 0.027$ |
| ML-VAE | $0.182 \pm 0.013$ | $0.236 \pm 0.019$ | $0.222 \pm 0.024$ | $0.288 \pm 0.030$ |
| Shu | $0.151 \pm 0.016$ | $0.343 \pm 0.024$ | $0.211 \pm 0.026$ | $0.482 \pm 0.075$ |
| | Symbolic vector-tokened compositional representations | | | |
| VCT | $0.086 \pm 0.051$ | $0.189 \pm 0.107$ | $0.119 \pm 0.070$ | $0.246 \pm 0.137$ |
| COMET | $-0.051 \pm 0.015$ | $0.000 \pm 0.000$ | $-0.042 \pm 0.018$ | $0.000 \pm 0.000$ |
| | Fully continuous compositional representations | | | |
| Ours | $\mathbf{0.490 \pm 0.068}$ | $\mathbf{0.556 \pm 0.078}$ | $\mathbf{0.594 \pm 0.056}$ | $\mathbf{0.665 \pm 0.067}$ |

Table 5: Abstract visual reasoning accuracy in the low-sample regime of 500 samples.

| Models | Symbolic scalar-tokened |
|---|---|
| Slow-VAE | $0.196 \pm 0.028$ |
| Ada-GVAE-k | $0.203 \pm 0.007$ |
| GVAE | $0.182 \pm 0.013$ |
| ML-VAE | $0.193 \pm 0.012$ |
| Shu | $0.200 \pm 0.010$ |
| | Symbolic vector-tokened |
| VCT | $0.277 \pm 0.039$ |
| COMET | $0.259 \pm 0.016$ |
| | Fully continuous |
| Ours | $\mathbf{0.360 \pm 0.033}$ |

Table 6: Downstream FoV $R^2$ scores for TPR and Soft TPR (number of samples in brackets)

| Representational form | Cars3D | Shapes3D | MPI3D |
|---|---|---|---|
| TPR (100 samples) | $0.361 \pm 0.031$ | $0.447 \pm 0.108$ | $0.415 \pm 0.050$ |
| Soft TPR (100 samples) | $\mathbf{0.638 \pm 0.010}$ | $\mathbf{0.481 \pm 0.081}$ | $\mathbf{0.531 \pm 0.069}$ |
| TPR (250 samples) | $0.537 \pm 0.053$ | $0.669 \pm 0.108$ | $0.598 \pm 0.081$ |
| Soft TPR (250 samples) | $\mathbf{0.832 \pm 0.027}$ | $\mathbf{0.728 \pm 0.021}$ | $\mathbf{0.621 \pm 0.072}$ |

Table 7: Effect of model properties on disentanglement (MPI3D dataset).

| Property | DCI Score |
|---|---|
| - RRC & DE | $0.225 \pm 0.034$ |
| - RRC | $0.598 \pm 0.093$ |
| - DE | $0.704 \pm 0.135$ |
| + RRC & DE (Full) | $0.828 \pm 0.015$ |

produce representations that precisely match the TPR-specified compositional form. For **Distributed Encoding**, we modify the model to encode compositional structure in an inherently less distributed manner, by modifying the sparsity of $\psi^*_{tpr}$ (see C.2.4). Results in Table 7 demonstrate that both the Soft TPR's distributed encoding and relaxed constraints are essential for learning representations with high compositional structure, as quantified by the disentanglement metrics. Additionally, we examine the impact of Soft TPR's flexible, *quasi-compositional* form on downstream performance by replacing each Soft TPR, $z$, with its quantised explicit TPR, $\psi^*_{tpr}$ in downstream tasks. Table 6 and Appendix C.6.3 indicates that Soft TPRs provide downstream models with *unique* performance improvements that explicit TPRs (which are produced under the same conditions) cannot account for. Please see Appendix C.6 for further details and full ablation results, including additional ablations.

## 5.5 Key Insights

The empirical results of the Soft TPR framework reveal two primary insights. Firstly, deep learning models may more effectively learn precise compositional structures when these structures are 1) *implicitly acquired* through *relaxed representational constraints*, allowing the model to acquire precise compositional forms (i.e., $\psi^*_{tpr}$) through quantisation of flexible Soft TPRs, $z$, and 2) when the compositional structure is assumed to be encoded in a *distributed* format. Secondly, the *flexible representational form* of Soft TPR offers downstream models *unique* advantages by enabling more efficient utilisation of *implicitly* encoded compositional information compared to their rigid counterparts, $\psi^*_{tpr}$. Together, these comprehensive model benefits empiricially demonstrate the value of the framework's 1) relaxed representational constraints, 2) distributed encoding of compositional structure, and 3) flexible representational form.

## 6 Conclusion

In this work, we tackle a challenge tracing its roots to the conception of the connectionist vs. classicalist debate: the fundamental mismatch between compositional structure and the inherently distributed nature of deep learning. To bridge this gap, we introduce the *Soft TPR*, a new, inherently *distributed*, *flexible* compositional representational form extending Smolensky's Tensor Product Representation, together with the *Soft TPR Autocoder*, a theoretically-principled architecture designed for learning Soft TPRs. Our *flexible*, *distributed* framework demonstrates substantial improvements in the visual representation learning domain – enhancing the representation of compositional structure, accelerating convergence in representation learners, and boosting efficiency in downstream models. Future work will extend this framework to hierarchical compositional structures, enabling bound fillers to recursively decompose into role-filler bindings for enhanced representational expressivity.

## Acknowledgements

This research has been supported by an UNSW University Postgraduate Award. We thank the reviewers for their valuable comments, J. Hershey for a healthy dose of skepticism which greatly improved this work, and B. Spehar for insightful discussions.

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

# Appendix

## A   TPR Framework

In this section, we provide additional details regarding Smolensky's TPR framework [3], as well as formal proofs of presented results. We refer interested readers to [3] for a more comprehensive dive into the TPR framework.

### A.1   Additional Details

By defining the constituent components of any *compositional object* as a set of role-filler bindings, the TPR defines the decomposition function, $\beta$, of Section 3.1 that maps from a set of compositional objects, $X$, to a set of parts, more explicitly as follows [3]:

$$\beta : X \to 2^{F \times R}; x \to \{(f, r)|f/r\}, \tag{8}$$

where $F$ denotes a set of fillers, $R$ denotes a set of roles, and $f/r$ denotes the binding of filler $f \in F$ to role $r \in R$. Note that in contrast to the formal definition of $\beta$ we use in Section 3.1, which assumes each $x \in X$ is decomposable into a set of $n$ parts, the above decomposition allows objects to be decomposed into a *variably-sized* set of role-filler bindings, with this set corresponding to an element in the powerset of $F \times R$. For the considered visual representation learning domain, all datasets clearly have the property of being decomposable into a *fixed* size set of role-filler bindings, as all images in these datasets contain the same number of FoV *types* and each FoV type is bound to a FoV *token*. Due to this property, we can take $\beta$ as a special subcase of the generalised definition in Equation 8:

$$\beta : X \to \mathcal{A} : x \to \{(f_{m(i)}, r_i) | f_{m(i)}/r_i\}. \tag{9}$$

where $m : \{1, \ldots, N_R\} \to \{1, \ldots, N_F\}$ denotes a matching function that associates each role $r_i$ with the filler it binds to in $\beta(x)$ (we again drop the dependence of $m$ on $x$ for ease of notation), and $\mathcal{A}$ denotes the set of all possible bindings produced by binding a filler to each of the $N_R$ roles, with size $N_F^{N_R}$ (we assume the same filler can bind to multiple roles).

## A.2 Formal Proofs

We now formally prove that the TPR with $\beta$ defined in 9 has form $\psi_{tpr}(x) = C(\psi_1(a_1), \ldots, \psi_n(a_n))$ and thus corresponds to the definition of a *compositional representation* in Section 3.1.

*Proof.* We denote the role embedding and filler embedding functions as $\xi_R : R \to V_R, \xi_F : F \to V_F$ respectively.

By definition of the TPR in Eq 1, we have that:

$$\psi_{tpr}(x) := \sum_i \xi_F(f_{m(i)}) \otimes \xi_R(r_i).$$

and hence,

$$\psi_{tpr}(x) = \sum_i \psi_i(f_{m(i)}, r_i),$$

where $a_i := (f_{m(i)}, r_i) \in \beta(x)$, $\psi_i : F \times R \to V_F \otimes V_R; (f_{m(i)}, r_i) \to \xi_F(f_{m(i)}) \otimes \xi_R(r_i)$, and $C$ is ordinary vector space addition. Hence, almost trivially, $\psi_{tpr}(x)$ clearly has the required form to be a *compositional representation*. □

Now, we prove the recoverability of the embedded components $\{\psi_i(f_{m(i)}, r_i)\}$ from the TPR, $\psi_{tpr}(x)$, provided that the set of all role embedding vectors, $\{\xi_R(r_i)\}$, are linearly independent. Similar variants of this proof can be found in [3], [19].

*Proof.* Assume the set of all role embedding vectors $\{\xi_R(r_i)\}$ are linearly independent. Then, the role embedding matrix, $M_{\xi_R} := (\xi_R(r_1) \ldots \xi_R(r_{N_R}))$ formed by taking the role embedding vectors as columns, has a left inverse, $U$, such that:

$$U M_{\xi_R} = I_{N_R \times N_R}.$$

Hence, we have that $(U M_{\xi_R})_{ij} = U_{i:} M_{\xi_R:j} = I_{ij}$.

For ease of notation, let $u_i$ denote the $i$-th column of $U^T$, and note that $\xi_R(r_j)$ clearly corresponds to $M_{\xi_R:j}$. So, $U_{i:} M_{\xi_R:j} = (U_{:i}^T)^T M_{\xi_R:j} = u_i^T \xi_R(r_j) = I_{ij}$.

Hence, we have that:

$$u_i^T \xi_R(r_j) = \delta_{ij} = \begin{cases} 1 & i = j \\ 0 & \text{otherwise.} \end{cases}$$

Using the definition of $\psi_{tpr}(x)$, $\psi_{tpr}(x) = \sum_i \xi_F(f_{m(i)}) \otimes \xi_R(r_i)$, we apply the (tensor) inner product of $\psi_{tpr}(x)$ with $u_i$:

$$\psi_{tpr}(x) = \left( \sum_i \xi_F(f_{m(i)}) \otimes \xi_R(r_i) \right) u_i$$

$$= \left( \sum_i \xi_F(f_{m(i)}) \xi_R(r_i)^T \right) u_i$$

$$= \sum_i \xi_F(f_{m(i)}) \delta_{ji}$$

$$= \xi_F(f_{m(i)}).$$

Thus, the embedding of the filler, $f_{m(i)}$, bound to each role, $r_i$, can be recovered through use of a tensor inner product with the *unbinding* vector, $u_i$, corresponding to the $i$-th column of $U^T$. Note that the representational components of $\psi_{tpr}(x)$, i.e., the embedded bindings, $\psi_i(f_{m(i)}, r_i)$ are fully determined by the embedding of the role, $\xi_R(r_i)$, and filler, $\xi_F(f_{m(i)})$, comprising the binding, as $\xi_F(f_{m(i)}, r_i)$ simply corresponds to their tensor product. Thus, recovering $(\xi(f_{m(i)}), \xi_R(r_i))$ for each binding in $\beta(x)$ corresponds to recovering the representational component, $\psi_i(f_{m(i)}, r_i)$. So, provided the set of role embeddings are linearly independent, and they can be obtained (e.g. through a look-up table of role embeddings), all representational components, $\psi_i(f_{m(i)}, r_i)$ can be directly recovered from the overall TPR representation, $\psi_{tpr}(x)$. □

### A.3 Shortcomings of Symbolic Compositional Representations and How TPR Helps

Here, we provide a more detailed elaboration on the shortcomings of symbolic compositional representations as outlined in 1. We additionally illustrate how TPR-based continuous compositional representations circumvent such limitations through use of a concrete example.

We employ the formal definition of a *symbolic* compositional representation from Section 3.1, which states a representation, $\psi_s(x)$, is a *symbolic compositional representation* if $\psi_s(x) = \left( \psi_1(a_1)^T, ..., \psi_n(a_n)^T \right)^T$, i.e., if $\psi_s(x)$ corresponds to a *concatenation* of representational components, $\{\psi_i(a_i)\}$.

For more direct comparison between the symbolic compositional representation, $\psi_s(x)$ and the TPR-based compositional representation, $\psi_{tpr}(x)$, we rewrite the TPR-based representation as $\psi_{tpr}(x) = \tilde{\psi}(\tilde{a}_1) + \ldots + \tilde{\psi}_n(\tilde{a}_n)$. Thus, the TPR can be viewed as an *ordinary vector sum* of components, $\{\tilde{\psi}(\tilde{a}_i)\}$, where each component, $\tilde{\psi}_i$ corresponds to an embedded role-filler binding.

We consider the following example: suppose there are 2 FoV types (i.e., roles), colour, shape and 2 FoV values (i.e., fillers), purple, square. For the *symbolic* approach, suppose the *representational components*, $\{\psi_i(a_i)\}$ are defined as follows: $\psi_{\text{colour}}(\text{purple}) = \begin{bmatrix} 1 \\ 0 \end{bmatrix}$ and $\psi_{\text{shape}}(\text{square}) = \begin{bmatrix} 2 \\ 3 \end{bmatrix}$. For the *distributed* approach, suppose the (linearly independent) role embeddings are: $\xi_R(\text{colour}) = \begin{bmatrix} 1 \\ 2 \end{bmatrix}$, $\xi_R(\text{shape}) = \begin{bmatrix} 1 \\ 1 \end{bmatrix}$, and the filler embeddings are: $\xi_F(\text{purple}) = \begin{bmatrix} 1 \\ 1 \end{bmatrix}$ and $\xi_F(\text{square}) = \begin{bmatrix} 2 \\ 3 \end{bmatrix}$.

Then, a symbolic compositional representation is given by (denoting the concatenation operation by $\oplus$):

$$\psi_s(x) = \psi_{\text{colour}}(\text{purple}) \oplus \psi_{\text{shape}}(\text{square})$$

$$= \begin{bmatrix} 1 \\ 0 \end{bmatrix} \oplus \begin{bmatrix} 2 \\ 3 \end{bmatrix} = \begin{bmatrix} 1 \\ 0 \\ 2 \\ 3 \end{bmatrix}$$

On the other hand, the distributed, TPR-based compositional representation is given by:

$$\psi_{tpr}(x) = \tilde{\psi}(\text{purple/colour}) + \tilde{\psi}(\text{square/shape})$$
$$= \xi_F(\text{purple}) \otimes \xi_R(\text{colour}) + \xi_F(\text{square}) \otimes \xi_R(\text{shape})$$
$$= \begin{bmatrix} 1 \\ 1 \\ 2 \\ 2 \end{bmatrix} + \begin{bmatrix} 2 \\ 3 \\ 2 \\ 3 \end{bmatrix} = \begin{bmatrix} 3 \\ 4 \\ 3 \\ 4 \end{bmatrix}$$

Note that *both* the symbolic and distributed, TPR-based representations share the same underlying representational space, i.e., $\mathbb{R}^4$.

1. **Gradient Flow and Learning**[6]: The *symbolic* compositional representation allocates the FoVs to 2 distinct, independent slots of $\psi_d(x)$. This creates discrete boundaries between the first two and last two dimensions of the representation, which may potentially complicate gradient-based optimisation for the following reasons:

   (a) **Discrete Boundaries and Restricted Gradient Flow**: In the symbolic compositional representation, each FoV is confined to specific, non-overlapping dimensions of the representation. This type of assignment means that updates to a particular FoV requires restricting gradient flow exclusively to the dimensions associated with its designated slot. For example, updating the FoV for square/shape requires modifying the last two dimensions (i.e., permitting gradient flow through these two dimensions), while keeping the first two dimensions unchanged (i.e., completely restricting gradient flow through these two dimensions). This constrains gradient flow across *all* dimensions of representation. Mathematically, the gradient update for the above example (modifying square/shape) can be expressed in the symbolic case as:

   $$-\eta \nabla_{\theta_s} L(\theta_s) = -\eta \begin{bmatrix} 0 \\ 0 \\ a \\ b \end{bmatrix} \text{, where } a, b \in \mathbb{R}$$

   which categorically restricts gradient flow dimensionwise across the underlying representation space, $\mathbb{R}^4$.

   (b) **Abrupt, Discontinuous Shifts in the Representational Space**: When multiple FoVs are updated consecutively, the symbolic, slot-based representational form may induce sudden and discontinuous changes in the representational space. For instance, updating purple/colour at time $t$ and then shape/square at $t+1$ produces:

   $$\psi_s(x)_t = \psi_s(x)_{t-1} - \eta \begin{bmatrix} 0 \\ 0 \\ a \\ b \end{bmatrix}$$

   $$\psi_s(x)_{t+1} = \psi_s(x)_t - \eta \begin{bmatrix} c \\ d \\ 0 \\ 0 \end{bmatrix}$$

   These abrupt shifts in the representation space may contribute to unstable and fragmented learning dynamics, potentially hindering the convergence of optimisation algorithms.

---

[6]Please note there is a small error in the main Author Rebuttal https://openreview.net/forum?id=oEVsxVdush&noteId=GiOPkcNVts in L16 under section '1) Incompatibility between disentangled representations and deep learning's continuous vector spaces', where the tensor product is mistakenly taken over $\psi_{col}(purple) \otimes \psi_{sh}(square)$ to produce the representation, instead of computing $\xi_F(purple) \otimes \xi_R(col) + \xi_F(square) \otimes \xi_R(sh)$, but the conclusion remains exactly the same. (Note that $\psi_{col}(purple) = \xi_F(purple) \otimes \xi_R(col)$ and $\psi_{sh}(square) = \xi_F(square) \otimes \xi_F(sh)$ should *both* be 4-dimensional, *not* 2-dimensional.

In contrast, the *continuous, distributed*, TPR-based compositional representation integrates the FoVs continuously into all *dimensions* of the representation (i.e., $\tilde{\psi}(\text{purple/colour}) = (1\ 1\ 2\ 2)^T$ and $\tilde{\psi}(\text{square/shape}) = (2\ 3\ 2\ 3)^T$ are summed together in $\mathbb{R}^4$), eliminating the necessary existence of dimension-wise slots with non-differentiable boundaries for each FoV. This alleviates the previously mentioned problems:

(a) **Smoother Gradient Flow**: The inherently *distributed* structure of the TPR allows gradients to flow *freely* across all dimensions of the representational space, $\mathbb{R}^4$. Updating the FoV for square/shape involves modifying all relevant dimensions simultaneously, avoiding categorical dimension-wise restrictions and potentially allowing for more flexible and effective gradient-based optimisation as FoVs can be updated in a *coordinated* manner (note that backpropagating through the dimensions corresponding to square/shape *can* also affect purple/colour, as the same dimensions are occupied):

$$-\eta \nabla_{\theta_c} L(\theta_c) = -\eta \begin{bmatrix} \hat{a} \\ \hat{b} \\ \hat{c} \\ \hat{d} \end{bmatrix}, \text{ where } \hat{a}, \hat{b}, \hat{c}, \hat{d} \in \mathbb{R}$$

(b) **Smoother Learning Dynamics**: By forgoing the slot-based structure, a distributed encoding of compositional structure ensures that update transitions between FoVs do not cause sudden shifts in the representational space. For example:

$$\psi_{tpr}(x)_t = \psi_{tpr}(x)_{t-1} - \eta \begin{bmatrix} \hat{a}_{t-1} \\ \hat{b}_{t-1} \\ \hat{c}_{t-1} \\ \hat{d}_{t-1} \end{bmatrix}$$

$$\psi_{tpr}(x)_{t+1} = \psi_{tpr}(x)_t - \eta \begin{bmatrix} \hat{a}_t \\ \hat{b}_t \\ \hat{c}_t \\ \hat{d}_t \end{bmatrix},$$

potentially promoting more stable and reliable convergence during training.

2. **Representational Expressivity**: In the example above, both $\psi_s(x)$ and $\psi_{tpr}(x)$ reside in $\mathbb{R}^4$. However, the symbolic, slot-based compositional representation, $\psi_s(x)$ restricts each FoV to a specific, 2-dimensional subset (i.e., $\psi_{\text{colour}}(\text{purple}), \psi_{\text{shape}}(\text{square})$ each belong to $\mathbb{R}^2$). This rigid allocation limits expressivity as each FoV is confined to a subset of available dimensions, preventing the full utilisation of $\mathbb{R}^4$. In contrast, an inherently distributed approach allows each FoV – such as $\tilde{\psi}(\text{purple/colour}), \tilde{\psi}(\text{square/shape})$ – to exist within the same $\mathbb{R}^4$ space as the overall representation, $\psi_{tpr}(x)$. This allows the representation learner to encode FoVs as combinations of the entire set of 4-dimensional basis vectors spanning the representational space, offering enhanced expressivity.

3. **Robustness to Noise**: In symbolic compositional representations, the impact of dimension-wise noise has a localised sensitivity: since each FoV is isolated within its designated slot, noise directly degrades the representation of the corresponding FoV. For example, a small perturbation in the third dimension (assigned to square/shape) has a localised, direct effect only on this FoV. In contrast, for the inherently distributed TPR-based structure, noise introduced to any dimension of the representation is *distributed* across multiple FoVs. For example, for $\psi_{tpr}(x)$, the same perturbation in the third dimension will have its impact distributed across both square/shape and purple/colour, as both these FoVs contribute to that dimension.

## A.4 TPR's Distributed Nature and the Equivalence between Degenerate TPRs and Symbolic Representations

The TPR offers an inherently *distributed* representation of compositional structure, allowing information from multiple FoVs to intermingle across the same dimensions of the representational space.

Here, we elaborate on 2 critical properties of the TPR in relation to this notion of a *distributed* embedding of compositional structure.

1. **Equivalence of Degenerate TPRs to Concatenated Fillers**: While the TPR naturally *supports* a distributed encoding of compositional structure, it is worth noting that when the role embedding vectors are aligned with the canonical basis vectors, the TPR reduces to concatenated filler vectors, thereby reducing to a symbolic compositional representation where the representational components $\psi_i(a_i)$ correspond to filler embeddings. In contrast to the TPR, however, it is impossible for a symbolic compositional representations to encode compositional structure in a distributed fashion.

2. **Distributed Representation through Dense Role Vectors**: In the general case where the role vectors are linearly independent (but do not correspond to the canonical basis), provided that the role vectors are *dense*, then TPRs facilitate a truly distributed representation by allowing each dimension of the representation to encapsulate a summation of information from multiple fillers. Such a representation is *not* reducible to a dimension-wise concatenation of fillers, as in the degenerate TPR case.

We now prove both statements:

**Equivalence of Degenerate TPRs to Concatenated Fillers:**

Let $\{e_1, e_2, \ldots, e_{D_R}\}$ denote the canonical basis vectors in $\mathbb{R}^{D_R}$, where each $e_k$ has a 1 in the $k$-th position and 0s elsewhere.

Assume that each role vector $r_i = e_i$ for $i = 1, 2, \ldots, N_R$.

The tensor product $f_{m(i)} \otimes r_i$ results in a matrix where the $i$-th column is $f_{m(i)}$ and all other columns are zero vectors:

$$f_{m(i)} \otimes r_i = f_{m(i)} e_i^\top = \begin{bmatrix} 0 & \cdots & f_i^1 & \cdots & 0 \\ 0 & \cdots & f_i^2 & \cdots & 0 \\ \vdots & & \vdots & & \vdots \\ 0 & \cdots & f_i^{D_F} & \cdots & 0 \end{bmatrix}$$

Summing over all $N_R$ role-filler bindings in $\beta(x)$ gives the required result:

$$\psi_{tpr}(x) = \sum_{i=1} f_{m(i)} \otimes e_i = \begin{bmatrix} f_1^1 & f_2^1 & \cdots & f_{D_R}^1 \\ f_1^2 & f_2^2 & \cdots & f_{D_R}^2 \\ \vdots & \vdots & \ddots & \vdots \\ f_1^{D_F} & f_2^{D_F} & \cdots & f_{D_R}^{D_F} \end{bmatrix}$$

This shows that in the case where the role vectors correspond to the canonical basis vectors, the the TPR effectively concatenates the filler vectors $f_i$ along distinct, non-overlapping subsets of representational dimensions. Consequently, the TPR $\psi_{tpr}(x)$ behaves equivalently to a symbolic, slot-based compositional structure where each filler occupies a unique representational slot without interacting with other fillers, hence validating the equivalence of degenerate TPRs to concatenated fillers.

**Distributed Representation through Dense Role Vectors:**

Assume that the role vectors $\{r_1, r_2, \ldots, r_N\}$ in $\mathbb{R}^{D_R}$ are dense and linearly independent but are not aligned with the canonical basis vectors.

Each tensor product $f_{m(i)} \otimes r_i$ yields a matrix where each element $(j, k)$ is $f_{m(i)}^j r_i^k$. Summing across all role-filler bindings yields:

$$\psi_{tpr}(x) = \sum_{i=1} f_m(i) \otimes r_i = \begin{bmatrix} \sum_{i=1} f_{m(i)}^1 r_i^1 & \sum_{i=1} f_{m(i)}^1 r_i^2 & \cdots & \sum_{i=1}^N f_{m(i)}^1 r_i^{D_R} \\ \sum_{i=1}^N f_{m(i)}^2 r_i^1 & \sum_{i=1}^N f_{m(i)}^2 r_i^2 & \cdots & \sum_{i=1}^N f_{m(i)}^2 r_i^{D_R} \\ \vdots & \vdots & \ddots & \vdots \\ \sum_{i=1}^N f_{m(i)}^{D_F} r_i^1 & \sum_{i=1}^N f_{m(i)}^{D_F} r_i^2 & \cdots & \sum_{i=1}^N f_{m(i)}^{D_F} r_i^{D_R} \end{bmatrix}$$

Given that role vectors are dense, each element $\psi_{tpr}^{j,k}$ incorporates contributions from multiple fillers $f_{m(i)}^j$, each weighted by the respective component $r_i^k$ of their role vectors. This intermingling ensures

that information from different filler vectors are distributed across the same dimensions of $\psi_{tpr}(x)$, rather than being confined to distinct subsets of $\psi_{tpr}(x)$'s dimensions, as in the degenerate TPR case.

## B  Soft TPR Framework

In this section, we provide additional information regarding our Soft TPR framework.

### B.1  Shortcomings of the TPR and How Soft TPR Helps

In this subsection, we discuss potential practical limitations of traditional TPR due to their rigid specification (Eq 1) and demonstrate how Soft TPR addresses these issues using a concrete example.

Consider the following set of roles $R = \{\text{shape}, \text{colour}\}$, and fillers $F = \{\text{red}, \text{blue}, \text{square}\}$, with embedding functions $\xi_R : R \to \mathbb{R}^2$ and $\xi_F : F \to \mathbb{R}^3$ defined as:

$$\xi_R(\text{shape}) = \begin{bmatrix} 1 \\ 0 \end{bmatrix},$$

$$\xi_R(\text{colour}) = \begin{bmatrix} 1 \\ 1 \end{bmatrix},$$

$$\xi_F(\text{red}) = \begin{bmatrix} 1 \\ 2 \\ 3 \end{bmatrix},$$

$$\xi_F(\text{blue}) = \begin{bmatrix} 2 \\ 2 \\ 3 \end{bmatrix},$$

$$\xi_F(\text{square}) = \begin{bmatrix} 0 \\ 0 \\ 1 \end{bmatrix},$$

1. **Discrete Mapping**: The set of possible TPRs $T$ generated by these roles and fillers is limited to discrete points. For instance:

$$\psi_{tpr}(x_{\text{red\_square}}) = \xi_F(\text{red}) \otimes \xi_R(\text{colour}) + \xi_F(\text{square}) \otimes \xi_R(\text{shape})$$

$$= \begin{bmatrix} 1 \\ 2 \\ 3 \end{bmatrix} \otimes \begin{bmatrix} 1 \\ 1 \end{bmatrix} + \begin{bmatrix} 0 \\ 0 \\ 1 \end{bmatrix} \otimes \begin{bmatrix} 1 \\ 0 \end{bmatrix} \cong \begin{bmatrix} 1 \\ 2 \\ 3 \\ 1 \\ 2 \\ 3 \end{bmatrix} + \begin{bmatrix} 0 \\ 0 \\ 1 \\ 0 \\ 0 \\ 0 \end{bmatrix}$$

$$= \begin{bmatrix} 1 \\ 2 \\ 4 \\ 1 \\ 2 \\ 3 \end{bmatrix}$$

$$\psi_{tpr}(x_{\text{blue\_square}}) = \xi_F(\text{blue}) \otimes \xi_R(\text{colour}) + \xi_F(\text{square}) \otimes \xi_R(\text{shape})$$

$$= \begin{bmatrix} 2 \\ 2 \\ 3 \end{bmatrix} \otimes \begin{bmatrix} 1 \\ 1 \end{bmatrix} + \begin{bmatrix} 0 \\ 0 \\ 1 \end{bmatrix} \otimes \begin{bmatrix} 1 \\ 0 \end{bmatrix} \cong \begin{bmatrix} 2 \\ 2 \\ 3 \\ 2 \\ 2 \\ 3 \end{bmatrix} + \begin{bmatrix} 0 \\ 0 \\ 1 \\ 0 \\ 0 \\ 0 \end{bmatrix}$$

$$= \begin{bmatrix} 2 \\ 2 \\ 4 \\ 2 \\ 2 \\ 3 \end{bmatrix}$$

Thus, $T = \{(2\ 2\ 4\ 2\ 2\ 3)^T, (1\ 2\ 4\ 1\ 2\ 3)^T\}$, a discrete subset of $\mathbb{R}^6$. By relaxing the TPR specification to the *Soft TPR*, we allow any $z \in \mathbb{R}^6$ within an $\epsilon$-neighbourhood of some $\psi_{tpr}$ in $T$, thereby forming continuous regions defined as: $T_S := \{(2\ 2\ 4\ 2\ 2\ 3)^T + \alpha, (1\ 2\ 4\ 1\ 2\ 3)^T + \alpha : |\alpha| < \epsilon\}$. This relaxation significantly expands the representational space, as $T_S$ has strictly more points than $T$. Consequently, learning theoretically becomes easier because there are more functions parameterising the mapping from the observed data to $T_S$ compared to $T$. Furthermore, in contrast to $T$, which contains discrete, singular points scattered throughout $\mathbb{R}^6$, $T_S$ comprises continuous spherical regions centered around each $\psi_{tpr} \in T$. These factors collectively make the Soft TPR representation potentially easier to learn and extract information from compared to the TPR. This advantage is reflected in our empirical results in Section C.6.3, where the Soft TPR demonstrates 1) greater representation learner convergence, 2) superior sample efficiency for downstream tasks, and 3) superior raw downstream performance in the low sample regime compared to the traditional TPR. Additionally, our ablation results in Table 43 indicate that imposing stricter constraints on the encoder to produce representations in the form of explicit TPRs substantially impairs the disentanglement of the resulting representations.

2. **Quasi-Compositional Structure**: The traditional TPR enforces a strict algebraic definition of compositionality, specifically of the form $\sum_i \xi_F(f_{m(i)}) \otimes \xi_R(r_i)$. However, even in algebraically-structured domains such as natural language, compositional structures do not always adhere to this rigid specification. For instance, French liaison consonants consist of a weighted sum of fillers bound to a single role [9], which deviates from the strict TPR formulation. The Soft TPR's continuous relaxation of this constraint permits the representation of structures that only *approximately* satisfy the TPR's algebraic definition of compositional structure as a set of discrete role-filler bindings. Consequently, Soft TPRs can encode forms of quasi-compositionality that are not strictly algebraically precise or directly amenable to TPR-based symbolic expression (see B.2 for a proof).

3. **Duality**: Although Soft TPRs employ a relaxed structural specification to encode *quasi*-compositional structures, they inherently preserve the exact algebraic form of compositionality through their corresponding explicit TPRs, $\psi_{tpr}^*$, via quantisation using the TPR Decoder. In other words, while Soft TPRs facilitate approximate and flexible compositional representations, they retain the capability to revert to the precise algebraic structures defined by traditional TPRs when necessary. This dual functionality ensures that Soft TPRs strike a balance between the flexibility required to represent complex, approximately compositional structures—which benefits learning through relaxed representational constraints—and the preservation of exact algebraic forms essential for symbolic manipulation and interpretation.

4. **Serial Construction**: Building explicit TPRs requires that the embedded fillers and roles comprising a binding (i.e. $(\xi_F(f_{m(i)}), \xi_R(r_i))$) are first *tokened* before the entire compositional representation, $\psi_{tpr}(x)$ can be produced [19, 23, 28, 34, 38, 51, 52]. This sort of sequential approach, where constituents must be tokened before the compositional representation can be formed, is a key characteristic of the symbolic representation of compositional structure [45]. In contrast, the Soft TPR allows the Encoder to produce any *arbitrary* element of $V_F \otimes V_R$ (in this case $\cong \mathbb{R}^6$), provided that the sufficient closeness requirement holds. Thus, once the Encoder is trained, it is theoretically possible for the Soft TPR Autoencoder

to generate approximately compositional representations by mapping directly from the data to a Soft TPR in a single step, without needing to token any representational constituents.

## B.2  Proof that Soft TPRs Do Not Necessarily Have the Explicit Form of a TPR

It is intuitive that Soft TPRs, due to their continuous relaxation, do not necessarily conform to the explicit algebraic form $\sum_i \xi_F(f_{m(i)}) \otimes \xi_R(r_i)$ required by traditional TPRs. We provide a formal proof for completeness.

*Proof.* Let $\xi_R : R \to V_R$ and $\xi_F : F \to V_F$ be fixed role and filler embedding functions, respectively. A traditional TPR is defined as:

$$\psi_{tpr}(x) = \sum_i \xi_F(f_{m(i)}) \otimes \xi_R(r_i),$$

where $m(i)$ is a matching function assigning each role $r_i$ to a filler $f_{m(i)}$.

A Soft TPR $z$ is any element within an $\epsilon$-neighborhood of some traditional TPR in $V_F \otimes V_R$:

$$z \in \mathcal{B}_\epsilon(\psi_{tpr}) = \left\{ z \in V_F \otimes V_R \mid \|z - \psi_{tpr}\| < \epsilon \right\},$$

where $\epsilon > 0$.

Suppose, for contradiction, that every Soft TPR $z \in \mathcal{B}_\epsilon(\psi_{tpr})$ can be expressed as another traditional TPR with a different matching function $m'$:

$$z = \sum_i \xi_F(f_{m'(i)}) \otimes \xi_R(r_i).$$

Given that $\xi_R$ and $\xi_F$ are fixed, varying $m'$ only changes the assignment of fillers to roles without altering the underlying embeddings.

Let $\delta = z - \psi_{tpr}$, where $\|\delta\| < \epsilon$. Substituting, we obtain:

$$\sum_i \xi_F(f_{m'(i)}) \otimes \xi_R(r_i) = \psi_{tpr} + \delta.$$

Since $\delta \neq 0$, this equality implies:

$$\sum_i \xi_F(f_{m'(i)}) \otimes \xi_R(r_i) \neq \psi_{tpr}.$$

However, because $\xi_R$ and $\xi_F$ are fixed, the only way to satisfy this equality is if the matching function $m'$ compensates exactly for $\delta$. This is generally impossible due to the following constraints:

1. **Fixed Embeddings**: $m'$ can only reassign fillers to roles without altering $\xi_R(r_i)$ or $\xi_F(f_{m(i)})$. The fixed $\xi_F(f_{m(i)})$ restrict the possible variations, making it infeasible to account for arbitrary $\delta$ through mere reassignment.

2. **Dimensionality and Span**: In high-dimensional spaces $V_F \otimes V_R$, $\delta$ may lie outside the span of all traditional TPRs generated by varying $m'$, especially as $\epsilon$ increases.

Consequently, the perturbation $\delta$ cannot be exactly compensated by any reassignment $m'$, leading to a contradiction. Therefore, Soft TPRs with non-zero $\epsilon$-distances from their corresponding quantised TPRs, $\psi_{tpr}^*$, are also **not necessarily expressible** as other TPRs in the same underlying space.

$\square$

The introduction of a continuous relaxation in Soft TPRs ensures that they do not strictly adhere to the traditional TPR's explicit algebraic form. This allows Soft TPRs to represent quasi-compositional structures that deviate from strict algebraic compositionality, which extends distributed compositional representations beyond the formal, explicitly algebraic domains required by traditional TPRs.

### B.3 Alternative Formulations

In contrast to the greedily optimal analytic form of $\psi^*_{tpr}$ derived in Equation 5, a globally optimal TPR with respect to $||z - \psi_{tpr}||_F$ can also be formulated. By fixing the sets of fillers $F$ and roles $R$, along with the embedding functions $\xi_F : F \to \mathbb{R}^{D_F}$ and $\xi_R : R \to \mathbb{R}^{D_R}$, the only degree of freedom lies in defining $\mathcal{M}^*$, the set of role-filler matching functions that specify permissible role-filler binding decompositions for $x \in X$. The globally optimal TPR that best approximates $z$, denoted $\psi^{opt}_{tpr}$, is given by:

$$\psi^{opt}_{tpr} := \underset{\psi_{tpr} \in \mathcal{T}}{\arg\min} \, ||z - \psi_{tpr}||_F, \tag{10}$$

where $\mathcal{T} := \left\{ \sum_i \xi_F(f_{m(i)}) \otimes \xi_R(r_i) \right\}_{m \in \mathcal{M}^*}$ denotes the set of all possible traditional TPRs defined by the choices of $F$, $R$, $\xi_F$, $\xi_R$, and $\mathcal{M}^*$. For clarity, we omit the dependency of both $\psi_{tpr}$ and $m$ on $x$.

If $\mathcal{M}^*$ is a restricted subset of the entire set of matching functions $\mathcal{M}$, meaning there are constraints on which fillers can be bound to which roles, then solving Equation 10 is NP-complete. Conversely, if $\mathcal{M}^* = \mathcal{M}$, allowing any filler to be bound to any role with possible reuse, the optimisation simplifies significantly. In this unrestricted scenario, the problem reduces to independently assigning each role to the filler that minimises the local contribution to $||z - \psi_{tpr}||_F$, which can be solved efficiently with a time complexity of $O(N_R \times N_F)$. Acknowledging the impracticality of exact global optimisation, and guided by the intuition that $\psi^*_{tpr}$ should have an explicit dependency on $z$, we propose the greedily optimal definition of $\psi^*_{tpr}$ in Equation 5, which links the unbound *soft* fillers of $z$, $\{\tilde{f}_i\}$, to $\psi^*_{tpr}$.

### B.4 Soft TPR Autoencoder

Defining $m(i) := \arg\min_j ||\tilde{f}_k - \xi_F(f_j)||_2$ in Equation 5 allows any filler $f_j$ to bind to a role $r_i$, which may not preserve the ground-truth semantics of the role-filler bindings in the image. To address this, the Soft TPR Autoencoder targets two objectives jointly: 1) *Representational Form*, ensuring the output resembles a Soft TPR, and 2) *Representational Content*, ensuring sensible role-filler bindings reflecting ground-truth semantics.

#### B.4.1 Representational Form

We now describe how our method satisfies the representational form property. As detailed in Section 4.2, we penalize the Euclidean distance between the encoder $E$'s output $z$ and the explicit TPR $\psi^*_{tpr}$ defined in Equation 5. This penalty encourages the Soft TPR Autoencoder to produce encodings that resemble a Soft TPR by enforcing $||z - \psi_{tpr}||_2 < \epsilon$[7]. To obtain $\psi^*_{tpr}$, we employ our TPR decoder, which consists of three modules: **1) Unbinding**, **2) Quantization**, and **3) TPR Construction**.

**1) Unbinding**: Although multiple roles can bind to the same filler (e.g., object colour/magenta and floor colour/magenta), each role represents an independent concept type. Therefore, we use linearly independent embedding vectors for roles to ensure the recoverability of embedded components $(\xi_F(f_{m(i)}), \xi_R(r_i))$ from the TPR. Typically, a randomly initialised role embedding matrix $M_{\xi_R} \in \mathbb{R}^{D_R \times N_R}$ with $D_R \gg N_R$ will have linearly independent columns. To maintain this property during training, it is possible to apply orthogonality regularisation: $||M_{\xi_R}^T M_{\xi_R} - I||_F$.

However, using arbitrary linearly independent role embeddings poses a computational challenge in obtaining $U$, the (left) inverse of $M_{\xi_R}$, necessary for unbinding vectors $\{u_i\}$. To address this, we utilise semi-orthogonal matrices $A \in \mathbb{R}^{d \times n}$ where $A^T A = I$. In this case, $U^T = M_{\xi_R}$, and the unbinding vector $u_i$ is simply the role embedding $\xi_R(r_i)$. Thus, unbinding a soft filler from $z$ bound to role $r_i$ involves taking the tensor inner product between $z$ and $\xi_R(r_i)$.

---

[7]We assume vector spaces over the reals and utilise the isomorphism $\mathbb{R}^{D_F} \otimes \mathbb{R}^{D_R} \cong \mathbb{R}^{D_F \cdot D_R}$ to integrate the TPR framework with the autoencoding framework.

To implement this, our unbinding module initialises $M_{\xi_R}$ as a dense, semi-orthogonal matrix using torch.nn.utils.parameterizations.orthogonal and keeps $M_{\xi_R}$ fixed during training to preserve its semi-orthogonal property [8].

Now, we proceed to formally prove the aforementioned semi-orthogonality properties.

We first prove that for a (left-invertible) semi-orthogonal matrix, $A \in \mathbb{R}^{d \times n}$, $A^T A = I_{n \times n}$.

*Proof.* Let $A$ denote a (left-invertible) semi-orthogonal matrix of dimension $\mathbb{R}^{d \times n}$. Writing $A$ out using its columns, $\{a_1, \ldots, a_n\}$, we have:

$$A^T A = \begin{pmatrix} a_1^T \\ \vdots \\ a_n^T \end{pmatrix} \begin{pmatrix} a_1 & \cdots & a_n \end{pmatrix}$$

As the columns $\{a_1, \ldots, a_n\}$ are orthonormal, we have that: $a_i \cdot a_j = \delta_{ij}$, where $\cdot$ denotes the dot product, and

$$\delta_{ij} = \begin{cases} 1 & i = j \\ 0 & \text{otherwise,} \end{cases}$$

from which the result clearly follows. $\square$

For completeness, we also prove that $u_i$ corresponds to the $i$-th role embedding vector, though this follows rather trivially from the properties of semi-orthogonal matrices, and the definition of $u_i$.

*Proof.* Let $M_{\xi_R}$ be a (left-invertible) semi-orthogonal matrix. Then, from the properties of semi-orthogonal matrices proved earlier, we have that $M_{\xi_R}^T M_{\xi_R} = I$, and so, $U^T = \left( M_{\xi_R}^T \right)^T = M_{\xi_R}$. Hence, the $i$-th column of $U^T$, which corresponds to the unbinding vector, $u_i$, by definition, is simply $(M_{\xi_R})_{:i} = \xi_R(r_i)$, the $i$-th role embedding vector. $\square$

## 2) Quantisation

The quantisation module utilises the VQ-VAE algorithm [11] to learn the filler embedding matrix $M_{\xi_F}$ and to quantise the soft filler embeddings $\{\tilde{f}_i\}$ into explicit embeddings $\{\xi_F(f_i)\}$ based on the closest Euclidean distances. This process aligns with the matching function $m$ in Equation 5, where each soft filler $\tilde{f}_i$ is assigned to the nearest explicit filler in $M_{\xi_F}$. Since quantisation involves an argmax operation and is non-differentiable, VQ-VAE employs an $L_2$ *codebook loss* to adjust the embedding vectors towards the soft fillers and a *commitment loss* to ensure the encoder commits to specific embeddings, preventing the embedding space from expanding indefinitely.

For further details on the quantisation process, refer to [11].

## 3) TPR Construction

The TPR construction module is parameter-free and deterministically binds the quantised filler embeddings $\{\xi_F(f_{m(i)})\}$ to their corresponding role embeddings $\{\xi_R(r_i)\}$ from $M_{\xi_R}$, forming the explicit TPR:

$$\psi_{tpr}^* = \sum_i \xi_F(f_{m(i)}) \otimes \xi_R(r_i).$$

### B.4.2 Representational Content

Here, we explicitly define the 'swapped' TPRs, $\psi_{tpr}^s(x)$ and $\psi_{tpr}^s(x')$, used in our weakly supervised reconstruction loss (second term of Equation 7). For image pairs $(x, x')$ sharing all role-filler bindings

---

[8]Astute readers may notice that the explicit TPRs our representation learner uses are rotationally isomorphic to the degenerate TPR case (i.e., concatenation of fillers). See C.6.1 for further discussion.

except for role $r_i$, we construct the swapped TPRs by exchanging the quantised filler embeddings for $r_i$:

$$\psi_{tpr}^s(x) = \sum_{j \neq i} \xi_F(f_{m(j)}) \otimes \xi_R(r_j) + \xi_F(f_{m'(i)}) \otimes \xi_R(r_i), \tag{11}$$

$$\psi_{tpr}^s(x') = \sum_{j \neq i} \xi_F(f_{m'(j)}) \otimes \xi_R(r_j) + \xi_F(f_{m(i)}) \otimes \xi_R(r_i), \tag{12}$$

where $m$ and $m'$ are the matching functions for $x$ and $x'$, respectively.

### B.4.3 Model Architecture

We now provide concrete details of our model architecture in Table 8, 9 and Table 10. It is worth noting that Soft TPR Autoencoder only adds an additional $D_F \cdot N_F$ parameters to a standard (variational) autoencoding framework consisting of the encoder and decoder as the only learnable parameters we introduce are in the filler embedding matrix, $M_{\xi_F}$.

For all our experiments, we use the same general architecture for the encoder, $E$, the TPR decoder, and the decoder, $D$.

Table 8: Encoder $(E)$ architecture.

| Encoder |
| --- |
| Input $64 \times 64 \times c$ |
| Conv 32, $4 \times 4$, stride $= 2$, padding $= 1$ |
| BatchNorm 32 |
| ReLU 32 |
| Conv 64, $4 \times 4$, stride $= 2$, padding $= 1$ |
| BatchNorm 64 |
| ReLU 64 |
| Conv 256, $4 \times 4$, stride $= 2$, padding $= 1$ |
| BatchNorm 256 |
| ReLU 256 |
| Conv 512, $4 \times 4$, stride $= 2$, padding $= 1$ |
| BatchNorm 512 |
| ReLU 512 |
| Flatten $512 \times 4 \times 4$ |
| Linear 1024 |
| ReLU 1024 |
| Linear 512 |
| ReLU 512 |
| Linear 512 |
| ReLU 512 |
| Linear $D_F \cdot D_R$ |

Table 9: TPR Decoder architecture.

| TPR Decoder |
| --- |
| Role embedding matrix (*fixed*) $D_R \times N_R$ |
| Filler embedding matrix $D_F \times N_F$ |

Table 10: Decoder $(D)$ architecture.

| Decoder |
| --- |
| Input $D_F \cdot D_R$ |
| ConvTranspose 512, $4 \times 4$, $\mathrm{stride} = 2$, $\mathrm{padding} = 1$ |
| BatchNorm 512 |
| ReLU 512 |
| ConvTranspose 256, $4 \times 4$, $\mathrm{stride} = 2$, $\mathrm{padding} = 1$ |
| BatchNorm 256 |
| ReLU 256 |
| ConvTranspose 64, $4 \times 4$, $\mathrm{stride} = 2$, $\mathrm{padding} = 1$ |
| BatchNorm 64 |
| ReLU 64 |
| ConvTranspose 32, $4 \times 4$, $\mathrm{stride} = 2$, $\mathrm{padding} = 1$ |
| BatchNorm 32 |
| ReLU 32 |
| ConvTranspose 3, $4 \times 4$, $\mathrm{stride} = 2$, $\mathrm{padding} = 1$ |

## B.5   Model Hyperparameters and Hyperparameter Tuning

The Soft TPR Autoencoder has the following tunable hyperparameters:

**Architectural Hyperparameters**:

$$N_R, \quad N_F, \quad D_R, \quad D_F$$

These correspond to the number of role and filler embedding vectors and the dimensionalities of their respective embedding spaces.

**Loss Function Hyperparameters**:

$$\beta \quad \text{(which weights the VQ-VAE commitment loss as per Equation 6 and)}$$
$$\lambda_1, \quad \lambda_2 \quad \text{(which weigh the unsupervised and supervised losses in Equation 7)}$$

Following the VQ-VAE framework [11], we set $\beta$, the coefficient for the commitment loss, to 0.5. To ensure fair comparisons with scalar-tokened generative baselines and COMET, which assume 10 FoV types (and VCT assumes 20), we fix $N_R = 10$.

We optimise the remaining hyperparameters using the Weights and Biases (WandB) hyperparameter sweep framework. The optimisation employs Bayesian search with the unsupervised MSE reconstruction loss from Equation 6 as the criterion. During this optimization, models are trained for between 50,000 and 100,000 iterations.

The final hyperparameter values for the Soft TPR Autoencoder are listed in Table 11. For ablation experiments demonstrating the model's robustness to different hyperparameter settings, refer to Section C.6.4.

Table 11: Hyperparameter values.

| Hyperparameter | Cars3D | Shapes3D | MPI3D |
| --- | --- | --- | --- |
| | Architectural hyperparameters | | |
| $D_R$ | 12 | 16 | 12 |
| $N_R$ *(fixed)* | 10 | 10 | 10 |
| $D_F$ | 128 | 32 | 32 |
| $N_F$ | 106 | 57 | 50 |
| | Loss function hyperparameters | | |
| $\lambda_1$ | 0.00024 | 0.00091 | 0.0000 |
| $\lambda_2$ | 0.02200 | 0.00228 | 1.16050 |
| $\beta$ *(fixed)* | 0.5 | 0.5 | 0.5 |

Our model is implemented in Pytorch [27] and trained using the Adam optimiser on the loss corresponding to 7. We use a learning rate of $1\mathrm{e}{-4}$, and the default setting of $(\beta_1, \beta_2) = (0.9, 0.999)$ across all instances of model training.

# C  Results

In this section, we provide additional details about our experiments, and present additional results.

## C.1  Datasets

For disentanglement, we utilise standard disentanglement datasets: Cars3D [4], Shapes3D [24], and the 'real' variant of MPI3D [21], containing 3, 6, and 7 ground-truth FoVs respectively. These datasets are procedurally generated via taking the Cartesian product of all possible FoV values per FoV type. Metadata is provided in Tables 12, 13 and 14, with continuous-valued FoV types marked by $^{\ddagger}$. We treat colour FoVs as continuous variables due to their 10 linearly spaced, naturally ordered values, in line with [42].

<table>
<tr><td colspan="2">Table 12: Cars3D dataset</td><td colspan="2">Table 13: Shapes3D dataset</td><td colspan="2">Table 14: MPI3D dataset</td></tr>
<tr><td>FoV types</td><td>FoV values</td><td>FoV types</td><td>FoV values</td><td>FoV types</td><td>FoV values</td></tr>
<tr><td>Car type</td><td>199 types</td><td>Floor colour$^{\ddagger}$</td><td>10 colours</td><td>Object colour</td><td>6 colours</td></tr>
<tr><td>Rotation$^{\ddagger}$</td><td>24 angles</td><td>Wall colour$^{\ddagger}$</td><td>10 colours</td><td>Object shape</td><td>6 shapes</td></tr>
<tr><td>Camera elevation$^{\ddagger}$</td><td>4 types</td><td>Object colour$^{\ddagger}$</td><td>10 colours</td><td>Object size$^{\ddagger}$</td><td>2 sizes</td></tr>
<tr><td></td><td></td><td>Object size$^{\ddagger}$</td><td>8 sizes</td><td>Camera height</td><td>3 heights</td></tr>
<tr><td></td><td></td><td>Object type</td><td>4 shapes</td><td>Background colour</td><td>3 colours</td></tr>
<tr><td></td><td></td><td>Azimuth$^{\ddagger}$</td><td>15 angles</td><td>Horizontal axis$^{\ddagger}$</td><td>40 values</td></tr>
<tr><td></td><td></td><td></td><td></td><td>Vertical axis$^{\ddagger}$</td><td>40 values</td></tr>
</table>

For the downstream regression task, we use these disentanglement datasets and train downstream models to regress on each dataset's continuous-valued FoVs, as indicated by the $^{\ddagger}$ symbol in Tables 12, 13 and 14. For abstract visual reasoning [30], we employ the Raven's Progressive Matrix (RPM) style dataset from [30], where each sample consists of 8 context panels and 6 answer panels derived from Shapes3D images. The model must infer abstract visual relations from the context panels to select the correct answer among six options.

## C.2  Baseline Implementations and Experimental Settings

### C.2.1  Experiment Compute Resources

For the generative weakly supervised, scalar-valued baselines (i.e., AdaGVAE-k, GVAE, MLVAE, SlowVAE, the Shu model), and our model, we perform model training on a single Nvidia RTX4090 GPU. We also perform all associated experiments for these models (i.e., downstream model training, downstream model evaluation, disentanglement evaluation, etc.) on this single Nvidia RTX4090 GPU. Our model takes approximately 1.5 hours to fully train (i.e., run 200,000 iterations) on the Cars3D and Shapes dataset, and approximately 4.0 hours to fully train on the MPI3D dataset.

For the vector-valued baselines of COMET and VCT, we perform model training, and all associated experiments on a single Nvidia V100 GPU.

### C.2.2  Disentanglement Models

For the generative weakly supervised scalar-valued baselines (AdaGVAE-k, GVAE, MLVAE, SlowVAE, Shu model) and our model, training and all related experiments (downstream training, evaluation, disentanglement) were conducted on a single Nvidia RTX4090 GPU. Our model requires approximately 1.5 hours to train (200,000 iterations) on the Cars3D and Shapes datasets, and about 4.0 hours on the MPI3D dataset.

Vector-valued baselines COMET and VCT were trained and evaluated on a single Nvidia V100 GPU. We utilised the PyTorch-based open-source implementations from [42] for Ada-GVAE, GVAE, MLVAE, and SlowVAE, verified to reproduce reported results. Official implementations from the authors were used for COMET and VCT [36, 47]. For the GAN-based Shu model, we converted the TensorFlow implementation from [35] to PyTorch and ensured it reproduced official results. Baseline models employed recommended hyperparameters; otherwise, hyperparameter tuning was conducted using the same Weights and Biases (WandB) hyperparameter sweep as our model.

As discussed in Section 5, all weakly supervised baselines except Ada-GVAE assume access to $I$, the differing FoV between each pair $(x, x')$, matching our model's supervision level. To make Ada-GVAE

comparable, we modified it to receive the ground-truth $k = |I|$, the number of changing FoVs per sample, termed Ada-GVAE-k. This adjustment resulted in superior or comparable performance to the original Ada-GVAE. All models, except SlowVAE (which assumes all FoVs change), were trained with $k = 1$, ensuring consistent training conditions across models.

For each model's selected hyperparameters, including our own, we trained five models using five random seeds and aggregated the results. All models were trained for 200,000 iterations on each dataset.

### C.2.3 Downstream Models

For FoV regression, following [42], we use a simple, generic MLP model, with the architecture listed in Table 15. For each disentanglement dataset, the MLP regression model receives representations of images produced by a representation learner, and is trained to predict the ground-truth FoV values in a supervised fashion. Performance is measured using the $R^2$ score on a held out, randomly selected test set of 1,000 samples.

For each representation learning model instantiated with a random seed, we train 2 MLPs by uniformly sampling the number of output nodes in the first, second, and fifth layers as specified in Table 15, resulting in a total of 10 MLPs per learner (we use 5 random seeds). Reported results are averaged over these models. Notably, the MLPs have no prior knowledge of the representation type (e.g., scalar-tokened symbolic, vector-tokened symbolic, or fully continuous compositional) since no inductive biases or representation-specific optimisations are applied, aside from the supervised MSE loss.

Each MLP is trained using the Adam optimiser with an MSE loss, a learning rate of $1 \times 10^{-4}$, and default $(\beta_1, \beta_2) = (0.9, 0.999)$ settings.

Table 15: MLP architecture

| MLP |
| --- |
| Linear $d_1, d_1 \in [256, 512]$ |
| ReLU |
| Linear $d_2, d_2 \in [256, 512]$ |
| ReLU |
| Linear $\dim(z)$ |
| ReLU |
| Linear $\dim(z)$ |
| ReLU |
| Linear $d_3, d_3 \in [128, 256]$ |
| ReLU |
| Linear $k$ |

For the abstract visual reasoning task, following [30], we use the Wild Relation Network (WReN) model [18] on representations produced by representation learners to predict ground-truth answers for each RPM matrix. For each of model instances produced by a random seeds, we randomly sample 2 possible configurations of the WReN model by uniformly sampling either 256 or 512 for the edge MLP, $g$, and 128 or 256 hidden units for the graph MLP, $f$, in line with [30]. We however, fix the number of hidden layers in $g$ to 2, and $f$ in 1, representing the smallest possible number of hidden layers, to constrain the capacity of the WReN model. As we use 5 random seeds, and sample 2 configurations per seed, we produce 10 WReN models for each representation learner. All WReN models are trained using Adam optimisation on the BCE loss between the predicted logits and ground-truth labels, with a learning rate of $1e-5$ and the default setting of $(\beta_1, \beta_2) = (0.9, 0.999)$.

### C.2.4 Experimental Controls

To ensure that the performance improvements of our model are attributable to the distributed, flexible nature of the Soft TPR framework, we implement controls to eliminate confounding variables.

**Disentanglement Controls**

To ensure performance gains in disentanglement metrics (Tables 16 and 17) are not attributable to our model's additional parameters (with our model's the filler embedding matrix $M_{\xi_F}$, adding

13,568, 1,824, and 1,600 parameters for Cars3D, Shapes3D, and MPI3D respectively to a standard autoencoding framework), we adjust baseline models with fewer parameters. Specifically, for SlowVAE, Ada-GVAE-k, GVAE, ML-VAE, and the Shu model, we increase their parameters by adding filters or layers until they match or exceed our model's parameter count, denoted with $^*$ in Tables 16 and 17. VCT and COMET are substantially more parameter hungry (10+ million) than our model, so parameter controls are not applicable to these vector-valued symbolic compositional baselines. Consistent with [40], we observe that increasing parameters in these generative, scalar-tokened baselines does not enhance disentanglement performance. Therefore, we proceed with representation learning convergence experiments and downstream model tests using the original models, not the parameter-adjusted variants.

**Downstream Task Controls**

To ensure that any performance boosts on the downstream tasks of FoV regression and abstract visual reasoning are not attributable to our representation's increased dimensionality, we post-process representations produced by *all* models (including models producing higher-dimensional representations) to match the dimensionality of our representation. Specifically, generative, weakly supervised models with scalar-tokened representations produce representations with a dimension of 10 for all datasets, whereas our Soft TPRs have dimensions of 1536 (Cars3D), 512 (Shapes3D), and 320 (MPI3D) of our Soft TPRs, highlighting the necessity of such a control.

To perform this control, we apply separate methods for the scalar-tokened and vector-tokened models. For scalar-tokened models, we multiply each dimension $k$ of the latent vector, $l_k$ with an random embedding vector $e_k \in \mathbb{R}^d$ from an fixed, randomly initialised, semi-orthogonal embedding matrix $E \in \mathbb{R}^{d \times 10}$, and subsequently concatenate all multiplied random embedding vectors together to form the dimensionality-controlled representation, $(l_1 e_k, \dots, l_{10} e_{10}) \in \mathbb{R}^{10d}$. $d$ is a dataset specific integer that ensures that the size of the dimensionality-controlled representation is at least as small as the dimensionality of the Soft TPR representation for the given dataset, i.e., $d := \lceil \dim(z) \rceil / 10$, where $z$ is the Soft TPR representation produced by a given dataset. Note that we choose a semi-orthogonal embedding matrix with the intuition that this ensures the maximal distinguishability of each continguous subset of dimensions corresponding to $l_i e_i$.

For COMET and VCT, which both produce vector-tokened representations, we consider alternative methods of performing dimensionality-control. COMET's representations have a dimensionality of 640 for all datasets, and so, the model produces lower-dimensional representations than ours for Shapes3D and Cars3D, but not MPI3D. VCT, on the other hand, produces representations with dimension 5120, and so, produces higher-dimensional representations than ours for all datasets. For any vector-tokened representation with *higher* dimensionality than our representation, we apply PCA-based postprocessing to the model representation, reducing the dimensionality of each vector-tokened value in the representation to the required dimensionality, $d$, where $d := \lceil \dim(z) \rceil / \dim(z_{baseline})$, before concatenating all PCA-reduced vectors together to produce the modified representation. For any vector-tokened representation produced by COMET with lower dimensionality than ours, we apply a simple matrix multiplication between the representation, and a randomly initialised matrix of required dimensionality to embed the representation into the same-dimensional space as ours.

We denote the results produced by all such postprocessing by $^\dagger$ in relevant tables, and indicate such postprocessing in the captions of relevant plots.

## C.3 Disentanglement

In reporting the disentanglement metric results for baseline models, we use published results where applicable, i.e., we use the results published in [47] for VCT and COMET and the published results for SlowVAE [39]. For GVAE, MLVAE, and Ada-GVAE-K, we evaluate disentanglement using the Pytorch-based implementation of disentanglement metrics, [42], which corresponds to a Pytorch-based implementation of the official disentanglement lib of [25]. We also use this implementation to evaluate the disentanglement of representations produced by all models.

### C.3.1 Disentanglement Metrics

In line with [47], we consider the following 4 disentanglement metrics: the FactorVAE score [24], the DCI Disentanglement score [16] (we refer to DCI Disentanglement as DCI), the BetaVAE score [10], and the MIG score [15]. We provide a brief overview of all 4 disentanglement metrics, and

refer interested readers to the papers these metrics are introduced in for further details, as well as the Appendix of [25] for more details on how the metrics are implemented in [42] and [25].

**FactorVAE Metric**: A randomly selected FoV of the dataset is fixed, and a mini-batch of observations is subsequently randomly sampled. The representation learner produces representations for the samples. Disentanglement is quantified using the accuracy of a majority vote classifier that predicts the index of the ground-truth fixed FoV based on the index of the representation vector with the smallest variance.

**DCI Metric**: A Gradient Boosted Tree (GBT) is trained to predict the ground-truth FoV values from representations produced by a representation learner. The predictive importance of the dimensions of a representation is obtained using the model's feature importances. For each sample, a score is computed that corresponds to one minus the entropy of the probability that a dimension of the learned representation is useful for FoV prediction, weighted by the relative entropy of the corresponding dimension. An average of these scores over the mini-batch of samples is taken to produce the final score.

**BetaVAE Metric**: This metric quantifies disentanglement by predicting the index of a fixed FoV from representations produced by a representation learner, similar to the FactorVAE metric. In contrast to the FactorVAE metric, however, the BetaVAE metric uses a linear classifier on difference vectors to predict the index of the fixed FoV. Each difference vector is produced by taking the difference between representations produced for a pair of samples $(x, x')$ with one underlying fixed FoV.

**MIG Metric**: For each FoV, the MIG metric computes the mutual information between each dimension in the representation, and the corresponding FoV. The score is obtained by computing the average, normalised difference between the highest and second highest mutual information of each FoV with the dimensions of the representation.

### C.3.2 Evaluating the Disentanglement of Soft TPRs

Since disentanglement metrics are typically computed under the assumption that the representation corresponds to a concatenation of *scalar-valued* FoV tokens, we now detail how we compute the above disentanglement metrics on the Soft TPR, a *continuous* compositional representation. Similar amendments have been made in COMET and VCT, so that the disentanglement of their representations, corresponding to a concatenation of *vector-valued* FoV tokens, can be computed.

**FactorVAE metric**: To compute the FactorVAE score of our Soft TPR, we produce a $N_R$-dimensional vector, $v$, for each Soft TPR, where $N_R$ corresponds to the number of roles, i.e. the FoV *types*. We produce $v$ by simply populating each dimension, $v_i$, with the index of the quantised filler that role $r_i$ is bound to, i.e. we set $v_i$ to $m(i)$. We use the resulting $v$'s to compute the variances required by the FactorVAE metric, noting that if the FoV type corresponding to role $i$ is fixed, and this is reflected in our representation, dimension $i$ of the $v$ vectors produced in a mini-batch should clearly have the smallest variance, as all fillers, $f_{m(i)}$, that role $r_i$ binds to, will have the same identity across the mini-batch.

**DCI metric**: As the DCI relies on computing the ground-truth FoV values from $N_R$-dimensional representations produced by the representation learner, we again, follow a similar procedure as the above, in converting our $(D_F \cdot D_R)$-dimensional Soft TPR representation to a $N_R$-dimensional representation. That is, we simply consider a $N_R$-dimensional vector, $v$, where each dimension, $v_i$, is populated by the index of the quantised filler, $m(i)$ that role $r_i$ is bound to, and use this vector to compute the corresponding DCI result.

**BetaVAE metric**: For the BetaVAE metric, as each sample used to train the linear classifier consists of a $N_R$-dimensional difference vector obtained by computing the difference between the $N_R$-dimensional scalar-tokened compositional representations, we obtain the following difference vector, $d$, for our Soft TPR representations. For each role $i \in \{1, \ldots, N_R\}$, we obtain the corresponding quantised filler embeddings for each sample $x, x'$ in the pair, i.e., we obtain $\xi_F(f_{m(i)}), \xi_F(f_{m'(i)})$. For each pair of quantised filler embeddings, we obtain a scalar distance measure corresponding to the cosine similarity between the the pair. The $i$-th dimension of $d$ is populated using this value. We use difference vectors obtained in this way to compute the BetaVAE metric.

**MIG metric**: For the MIG metric, which relies on a discretisation of the values in each dimension $i$ of the $N_R$-dimensional scalar-tokened compositional representations to compute the discrete mutual

information, we apply the same postprocessing technique as in the FactorVAE, and DCI metric, to evaluate MIG on our Soft TPRs. That is, we produce the same $N_R$-dimensional vector, $v$, noting that this choice of postprocessing also performs the discretisation required by the MIG computation.

### C.3.3 Full Results

We now present unabridged results for all considered disentanglement metrics, denoting the learnable parameter control modification of each relevant baseline with the symbol $*$. As can be seen in Tables 16 and 17, our Soft TPR representations are explicitly more compositional (as quantified by the disentanglement metric scores) compared to all considered baselines, with especially notable performance increases on the more challenging datasets of Cars3D and MPI3D.

Table 16: FactorVAE and DCI scores

| Models | Cars3D | | Shapes3D | | MPI3D | |
|---|---|---|---|---|---|---|
| | FactorVAE score | DCI | FactorVAE score | DCI | FactorVAE score | DCI |
| Symbolic scalar-tokened compositional representations | | | | | | |
| Slow-VAE | $0.902 \pm 0.035$ | $0.509 \pm 0.027$ | $0.950 \pm 0.032$ | $0.850 \pm 0.047$ | $0.455 \pm 0.083$ | $0.355 \pm 0.027$ |
| Slow-VAE* | $0.872 \pm 0.038$ | $0.518 \pm 0.039$ | $0.961 \pm 0.028$ | $0.867 \pm 0.028$ | $0.421 \pm 0.091$ | $0.317 \pm 0.039$ |
| Ada-GVAE-k | $0.947 \pm 0.064$ | $0.664 \pm 0.167$ | $0.973 \pm 0.006$ | $\mathbf{0.963 \pm 0.077}$ | $0.496 \pm 0.095$ | $0.343 \pm 0.040$ |
| Ada-GVAE-k* | $0.931 \pm 0.051$ | $0.641 \pm 0.179$ | $0.932 \pm 0.012$ | $0.923 \pm 0.108$ | $0.451 \pm 0.129$ | $0.319 \pm 0.056$ |
| GVAE | $0.877 \pm 0.081$ | $0.262 \pm 0.095$ | $0.921 \pm 0.040$ | $0.842 \pm 0.040$ | $0.378 \pm 0.024$ | $0.245 \pm 0.074$ |
| GVAE* | $0.841 \pm 0.123$ | $0.219 \pm 0.012$ | $0.881 \pm 0.129$ | $0.810 \pm 0.102$ | $0.341 \pm 0.067$ | $0.216 \pm 0.109$ |
| ML-VAE | $0.870 \pm 0.052$ | $0.216 \pm 0.063$ | $0.835 \pm 0.111$ | $0.739 \pm 0.115$ | $0.390 \pm 0.026$ | $0.251 \pm 0.029$ |
| ML-VAE* | $0.881 \pm 0.041$ | $0.220 \pm 0.051$ | $0.823 \pm 0.091$ | $0.714 \pm 0.091$ | $0.401 \pm 0.019$ | $0.240 \pm 0.043$ |
| Shu | $0.573 \pm 0.062$ | $0.032 \pm 0.014$ | $0.265 \pm 0.043$ | $0.017 \pm 0.006$ | $0.287 \pm 0.034$ | $0.033 \pm 0.008$ |
| Shu* | $0.551 \pm 0.062$ | $0.019 \pm 0.021$ | $0.297 \pm 0.031$ | $0.020 \pm 0.006$ | $0.219 \pm 0.057$ | $0.029 \pm 0.010$ |
| Symbolic vector-tokened compositional representations | | | | | | |
| VCT | $0.966 \pm 0.029$ | $0.382 \pm 0.080$ | $0.957 \pm 0.043$ | $0.884 \pm 0.013$ | $0.689 \pm 0.035$ | $0.475 \pm 0.005$ |
| COMET | $0.339 \pm 0.008$ | $0.024 \pm 0.026$ | $0.168 \pm 0.005$ | $0.002 \pm 0.000$ | $0.145 \pm 0.024$ | $0.005 \pm 0.001$ |
| Fully continuous compositional representations | | | | | | |
| Ours | $\mathbf{0.999 \pm 0.001}$ | $\mathbf{0.863 \pm 0.027}$ | $\mathbf{0.984 \pm 0.012}$ | $0.926 \pm 0.028$ | $\mathbf{0.949 \pm 0.032}$ | $\mathbf{0.828 \pm 0.015}$ |

Table 17: BetaVAE and MIG scores

| Models | Cars3D | | Shapes3D | | MPI3D | |
|---|---|---|---|---|---|---|
| | BetaVAE score | MIG | BetaVAE score | MIG | BetaVAE score | MIG |
| Symbolic scalar-tokened compositional representations | | | | | | |
| Slow-VAE | $\mathbf{1.000 \pm 0.000}$ (=) | $0.104 \pm 0.018$ | $\mathbf{1.000 \pm 0.000}$ (=) | $0.615 \pm 0.045$ | $0.666 \pm 0.069$ | $0.329 \pm 0.026$ |
| Slow-VAE* | $\mathbf{1.000 \pm 0.000}$ (=) | $0.071 \pm 0.013$ | $\mathbf{1.000 \pm 0.000}$ (=) | $\mathbf{0.655 \pm 0.067}$ | $0.629 \pm 0.079$ | $0.287 \pm 0.045$ |
| Ada-GVAE-k | $\mathbf{1.000 \pm 0.000}$ (=) | $\mathbf{0.395 \pm 0.095}$ | $\mathbf{1.000 \pm 0.000}$ (=) | $0.556 \pm 0.064$ | $0.750 \pm 0.053$ | $0.213 \pm 0.064$ |
| Ada-GVAE-k* | $\mathbf{1.000 \pm 0.000}$ (=) | $0.321 \pm 0.102$ | $\mathbf{1.000 \pm 0.000}$ (=) | $0.498 \pm 0.073$ | $0.783 \pm 0.061$ | $0.241 \pm 0.079$ |
| GVAE | $\mathbf{1.000 \pm 0.000}$ (=) | $0.096 \pm 0.036$ | $0.998 \pm 0.004$ | $0.251 \pm 0.072$ | $0.704 \pm 0.072$ | $0.145 \pm 0.074$ |
| GVAE* | $\mathbf{1.000 \pm 0.000}$ (=) | $0.100 \pm 0.052$ | $\mathbf{1.000 \pm 0.000}$ | $0.203 \pm 0.089$ | $0.681 \pm 0.061$ | $0.109 \pm 0.087$ |
| MLVAE | $\mathbf{1.000 \pm 0.000}$ (=) | $0.088 \pm 0.020$ | $0.976 \pm 0.038$ | $0.354 \pm 0.165$ | $0.703 \pm 0.039$ | $0.142 \pm 0.062$ |
| MLVAE* | $\mathbf{1.000 \pm 0.000}$ (=) | $0.050 \pm 0.058$ | $0.921 \pm 0.052$ | $0.298 \pm 0.091$ | $0.689 \pm 0.041$ | $0.096 \pm 0.071$ |
| Shu | $0.912 \pm 0.022$ | $0.025 \pm 0.012$ | $0.498 \pm 0.064$ | $0.005 \pm 0.003$ | $0.353 \pm 0.040$ | $0.013 \pm 0.007$ |
| Shu* | $0.923 \pm 0.031$ | $0.015 \pm 0.009$ | $0.512 \pm 0.071$ | $0.006 \pm 0.002$ | $0.327 \pm 0.059$ | $0.009 \pm 0.011$ |
| Symbolic vector-tokened compositional representations | | | | | | |
| VCT | $\mathbf{1.000 \pm 0.000}$ (=) | $0.117 \pm 0.045$ | $0.999 \pm 0.0004$ | $0.525 \pm 0.028$ | $0.844 \pm 0.038$ | $0.227 \pm 0.048$ |
| COMET | $0.343 \pm 0.006$ | $0.000 \pm 0.000$ | $0.166 \pm 0.004$ | $0.0002 \pm 0.000$ | $0.144 \pm 0.005$ | $0.000 \pm 0.0001$ |
| Fully continuous compositional representations | | | | | | |
| Ours | $\mathbf{1.000 \pm 0.000}$ (=) | $0.348 \pm 0.0124$ | $\mathbf{1.000 \pm 0.000}$ (=) | $0.471 \pm 0.088$ | $\mathbf{1.000 \pm 0.000}$ | $\mathbf{0.620 \pm 0.067}$ |

### C.4 Representation Learning Convergence

We additionally examine representation learning convergence by evaluating the representations produced at 100, 1,000, 10,000, 100,000, and 200,000 iterations of model training, where the latter stage of 200,000 iterations corresponds to fully trained models. To quantify representation learning convergence, we evaluate both 1) the explicit compositionality of representations produced at these stages of training (as quantified by disentanglement metric performance), and 2) the usefulness of these representations for the downstream tasks of FoV regression and abstract visual reasoning.

In all line plots, we plot the mean, and indicate the standard deviation by the shaded regions. We use the same legend for all plots, where $0$ (grey) denotes SlowVAE, $1$ (orange) denotes AdaGVAE-k, $2$ (green) denotes GVAE, $3$ (red) denotes MLVAE, $4$ (purple) denotes Shu, $5$ (pink) denotes VCT, $6$ (brown) denotes COMET, and $7$ (blue) denotes our model, Soft TPR Autoencoder.

### C.4.1 Disentanglement

We first present line plots of representation learner convergence for each of the four considered disentanglement metrics (i.e., the FactorVAE, DCI, BetaVAE and MIG scores) for all three disentanglement datasets (Cars3D, Shapes3D, MPI3D). A series of tables containing the values associated with these line plots is presented following the plots. As the disentanglement results produced by our learnable parameter controls for the models of Ada-GVAE-k, GVAE, ML-VAE and Shu, do not achieve superior disentanglement results compared to the original models, we only present disentanglement convergence results for the original variants of all baseline models.

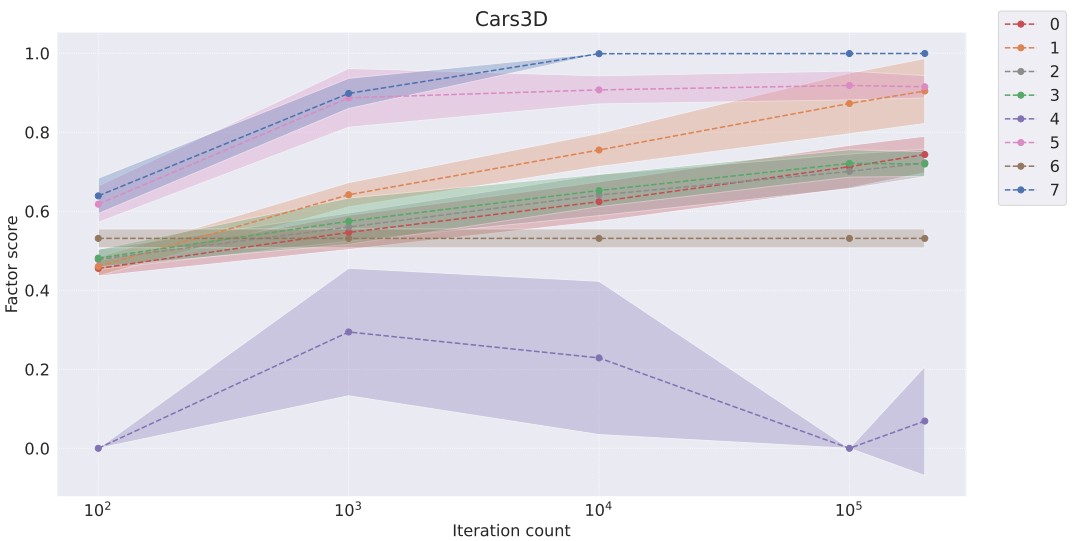

Figure 3: Factor score convergence on the Cars3D dataset

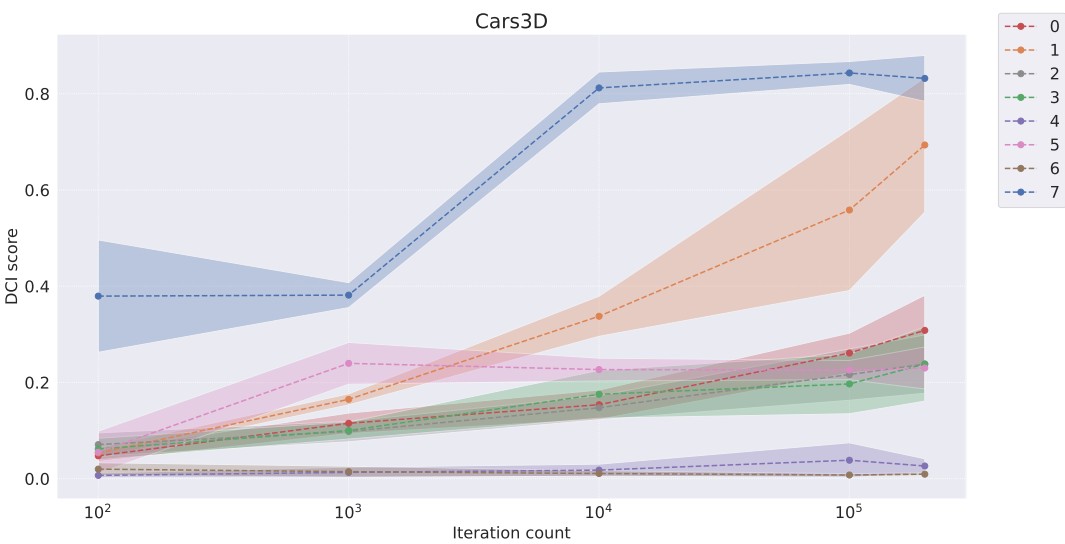

Figure 4: DCI score convergence on the Cars3D dataset

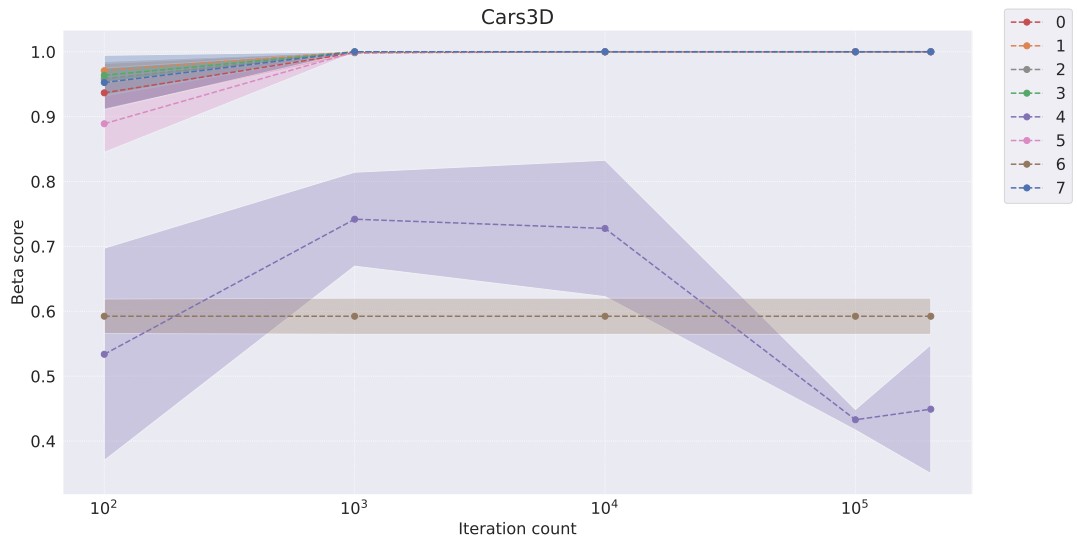

Figure 5: BetaVAE score convergence on the Cars3D dataset

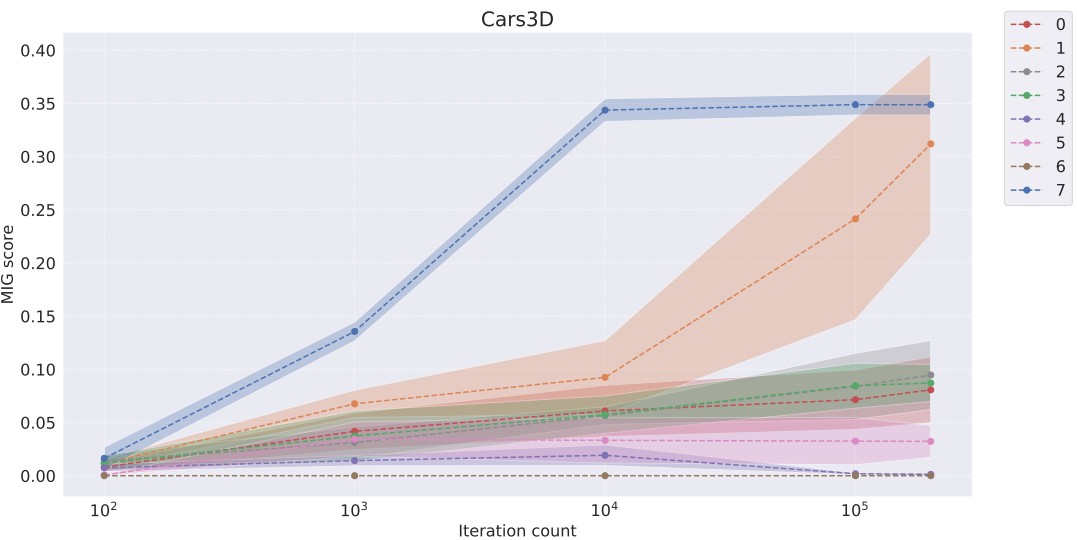

Figure 6: MIG score convergence on the Cars3D dataset

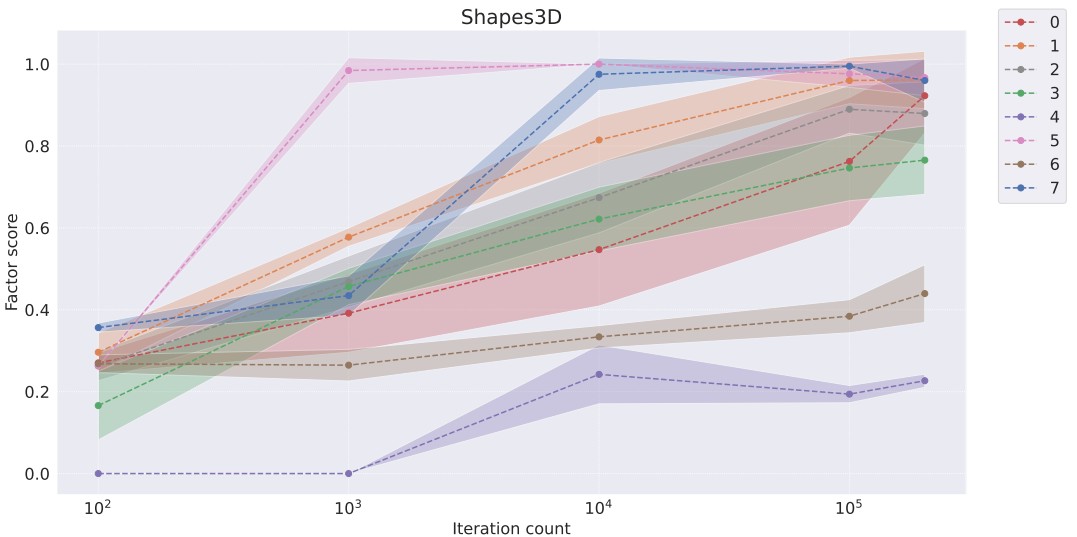

Figure 7: Factor score convergence on the Shapes3D dataset

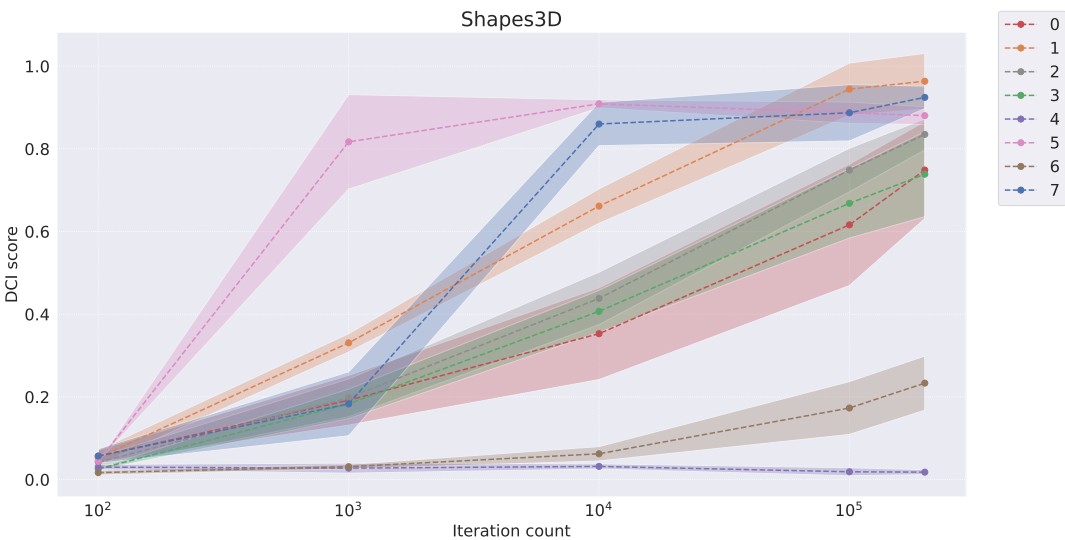

Figure 8: DCI score convergence on the Shapes3D dataset

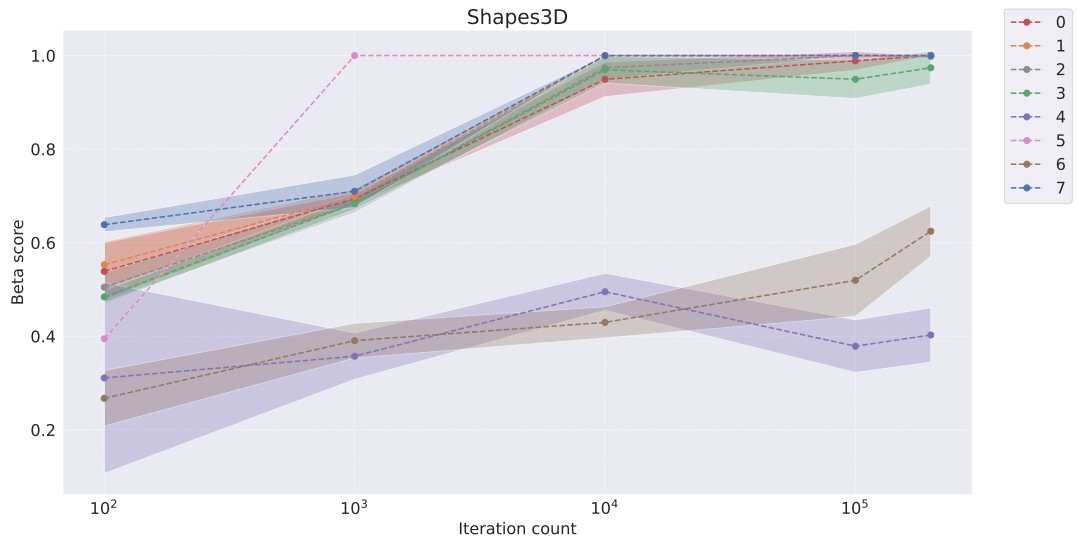

Figure 9: BetaVAE score convergence on the Shapes3D dataset

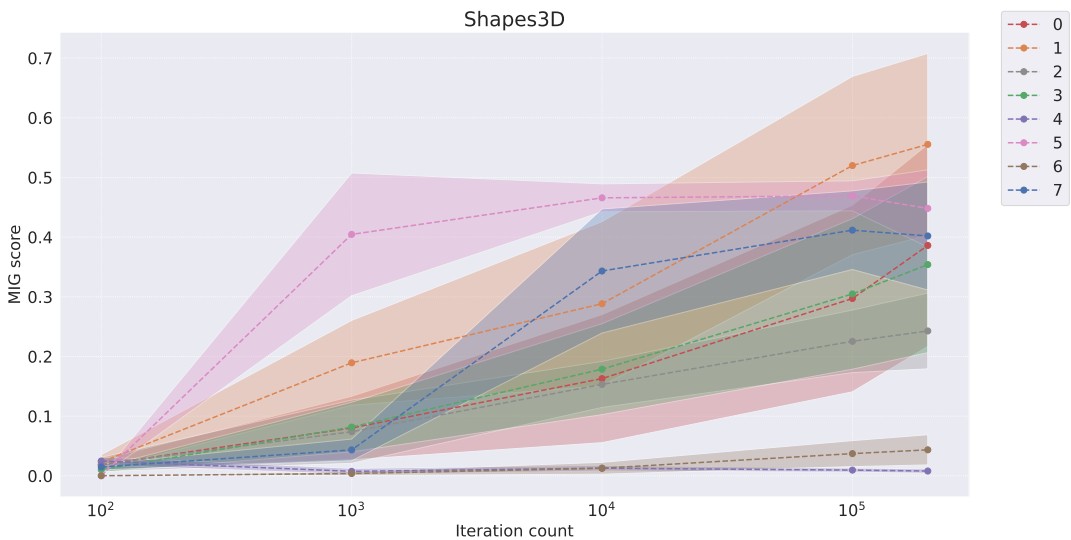

Figure 10: MIG score convergence on the Shapes3D dataset

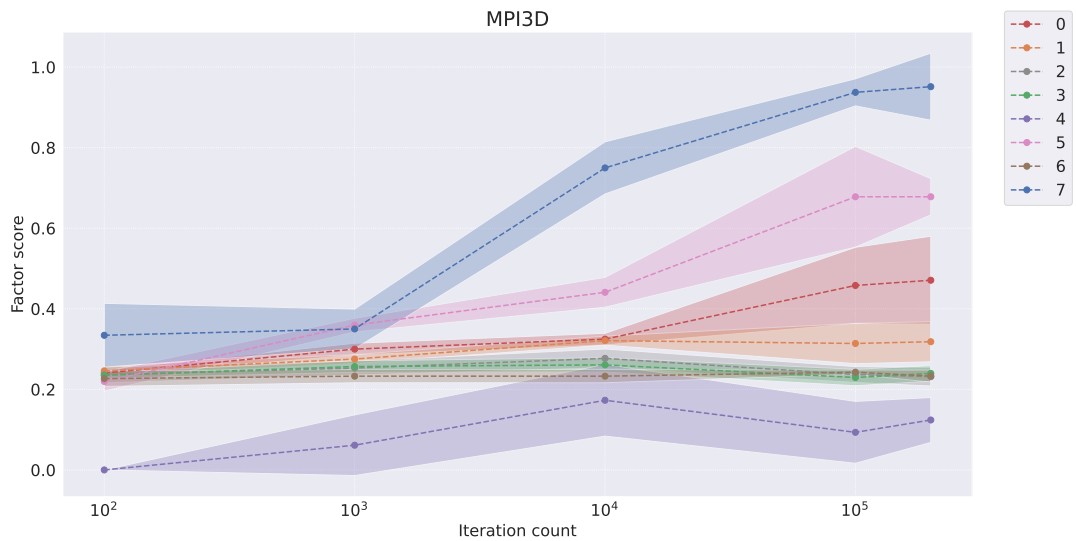

Figure 11: Factor score convergence on the MPI3D dataset

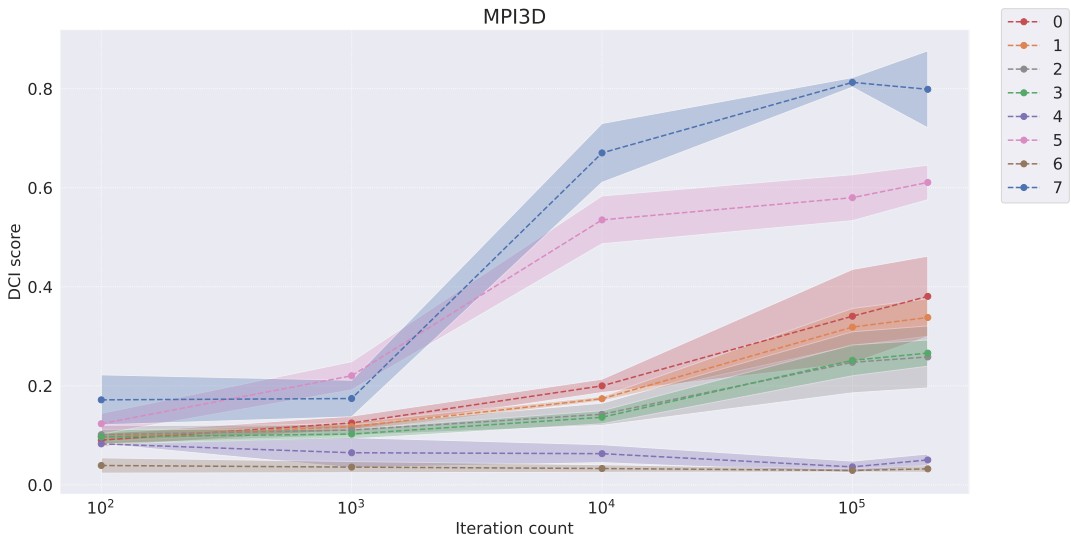

Figure 12: DCI score convergence on the MPI3D dataset

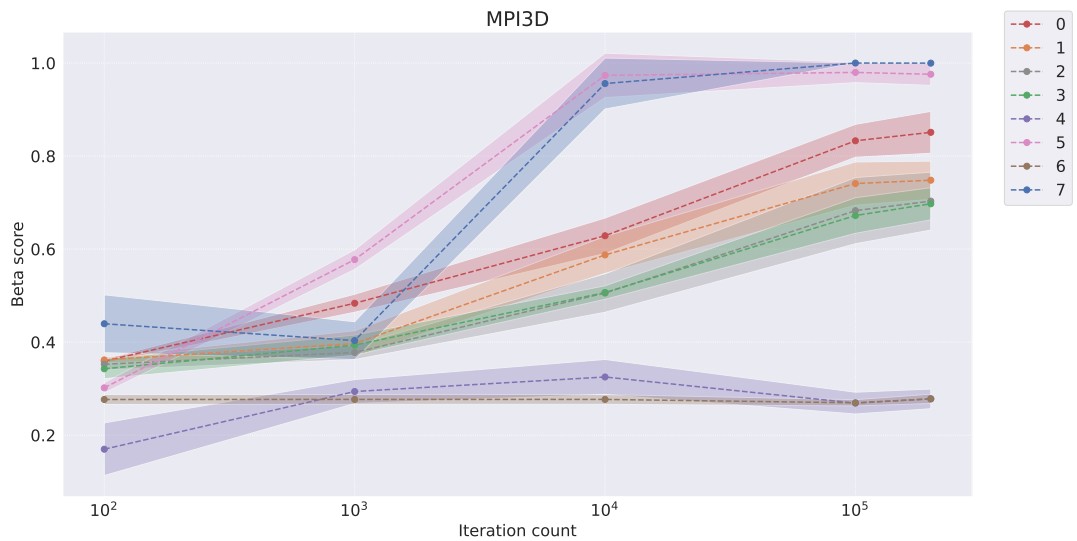

Figure 13: BetaVAE score convergence on the MPI3D dataset

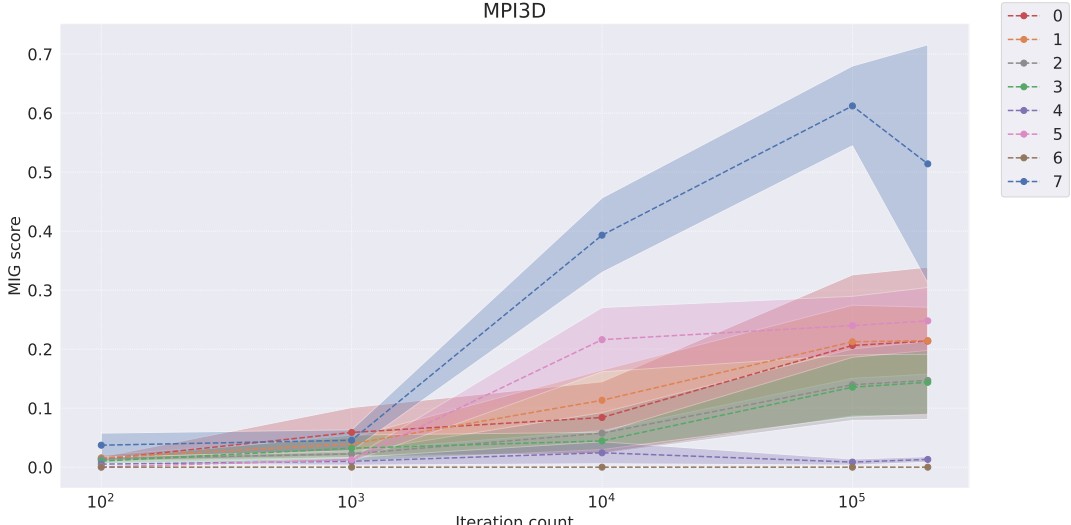

Figure 14: MIG score convergence on the MPI3D dataset

Table 18: Representation learner convergence on the Cars3D dataset (Factor score)

| Models | Factor score | | | | |
|---|---|---|---|---|---|
| | $10^2$ iterations | $10^3$ iterations | $10^4$ iterations | $10^5$ iterations | $2 \times 10^5$ iterations |
| Symbolic scalar-tokened compositional representations | | | | | |
| Slow-VAE | $0.455 \pm 0.020$ | $0.546 \pm 0.043$ | $0.624 \pm 0.050$ | $0.713 \pm 0.054$ | $0.743 \pm 0.046$ |
| Ada-GVAE-k | $0.460 \pm 0.024$ | $0.642 \pm 0.031$ | $0.755 \pm 0.042$ | $0.873 \pm 0.077$ | $0.904 \pm 0.082$ |
| GVAE | $0.478 \pm 0.026$ | $0.560 \pm 0.036$ | $0.641 \pm 0.052$ | $0.701 \pm 0.044$ | $0.722 \pm 0.035$ |
| MLVAE | $0.481 \pm 0.022$ | $0.575 \pm 0.059$ | $0.652 \pm 0.041$ | $0.721 \pm 0.035$ | $0.719 \pm 0.031$ |
| Shu | $0.000 \pm 0.000$ | $0.294 \pm 0.162$ | $0.229 \pm 0.194$ | $0.000 \pm 0.000$ | $0.069 \pm 0.138$ |
| Symbolic vector-tokened compositional representations | | | | | |
| VCT | *$0.618 \pm 0.046$* | $0.887 \pm 0.074$ | *$0.907 \pm 0.036$* | *$0.918 \pm 0.037$* | *$0.915 \pm 0.028$* |
| COMET | $0.531 \pm 0.024$ | $0.531 \pm 0.024$ | $0.531 \pm 0.024$ | $0.531 \pm 0.024$ | $0.531 \pm 0.024$ |
| Fully continuous compositional representations | | | | | |
| Ours | **$0.639 \pm 0.045$** | **$0.898 \pm 0.038$** | **$0.999 \pm 0.001$** | **$0.999 \pm 0.001$** | **$0.999 \pm 0.001$** |

Table 19: Representation learner convergence on the Cars3D dataset (DCI score)

| Models | DCI score | | | | |
|---|---|---|---|---|---|
| | $10^2$ iterations | $10^3$ iterations | $10^4$ iterations | $10^5$ iterations | $2 \times 10^5$ iterations |
| Symbolic scalar-tokened compositional representations | | | | | |
| Slow-VAE | $0.047 \pm 0.012$ | $0.115 \pm 0.021$ | $0.154 \pm 0.031$ | $0.261 \pm 0.041$ | $0.308 \pm 0.073$ |
| Ada-GVAE-k | $0.054 \pm 0.011$ | $0.165 \pm 0.012$ | *$0.338 \pm 0.042$* | *$0.558 \pm 0.167$* | *$0.694 \pm 0.140$* |
| GVAE | $0.071 \pm 0.024$ | $0.098 \pm 0.021$ | $0.147 \pm 0.025$ | $0.216 \pm 0.053$ | $0.238 \pm 0.061$ |
| MLVAE | *$0.062 \pm 0.022$* | $0.100 \pm 0.018$ | $0.175 \pm 0.050$ | $0.197 \pm 0.062$ | $0.238 \pm 0.077$ |
| Shu | $0.007 \pm 0.005$ | $0.013 \pm 0.011$ | $0.018 \pm 0.012$ | $0.038 \pm 0.036$ | $0.026 \pm 0.015$ |
| Symbolic vector-tokened compositional representations | | | | | |
| VCT | $0.055 \pm 0.044$ | *$0.240 \pm 0.043$* | $0.227 \pm 0.024$ | $0.226 \pm 0.020$ | $0.230 \pm 0.044$ |
| COMET | $0.020 \pm 0.013$ | $0.015 \pm 0.011$ | $0.011 \pm 0.006$ | $0.008 \pm 0.003$ | $0.009 \pm 0.003$ |
| Fully continuous compositional representations | | | | | |
| Ours | **$0.379 \pm 0.117$** | **$0.381 \pm 0.026$** | **$0.812 \pm 0.033$** | **$0.843 \pm 0.024$** | **$0.832 \pm 0.048$** |

Table 20: Representation learner convergence on the Cars3D dataset (BetaVAE score)

| Models | BetaVAE score | | | | |
|---|---|---|---|---|---|
| | $10^2$ iterations | $10^3$ iterations | $10^4$ iterations | $10^5$ iterations | $2 \times 10^5$ iterations |
| Symbolic scalar-tokened compositional representations | | | | | |
| Slow-VAE | $0.937 \pm 0.025$ | $0.999 \pm 0.002$ (=) | **$1.000 \pm 0.000$** (=) | **$1.000 \pm 0.000$** (=) | **$1.000 \pm 0.000$** (=) |
| Ada-GVAE-k | **$0.971 \pm 0.011$** | **$1.000 \pm 0.000$** (=) | **$1.000 \pm 0.000$** (=) | **$1.000 \pm 0.000$** (=) | **$1.000 \pm 0.000$** (=) |
| GVAE | *$0.958 \pm 0.027$* | *$0.999 \pm 0.001$* (=) | **$1.000 \pm 0.000$** (=) | **$1.000 \pm 0.000$** (=) | **$1.000 \pm 0.000$** (=) |
| MLVAE | $0.964 \pm 0.011$ | **$1.000 \pm 0.000$** (=) | **$1.000 \pm 0.000$** (=) | **$1.000 \pm 0.000$** (=) | **$1.000 \pm 0.000$** (=) |
| Shu | $0.534 \pm 0.164$ | $0.742 \pm 0.072$ | *$0.728 \pm 0.105$* | $0.433 \pm 0.016$ | $0.449 \pm 0.099$ |
| Symbolic vector-tokened compositional representations | | | | | |
| VCT | $0.889 \pm 0.044$ | **$1.000 \pm 0.001$** (=) | **$1.000 \pm 0.000$** (=) | **$1.000 \pm 0.000$** (=) | **$1.000 \pm 0.000$** (=) |
| COMET | $0.592 \pm 0.027$ | $0.592 \pm 0.028$ | $0.592 \pm 0.028$ | *$0.592 \pm 0.028$* | *$0.592 \pm 0.028$* |
| Fully continuous compositional representations | | | | | |
| Ours | $0.953 \pm 0.042$ | **$1.000 \pm 0.000$** (=) | **$1.000 \pm 0.000$** (=) | **$1.000 \pm 0.000$** (=) | **$1.000 \pm 0.000$** |

Table 21: Representation learner convergence on the Cars3D dataset (MIG score)

| Models | MIG score | | | | |
|---|---|---|---|---|---|
| | $10^2$ iterations | $10^3$ iterations | $10^4$ iterations | $10^5$ iterations | $2 \times 10^5$ iterations |
| Symbolic scalar-tokened compositional representations | | | | | |
| Slow-VAE | $0.008 \pm 0.004$ | $0.042 \pm 0.017$ | $0.061 \pm 0.024$ | $0.071 \pm 0.028$ | $0.081 \pm 0.031$ |
| Ada-GVAE-k | *0.011 ± 0.005* | *0.068 ± 0.013* | *0.092 ± 0.035* | *0.241 ± 0.095* | *0.312 ± 0.085* |
| GVAE | $0.014 \pm 0.006$ | $0.031 \pm 0.015$ | $0.056 \pm 0.008$ | $0.084 \pm 0.031$ | $0.095 \pm 0.032$ |
| MLVAE | $0.012 \pm 0.007$ | $0.037 \pm 0.024$ | $0.057 \pm 0.018$ | $0.085 \pm 0.021$ | $0.087 \pm 0.017$ |
| Shu | $0.007 \pm 0.004$ | $0.014 \pm 0.005$ | $0.019 \pm 0.010$ | $0.002 \pm 0.001$ | $0.001 \pm 0.001$ |
| Symbolic vector-tokened compositional representations | | | | | |
| VCT | $0.001 \pm 0.001$ | $0.034 \pm 0.016$ | $0.033 \pm 0.021$ | $0.033 \pm 0.022$ | $0.032 \pm 0.015$ |
| COMET | $0.000 \pm 0.000$ | $0.000 \pm 0.000$ | $0.000 \pm 0.000$ | $0.000 \pm 0.000$ | $0.000 \pm 0.000$ |
| Fully continuous compositional representations | | | | | |
| Ours | **0.016 ± 0.010** | **0.136 ± 0.009** | **0.344 ± 0.011** | **0.349 ± 0.010** | **0.349 ± 0.010** |

Table 22: Representation learner convergence on the Shapes3D dataset (Factor score)

| Models | Factor score | | | | |
|---|---|---|---|---|---|
| | $10^2$ iterations | $10^3$ iterations | $10^4$ iterations | $10^5$ iterations | $2 \times 10^5$ iterations |
| Symbolic scalar-tokened compositional representations | | | | | |
| Slow-VAE | $0.270 \pm 0.025$ | $0.392 \pm 0.095$ | $0.547 \pm 0.138$ | $0.762 \pm 0.156$ | $0.923 \pm 0.091$ |
| Ada-GVAE-k | *0.296 ± 0.047* | *0.577 ± 0.023* | *0.815 ± 0.057* | $0.960 \pm 0.057$ | *0.961 ± 0.070* |
| GVAE | $0.263 \pm 0.036$ | $0.469 \pm 0.063$ | $0.674 \pm 0.086$ | $0.890 \pm 0.058$ | $0.879 \pm 0.077$ |
| MLVAE | $0.166 \pm 0.084$ | $0.457 \pm 0.045$ | $0.621 \pm 0.078$ | $0.746 \pm 0.080$ | $0.765 \pm 0.084$ |
| Shu | $0.000 \pm 0.000$ | $0.000 \pm 0.000$ | $0.242 \pm 0.072$ | $0.194 \pm 0.021$ | $0.227 \pm 0.016$ |
| Symbolic vector-tokened compositional representations | | | | | |
| VCT | $0.263 \pm 0.000$ | **0.984 ± 0.031** | **1.000 ± 0.000** | *0.976 ± 0.033* | **0.967 ± 0.044** |
| COMET | $0.268 \pm 0.022$ | $0.265 \pm 0.039$ | $0.334 \pm 0.028$ | $0.384 \pm 0.040$ | $0.440 \pm 0.071$ |
| Fully continuous compositional representations | | | | | |
| Ours | **0.356 ± 0.011** | $0.435 \pm 0.048$ | *0.975 ± 0.040* | **0.995 ± 0.005** | $0.960 \pm 0.053$ |

Table 23: Representation learner convergence on the Shapes3D dataset (DCI score)

| Models | DCI score | | | | |
|---|---|---|---|---|---|
| | $10^2$ iterations | $10^3$ iterations | $10^4$ iterations | $10^5$ iterations | $2 \times 10^5$ iterations |
| Symbolic scalar-tokened compositional representations | | | | | |
| Slow-VAE | *0.056 ± 0.015* | $0.192 \pm 0.060$ | $0.353 \pm 0.111$ | $0.616 \pm 0.147$ | $0.749 \pm 0.118$ |
| Ada-GVAE-k | $0.053 \pm 0.018$ | *0.331 ± 0.022* | $0.662 \pm 0.042$ | **0.944 ± 0.063** | **0.964 ± 0.067** |
| GVAE | $0.042 \pm 0.013$ | $0.198 \pm 0.046$ | $0.438 \pm 0.064$ | $0.749 \pm 0.053$ | $0.835 \pm 0.038$ |
| MLVAE | $0.026 \pm 0.003$ | $0.183 \pm 0.036$ | $0.407 \pm 0.051$ | $0.668 \pm 0.084$ | $0.739 \pm 0.103$ |
| Shu | $0.030 \pm 0.007$ | $0.028 \pm 0.012$ | $0.032 \pm 0.006$ | $0.019 \pm 0.010$ | $0.018 \pm 0.006$ |
| Symbolic vector-tokened compositional representations | | | | | |
| VCT | $0.044 \pm 0.000$ | **0.817 ± 0.114** | **0.909 ± 0.010** | *0.887 ± 0.026* | $0.880 \pm 0.022$ |
| COMET | $0.017 \pm 0.006$ | $0.031 \pm 0.005$ | $0.062 \pm 0.017$ | $0.173 \pm 0.064$ | $0.233 \pm 0.066$ |
| Fully continuous compositional representations | | | | | |
| Ours | **0.057 ± 0.018** | $0.183 \pm 0.077$ | *0.860 ± 0.052* | $0.887 \pm 0.068$ | *0.924 ± 0.027* |

Table 24: Representation learner convergence on the Shapes3D dataset (BetaVAE score)

| Models | BetaVAE score | | | | |
|---|---|---|---|---|---|
| | $10^2$ iterations | $10^3$ iterations | $10^4$ iterations | $10^5$ iterations | $2 \times 10^5$ iterations |
| Symbolic scalar-tokened compositional representations | | | | | |
| Slow-VAE | $0.539 \pm 0.061$ | $0.694 \pm 0.012$ | $0.949 \pm 0.037$ | $0.989 \pm 0.020$ | $\mathbf{1.000 \pm 0.000}$ (=) |
| Ada-GVAE-k | $0.553 \pm 0.050$ | $0.699 \pm 0.016$ | $0.999 \pm 0.001$ | $\mathbf{1.000 \pm 0.000}$ (=) | $\mathbf{1.000 \pm 0.000}$ (=) |
| GVAE | $0.505 \pm 0.031$ | $0.683 \pm 0.019$ | $0.974 \pm 0.025$ | $\mathbf{1.000 \pm 0.000}$ (=) | $0.998 \pm 0.004$ |
| MLVAE | $0.484 \pm 0.015$ | $0.684 \pm 0.013$ | $0.970 \pm 0.029$ | $0.949 \pm 0.041$ | $0.974 \pm 0.036$ |
| Shu | $0.311 \pm 0.204$ | $0.357 \pm 0.050$ | $0.495 \pm 0.040$ | $0.379 \pm 0.056$ | $0.403 \pm 0.058$ |
| Symbolic vector-tokened compositional representations | | | | | |
| VCT | $\mathbf{0.889 \pm 0.044}$ | $\mathbf{1.000 \pm 0.000}$ (=) | $\mathbf{1.000 \pm 0.000}$ (=) | $\mathbf{1.000 \pm 0.000}$ (=) | $\mathbf{1.000 \pm 0.000}$ (=) |
| COMET | $0.395 \pm 0.000$ | $\mathbf{1.000 \pm 0.000}$ (=) | $\mathbf{1.000 \pm 0.000}$ (=) | $\mathbf{1.000 \pm 0.000}$ (=) | $\mathbf{1.000 \pm 0.000}$ (=) |
| Fully continuous compositional representations | | | | | |
| Ours | $0.638 \pm 0.015$ | $0.710 \pm 0.035$ | $\mathbf{1.000 \pm 0.000}$ (=) | $\mathbf{1.000 \pm 0.000}$ (=) | $\mathbf{1.000 \pm 0.000}$ (=) |

Table 25: Representation learner convergence on the Shapes3D dataset (MIG score)

| Models | MIG score | | | | |
|---|---|---|---|---|---|
| | $10^2$ iterations | $10^3$ iterations | $10^4$ iterations | $10^5$ iterations | $2 \times 10^5$ iterations |
| Symbolic scalar-tokened compositional representations | | | | | |
| Slow-VAE | $0.018 \pm 0.009$ | $0.080 \pm 0.053$ | $0.163 \pm 0.107$ | $0.297 \pm 0.156$ | $0.386 \pm 0.169$ |
| Ada-GVAE-k | $0.023 \pm 0.012$ | $0.189 \pm 0.071$ | $0.288 \pm 0.138$ | $\mathbf{0.520 \pm 0.149}$ | $\mathbf{0.555 \pm 0.152}$ |
| GVAE | $0.019 \pm 0.012$ | $0.074 \pm 0.053$ | $0.153 \pm 0.039$ | $0.225 \pm 0.052$ | $0.243 \pm 0.064$ |
| MLVAE | $0.010 \pm 0.004$ | $0.082 \pm 0.042$ | $0.179 \pm 0.076$ | $0.305 \pm 0.126$ | $0.354 \pm 0.148$ |
| Shu | $\mathbf{0.025 \pm 0.007}$ | $0.008 \pm 0.005$ | $0.013 \pm 0.005$ | $0.010 \pm 0.003$ | $0.008 \pm 0.004$ |
| Symbolic vector-tokened compositional representations | | | | | |
| VCT | $0.000 \pm 0.000$ | $\mathbf{0.405 \pm 0.103}$ | $\mathbf{0.466 \pm 0.023}$ | $0.469 \pm 0.025$ | $0.448 \pm 0.065$ |
| COMET | $0.000 \pm 0.000$ | $0.004 \pm 0.003$ | $0.013 \pm 0.010$ | $0.037 \pm 0.022$ | $0.044 \pm 0.025$ |
| Fully continuous compositional representations | | | | | |
| Ours | $0.015 \pm 0.005$ | $0.043 \pm 0.018$ | $0.343 \pm 0.104$ | $0.412 \pm 0.066$ | $0.402 \pm 0.091$ |

Table 26: Representation learner convergence on the MPI3D dataset (Factor score)

| Models | Factor score | | | | |
|---|---|---|---|---|---|
| | $10^2$ iterations | $10^3$ iterations | $10^4$ iterations | $10^5$ iterations | $2 \times 10^5$ iterations |
| Symbolic scalar-tokened compositional representations | | | | | |
| Slow-VAE | $0.240 \pm 0.008$ | $0.300 \pm 0.015$ | $0.325 \pm 0.014$ | $0.458 \pm 0.095$ | $0.471 \pm 0.109$ |
| Ada-GVAE-k | $0.246 \pm 0.012$ | $0.275 \pm 0.011$ | $0.321 \pm 0.011$ | $0.314 \pm 0.050$ | $0.318 \pm 0.050$ |
| GVAE | $0.236 \pm 0.020$ | $0.253 \pm 0.017$ | $0.277 \pm 0.023$ | $0.237 \pm 0.020$ | $0.232 \pm 0.023$ |
| MLVAE | $0.235 \pm 0.008$ | $0.257 \pm 0.014$ | $0.261 \pm 0.018$ | $0.229 \pm 0.021$ | $0.240 \pm 0.019$ |
| Shu | $0.000 \pm 0.000$ | $0.061 \pm 0.075$ | $0.173 \pm 0.089$ | $0.093 \pm 0.077$ | $0.124 \pm 0.056$ |
| Symbolic vector-tokened compositional representations | | | | | |
| VCT | $0.220 \pm 0.024$ | $\mathbf{0.360 \pm 0.017}$ | $0.441 \pm 0.037$ | $0.678 \pm 0.125$ | $0.678 \pm 0.045$ |
| COMET | $0.227 \pm 0.019$ | $0.233 \pm 0.016$ | $0.233 \pm 0.016$ | $0.243 \pm 0.005$ | $0.233 \pm 0.015$ |
| Fully continuous compositional representations | | | | | |
| Ours | $\mathbf{0.334 \pm 0.079}$ | $0.350 \pm 0.049$ | $\mathbf{0.750 \pm 0.065}$ | $\mathbf{0.937 \pm 0.034}$ | $\mathbf{0.951 \pm 0.083}$ |

Table 27: Representation learner convergence on the MPI3D dataset (DCI score)

| Models | DCI score | | | | |
|---|---|---|---|---|---|
| | $10^2$ iterations | $10^3$ iterations | $10^4$ iterations | $10^5$ iterations | $2 \times 10^5$ iterations |
| Symbolic scalar-tokened compositional representations | | | | | |
| Slow-VAE | $0.090 \pm 0.013$ | $0.125 \pm 0.014$ | $0.200 \pm 0.014$ | $0.341 \pm 0.095$ | $0.381 \pm 0.081$ |
| Ada-GVAE-k | $0.097 \pm 0.008$ | $0.117 \pm 0.009$ | $0.174 \pm 0.005$ | $0.319 \pm 0.038$ | $0.338 \pm 0.038$ |
| GVAE | $0.101 \pm 0.019$ | $0.111 \pm 0.011$ | $0.142 \pm 0.022$ | $0.247 \pm 0.062$ | $0.258 \pm 0.063$ |
| MLVAE | $0.097 \pm 0.015$ | $0.103 \pm 0.010$ | $0.137 \pm 0.013$ | $0.252 \pm 0.031$ | $0.266 \pm 0.026$ |
| Shu | $0.083 \pm 0.000$ | $0.065 \pm 0.030$ | $0.063 \pm 0.018$ | $0.036 \pm 0.012$ | $0.050 \pm 0.012$ |
| Symbolic vector-tokened compositional representations | | | | | |
| VCT | $0.123 \pm 0.021$ | $\mathbf{0.220 \pm 0.029}$ | $0.535 \pm 0.048$ | $0.580 \pm 0.046$ | $0.611 \pm 0.035$ |
| COMET | $0.039 \pm 0.016$ | $0.036 \pm 0.011$ | $0.033 \pm 0.008$ | $0.029 \pm 0.003$ | $0.032 \pm 0.007$ |
| Fully continuous compositional representations | | | | | |
| Ours | $\mathbf{0.172 \pm 0.051}$ | $0.174 \pm 0.037$ | $\mathbf{0.670 \pm 0.060}$ | $\mathbf{0.813 \pm 0.010}$ | $\mathbf{0.799 \pm 0.078}$ |

Table 28: Representation learner convergence on the MPI3D dataset (BetaVAE score)

| Models | BetaVAE score | | | | |
|---|---|---|---|---|---|
| | $10^2$ iterations | $10^3$ iterations | $10^4$ iterations | $10^5$ iterations | $2 \times 10^5$ iterations |
| Symbolic scalar-tokened compositional representations | | | | | |
| Slow-VAE | $0.359 \pm 0.004$ | $0.484 \pm 0.019$ | $0.629 \pm 0.038$ | $0.833 \pm 0.035$ | $0.851 \pm 0.045$ |
| Ada-GVAE-k | $0.362 \pm 0.006$ | $0.397 \pm 0.028$ | $0.587 \pm 0.040$ | $0.741 \pm 0.047$ | $0.748 \pm 0.041$ |
| GVAE | $0.352 \pm 0.012$ | $0.378 \pm 0.015$ | $0.506 \pm 0.041$ | $0.683 \pm 0.071$ | $0.703 \pm 0.062$ |
| MLVAE | $0.343 \pm 0.021$ | $0.394 \pm 0.022$ | $0.507 \pm 0.015$ | $0.672 \pm 0.038$ | $0.697 \pm 0.035$ |
| Shu | $0.170 \pm 0.057$ | $0.294 \pm 0.026$ | $0.325 \pm 0.038$ | $0.269 \pm 0.023$ | $0.278 \pm 0.021$ |
| Symbolic vector-tokened compositional representations | | | | | |
| VCT | $\mathbf{0.889 \pm 0.044}$ | $\mathbf{1.000 \pm 0.001}$ | $\mathbf{1.000 \pm 0.000}$ | $\mathbf{1.000 \pm 0.000}$ | $\mathbf{1.000 \pm 0.000}$ |
| COMET | $0.277 \pm 0.011$ | $0.277 \pm 0.011$ | $0.277 \pm 0.011$ | $0.270 \pm 0.007$ | $0.278 \pm 0.010$ |
| Fully continuous compositional representations | | | | | |
| Ours | $0.302 \pm 0.011$ | $0.577 \pm 0.021$ | $0.973 \pm 0.048$ | $0.980 \pm 0.022$ | $0.976 \pm 0.024$ |

Table 29: Representation learner convergence on the MPI3D dataset (MIG score)

| Models | MIG score | | | | |
|---|---|---|---|---|---|
| | $10^2$ iterations | $10^3$ iterations | $10^4$ iterations | $10^5$ iterations | $2 \times 10^5$ iterations |
| Symbolic scalar-tokened compositional representations | | | | | |
| Slow-VAE | $0.014 \pm 0.004$ | $\mathbf{0.059 \pm 0.042}$ | $0.084 \pm 0.061$ | $0.206 \pm 0.120$ | $0.214 \pm 0.125$ |
| Ada-GVAE-k | $0.016 \pm 0.005$ | $0.039 \pm 0.011$ | $0.113 \pm 0.051$ | $0.213 \pm 0.062$ | $0.214 \pm 0.057$ |
| GVAE | $0.014 \pm 0.004$ | $0.022 \pm 0.003$ | $0.058 \pm 0.035$ | $0.140 \pm 0.061$ | $0.147 \pm 0.066$ |
| MLVAE | $0.010 \pm 0.004$ | $0.032 \pm 0.022$ | $0.044 \pm 0.015$ | $0.136 \pm 0.051$ | $0.144 \pm 0.054$ |
| Shu | $0.005 \pm 0.000$ | $0.010 \pm 0.006$ | $0.024 \pm 0.019$ | $0.009 \pm 0.005$ | $0.013 \pm 0.005$ |
| Symbolic vector-tokened compositional representations | | | | | |
| VCT | $0.000 \pm 0.000$ | $0.013 \pm 0.003$ | $0.216 \pm 0.055$ | $0.240 \pm 0.050$ | $0.248 \pm 0.057$ |
| COMET | $0.000 \pm 0.000$ | $0.000 \pm 0.000$ | $0.000 \pm 0.000$ | $0.000 \pm 0.000$ | $0.000 \pm 0.000$ |
| Fully continuous compositional representations | | | | | |
| Ours | $\mathbf{0.037 \pm 0.020}$ | $0.046 \pm 0.018$ | $\mathbf{0.393 \pm 0.063}$ | $\mathbf{0.612 \pm 0.068}$ | $\mathbf{0.514 \pm 0.202}$ |

### C.4.2 Downstream Performance

We additionally evaluate the representation learning convergence by examining the usefulness of representations produced at different stages of training (i.e., 100, 250, 500, 1,000, 10,000, 100,000

and 200,000 iterations of training). To quantify 'usefulness', we consider performance of downstream models on the tasks of FoV regression, and abstract visual reasoning when trained using representations produced by each stage of training. For each task, we present line plots, and additionally tables with values corresponding to each of the plots. We use the same legend as in the previous section, where 0 (grey) denotes SlowVAE, 1 (orange) denotes AdaGVAE-k, 2 (green) denotes GVAE, 3 (red) denotes MLVAE, 4 (purple) denotes Shu, 5 (pink) denotes VCT, 6 (brown) denotes COMET, and 7 (blue) denotes our model, Soft TPR Autoencoder. We additionally provide all results for the dimensionality-control setting, where the dimensionality of representations produced by all representation learners is held constant following the approach detailed in C.2.4, denoting this clearly in plot captions, and by the symbol $^{\dagger}$ in the tables.

**FoV Regression**

As clearly visible in the tables and plots, generic regression models are able to more effectively use representations produced by our Soft TPR Autoencoder produced across almost all stages of training for all disentanglement datasets. Improvements are most notable in the low-iteration regime of $10^2$ iterations, and across most stages of training for the more challenging task of FoV regression on the MPI3D dataset (Figures 19 and 20).

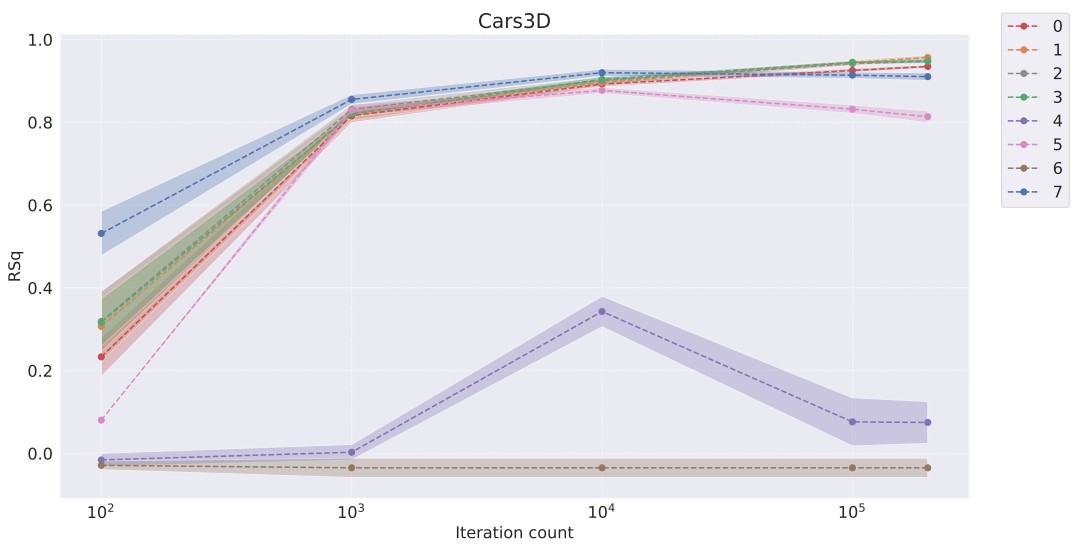

Figure 15: Convergence of representation learners as measured by FoV regression on the Cars3D dataset (original setting)

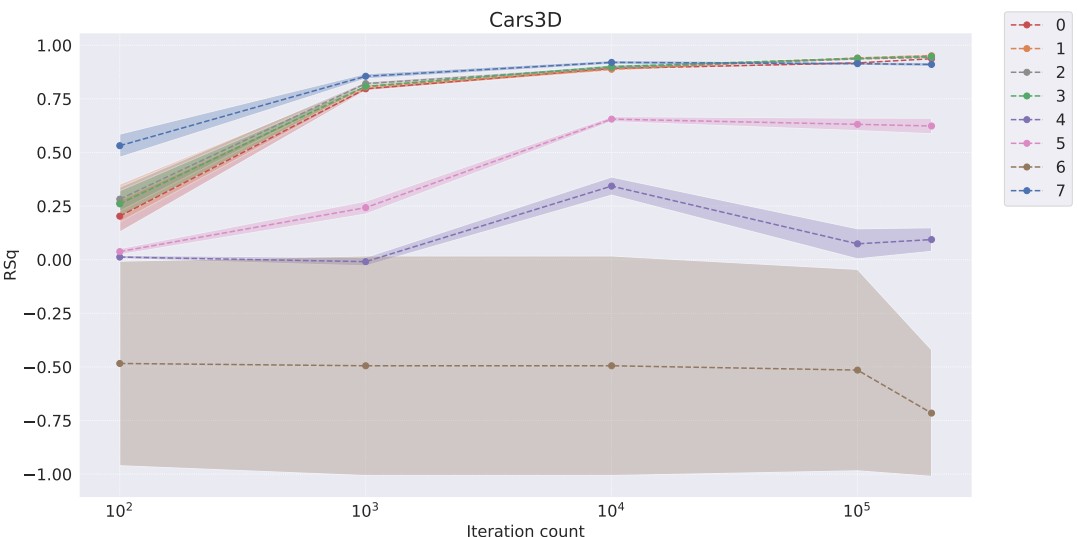

Figure 16: Convergence of representation learners as measured by FoV regression on the Cars3D dataset (dimensionality-controlled setting)

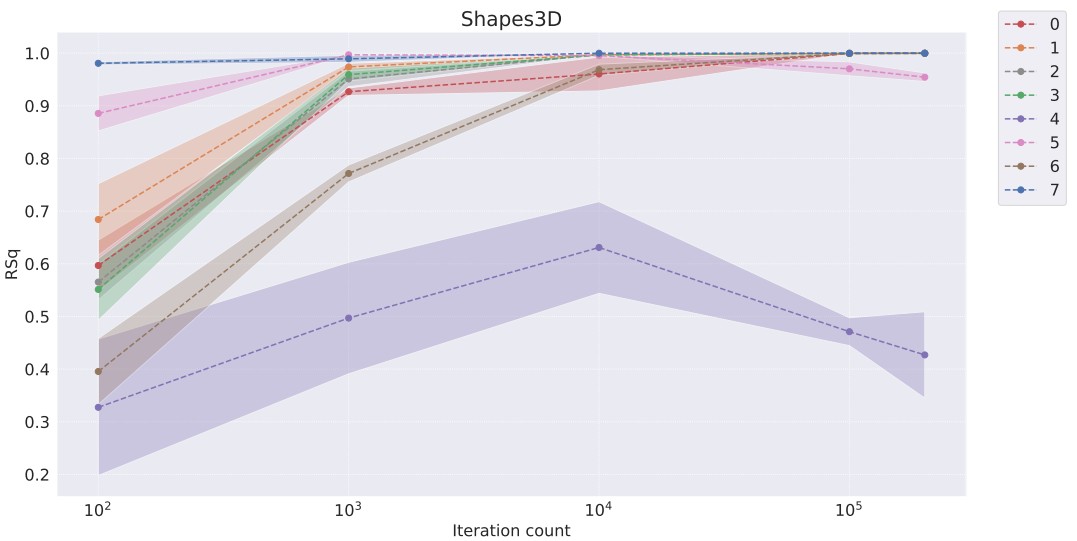

Figure 17: Convergence of representation learners as measured by FoV regression on the Shapes3D dataset (original setting)

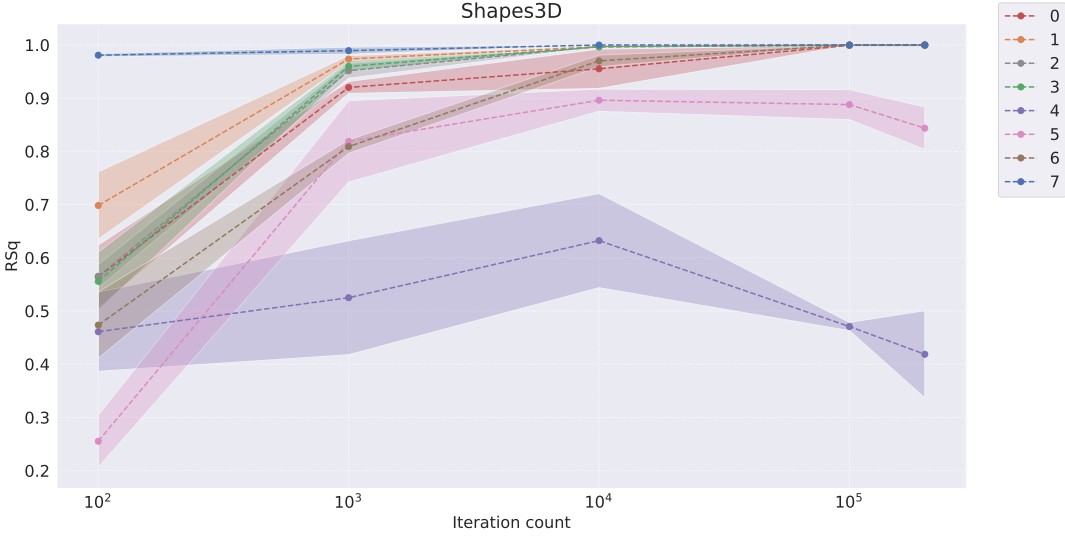

Figure 18: Convergence of representation learners as measured by FoV regression on the Shapes3D dataset (dimensionality-controlled setting)

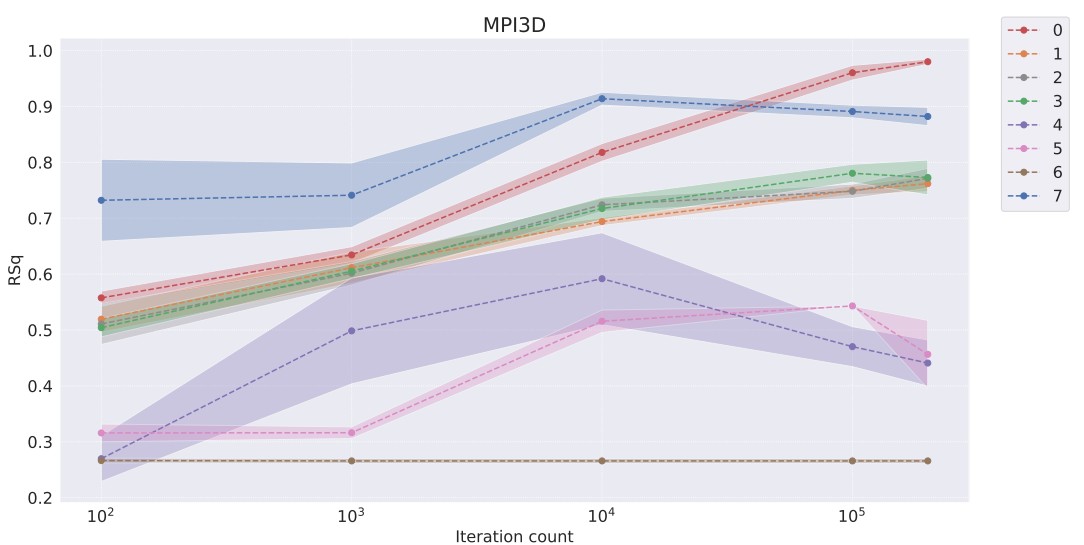

Figure 19: Convergence of representation learners as measured by FoV regression on the MPI3D dataset (original setting)

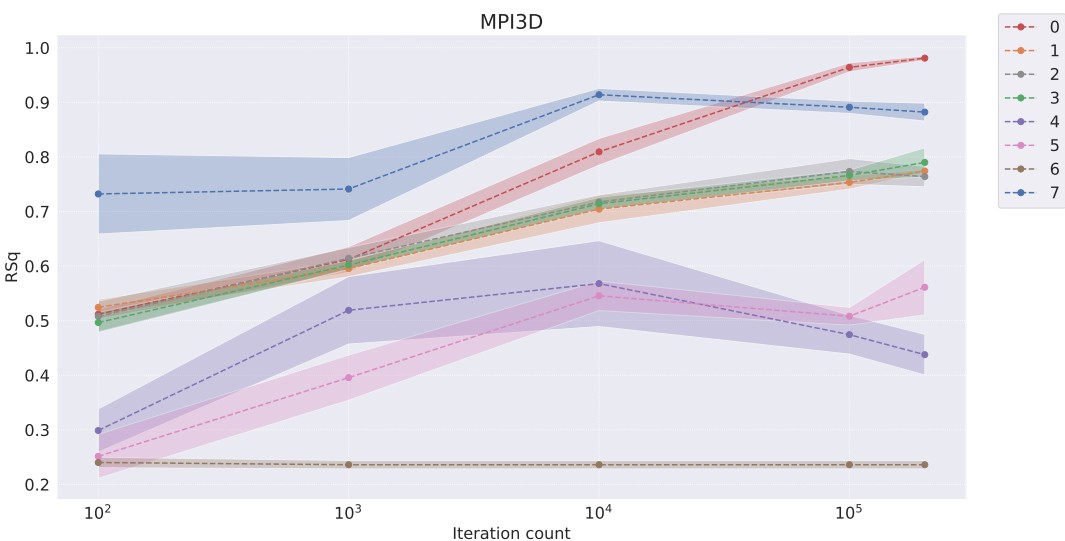

Figure 20: Convergence of representation learners as measured by FoV regression on the MPI3D dataset (dimensionality-controlled setting)

Table 30: Convergence of representation learners as measured by FoV regression on the Cars3D dataset

| Models | $R^2$ score | | | | |
|---|---|---|---|---|---|
| | $10^2$ iterations | $10^3$ iterations | $10^4$ iterations | $10^5$ iterations | $2 \times 10^5$ iterations |
| Symbolic scalar-tokened compositional representations | | | | | |
| Slow-VAE | $0.233 \pm 0.048$ | $0.816 \pm 0.017$ | $0.892 \pm 0.007$ | $0.925 \pm 0.005$ | $0.935 \pm 0.004$ |
| Slow-VAE[†] | $0.203 \pm 0.074$ | $0.797 \pm 0.009$ | $0.891 \pm 0.006$ | $0.917 \pm 0.006$ | $0.937 \pm 0.001$ |
| Ada-GVAE-k | $0.307 \pm 0.084$ | $0.821 \pm 0.016$ | $0.896 \pm 0.009$ | $0.944 \pm 0.007$ | $\mathbf{0.957 \pm 0.005}$ |
| Ada-GVAE-k[†] | $0.264 \pm 0.088$ | $0.805 \pm 0.015$ | $0.888 \pm 0.009$ | $0.940 \pm 0.007$ | $0.951 \pm 0.008$ |
| GVAE | $0.319 \pm 0.073$ | $0.831 \pm 0.012$ | $0.901 \pm 0.009$ | $0.942 \pm 0.004$ | $0.947 \pm 0.005$ |
| GVAE[†] | $0.282 \pm 0.056$ | $0.821 \pm 0.008$ | $0.895 \pm 0.009$ | $0.936 \pm 0.007$ | $0.943 \pm 0.004$ |
| MLVAE | $0.317 \pm 0.058$ | $0.820 \pm 0.007$ | $0.904 \pm 0.007$ | $\mathbf{0.945 \pm 0.004}$ | $0.948 \pm 0.007$ |
| MLVAE[†] | $0.26 \pm 0.061$ | $0.808 \pm 0.005$ | $0.900 \pm 0.009$ | $0.939 \pm 0.004$ | $0.948 \pm 0.007$ |
| Shu | $-0.016 \pm 0.016$ | $0.003 \pm 0.019$ | $0.343 \pm 0.037$ | $0.076 \pm 0.058$ | $0.075 \pm 0.051$ |
| Shu[†] | $0.012 \pm 0.007$ | $-0.009 \pm 0.02$ | $0.343 \pm 0.042$ | $0.074 \pm 0.070$ | $0.094 \pm 0.055$ |
| Symbolic vector-tokened compositional representations | | | | | |
| VCT | $0.080 \pm 0.001$ | $0.829 \pm 0.015$ | $0.877 \pm 0.007$ | $0.832 \pm 0.010$ | $0.813 \pm 0.014$ |
| VCT[†] | $0.038 \pm 0.014$ | $0.243 \pm 0.03$ | $0.655 \pm 0.011$ | $0.631 \pm 0.029$ | $0.623 \pm 0.036$ |
| COMET | $-0.029 \pm 0.011$ | $-0.035 \pm 0.023$ | $-0.035 \pm 0.023$ | $-0.035 \pm 0.023$ | $-0.035 \pm 0.023$ |
| COMET[†] | $-0.484 \pm 0.477$ | $-0.495 \pm 0.512$ | $-0.495 \pm 0.512$ | $-0.515 \pm 0.47$ | $-0.715 \pm 0.295$ |
| Fully continuous compositional representations | | | | | |
| Ours | $\mathbf{0.531 \pm 0.054}$ | $\mathbf{0.855 \pm 0.012}$ | $\mathbf{0.920 \pm 0.009}$ | $0.914 \pm 0.008$ | $0.910 \pm 0.010$ |

Table 31: Convergence of representation learners as measured by FoV regression on the Shapes3D dataset

| Models | $R^2$ score | | | | |
|---|---|---|---|---|---|
| | $10^2$ iterations | $10^3$ iterations | $10^4$ iterations | $10^5$ iterations | $2 \times 10^5$ iterations |
| Symbolic scalar-tokened compositional representations | | | | | |
| Slow-VAE | $0.597 \pm 0.048$ | $0.927 \pm 0.007$ | $0.960 \pm 0.032$ | $0.999 \pm 0.000 \ (=)$ | $\mathbf{1.000 \pm 0.000} \ (=)$ |
| Slow-VAE[†] | $0.564 \pm 0.060$ | $0.920 \pm 0.011$ | $0.955 \pm 0.037$ | $0.999 \pm 0.000 \ (=)$ | $0.999 \pm 0.000$ |
| Ada-GVAE-k | $0.684 \pm 0.068$ | $0.974 \pm 0.007$ | $0.997 \pm 0.000$ | $\mathbf{1.000 \pm 0.000} \ (=)$ | $\mathbf{1.000 \pm 0.000} \ (=)$ |
| Ada-GVAE-k[†] | $0.698 \pm 0.063$ | $0.973 \pm 0.007$ | $0.997 \pm 0.000$ | $\mathbf{1.000 \pm 0.000} \ (=)$ | $\mathbf{1.000 \pm 0.000} \ (=)$ |
| GVAE | $0.565 \pm 0.034$ | $0.951 \pm 0.015$ | $0.997 \pm 0.000$ | $\mathbf{1.000 \pm 0.000} \ (=)$ | $\mathbf{1.000 \pm 0.000} \ (=)$ |
| GVAE[†] | $0.565 \pm 0.022$ | $0.951 \pm 0.014$ | $0.998 \pm 0.001$ | $\mathbf{1.000 \pm 0.000} \ (=)$ | $\mathbf{1.000 \pm 0.000} \ (=)$ |
| MLVAE | $0.551 \pm 0.059$ | $0.959 \pm 0.012$ | $0.997 \pm 0.000$ | $0.999 \pm 0.000 \ (=)$ | $\mathbf{1.000 \pm 0.000} \ (=)$ |
| MLVAE[†] | $0.556 \pm 0.055$ | $0.960 \pm 0.010$ | $0.996 \pm 0.000$ | $0.999 \pm 0.000 \ (=)$ | $\mathbf{1.000 \pm 0.000} \ (=)$ |
| Shu | $0.327 \pm 0.13$ | $0.497 \pm 0.106$ | $0.631 \pm 0.087$ | $0.471 \pm 0.027$ | $0.427 \pm 0.082$ |
| Shu[†] | $0.461 \pm 0.075$ | $0.525 \pm 0.107$ | $0.632 \pm 0.088$ | $0.471 \pm 0.008$ | $0.419 \pm 0.082$ |
| Symbolic vector-tokened compositional representations | | | | | |
| VCT | $0.886 \pm 0.033$ | $\mathbf{0.997 \pm 0.000}$ | $0.995 \pm 0.001$ | $0.97 \pm 0.013$ | $0.954 \pm 0.008$ |
| VCT[†] | $0.255 \pm 0.049$ | $0.819 \pm 0.077$ | $0.896 \pm 0.021$ | $0.888 \pm 0.028$ | $0.843 \pm 0.04$ |
| COMET | $0.395 \pm 0.063$ | $0.772 \pm 0.016$ | $0.968 \pm 0.011$ | $\mathbf{1.000 \pm 0.000} \ (=)$ | $\mathbf{1.000 \pm 0.000} \ (=)$ |
| COMET[†] | $0.474 \pm 0.062$ | $0.809 \pm 0.013$ | $0.970 \pm 0.011$ | $\mathbf{1.000 \pm 0.000} \ (=)$ | $\mathbf{1.000 \pm 0.000} \ (=)$ |
| Fully continuous compositional representations | | | | | |
| Ours | $\mathbf{0.981 \pm 0.003}$ | $0.989 \pm 0.007$ | $\mathbf{1.000 \pm 0.000} \ (=)$ | $\mathbf{1.000 \pm 0.000} \ (=)$ | $\mathbf{1.000 \pm 0.000} \ (=)$ |

Table 32: Convergence of representation learners as measured by FoV regression on the MPI3D dataset

| Models | $R^2$ score | | | | |
|---|---|---|---|---|---|
| | $10^2$ iterations | $10^3$ iterations | $10^4$ iterations | $10^5$ iterations | $2 \times 10^5$ iterations |
| Symbolic scalar-tokened compositional representations | | | | | |
| Slow-VAE | $0.557 \pm 0.012$ | $0.634 \pm 0.015$ | $0.818 \pm 0.016$ | $0.960 \pm 0.013$ | $0.980 \pm 0.005$ |
| Slow-VAE[†] | $0.512 \pm 0.009$ | $0.612 \pm 0.023$ | $0.809 \pm 0.025$ | $\mathbf{0.964 \pm 0.008}$ | $\mathbf{0.981 \pm 0.004}$ |
| Ada-GVAE-k | $0.519 \pm 0.023$ | $0.611 \pm 0.028$ | $0.694 \pm 0.007$ | $0.750 \pm 0.008$ | $0.762 \pm 0.014$ |
| Ada-GVAE-k[†] | $0.524 \pm 0.015$ | $0.595 \pm 0.015$ | $0.704 \pm 0.025$ | $0.753 \pm 0.012$ | $0.774 \pm 0.003$ |
| GVAE | $0.511 \pm 0.037$ | $0.602 \pm 0.021$ | $0.724 \pm 0.012$ | $0.748 \pm 0.013$ | $0.771 \pm 0.018$ |
| GVAE[†] | $0.508 \pm 0.028$ | $0.614 \pm 0.020$ | $0.717 \pm 0.013$ | $0.773 \pm 0.024$ | $0.764 \pm 0.019$ |
| MLVAE | $0.504 \pm 0.016$ | $0.605 \pm 0.013$ | $0.717 \pm 0.020$ | $0.780 \pm 0.016$ | $0.773 \pm 0.031$ |
| MLVAE[†] | $0.497 \pm 0.019$ | $0.602 \pm 0.008$ | $0.714 \pm 0.012$ | $0.766 \pm 0.010$ | $0.790 \pm 0.027$ |
| Shu | $0.270 \pm 0.041$ | $0.498 \pm 0.095$ | $0.592 \pm 0.082$ | $0.470 \pm 0.036$ | $0.441 \pm 0.041$ |
| Shu[†] | $0.299 \pm 0.040$ | $0.519 \pm 0.062$ | $0.568 \pm 0.079$ | $0.474 \pm 0.036$ | $0.438 \pm 0.037$ |
| Symbolic vector-tokened compositional representations | | | | | |
| VCT | $0.316 \pm 0.016$ | $0.316 \pm 0.011$ | $0.516 \pm 0.02$ | $0.543 \pm 0.000$ | $0.456 \pm 0.061$ |
| VCT[†] | $0.251 \pm 0.04$ | $0.396 \pm 0.041$ | $0.546 \pm 0.027$ | $0.508 \pm 0.016$ | $0.561 \pm 0.051$ |
| COMET | $0.266 \pm 0.004$ | $0.266 \pm 0.004$ | $0.266 \pm 0.004$ | $0.266 \pm 0.004$ | $0.266 \pm 0.004$ |
| COMET[†] | $0.240 \pm 0.010$ | $0.236 \pm 0.008$ | $0.236 \pm 0.008$ | $0.236 \pm 0.008$ | $0.236 \pm 0.008$ |
| Fully continuous compositional representations | | | | | |
| Ours | $\mathbf{0.732 \pm 0.073}$ | $\mathbf{0.741 \pm 0.058}$ | $\mathbf{0.914 \pm 0.012}$ | $0.891 \pm 0.011$ | $0.882 \pm 0.016$ |

**Abstract Visual Reasoning**

We present now present our full suite of results for the downstream task of abstract visual reasoning. As demonstrated in Table 33 and the corresponding Figures 21 and 22, representations produced by our model at *only* $10^2$ iterations of training are able to be leveraged by downstream models to achieve a 80.04% accuracy for the challenging abstract visual reasoning task, in contrast to the value of 63.1% obtained by the best performing baseline, representing a 26.78% performance increase.

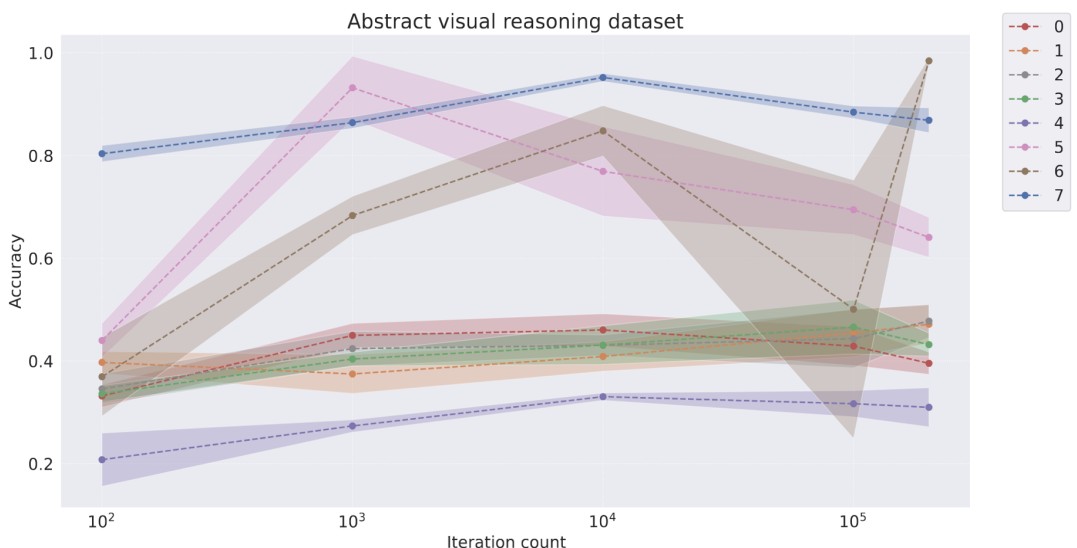

Figure 21: Convergence of representation learners as measured by classification performance on the abstract visual reasoning dataset (original setting)

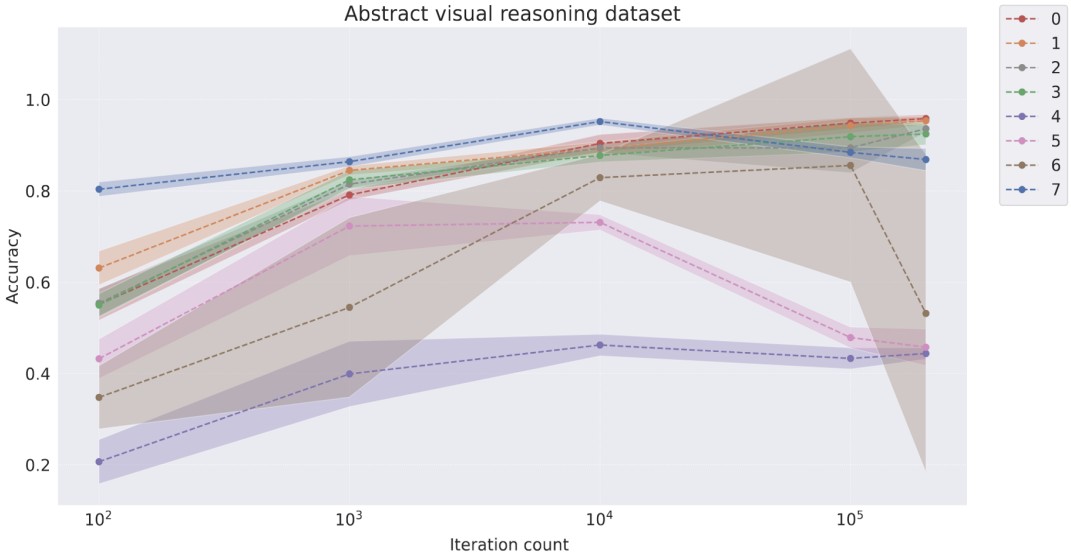

Figure 22: Convergence of representation learners as measured by classification performance on the abstract visual reasoning dataset (dimensionality-controlled setting)

Table 33: Convergence of representation learners as measured by classification performance on the abstract visual reasoning dataset

| Models | Classification accuracy | | | | |
|---|---|---|---|---|---|
| | $10^2$ iterations | $10^3$ iterations | $10^4$ iterations | $10^5$ iterations | $2 \times 10^5$ iterations |
| *Symbolic scalar-tokened compositional representations* | | | | | |
| Slow-VAE | $0.332 \pm 0.022$ | $0.450 \pm 0.023$ | $0.460 \pm 0.031$ | $0.429 \pm 0.038$ | $0.396 \pm 0.022$ |
| Slow-VAE[†] | $0.552 \pm 0.035$ | $0.791 \pm 0.011$ | $0.904 \pm 0.020$ | $\mathbf{0.949 \pm 0.012}$ | *0.959 ± 0.009* |
| Ada-GVAE-k | $0.397 \pm 0.021$ | $0.375 \pm 0.038$ | $0.409 \pm 0.029$ | $0.455 \pm 0.047$ | $0.472 \pm 0.037$ |
| Ada-GVAE-k[†] | *0.631 ± 0.037* | $0.845 \pm 0.010$ | $0.892 \pm 0.015$ | *0.943 ± 0.017* | $0.954 \pm 0.009$ |
| GVAE | $0.346 \pm 0.029$ | $0.424 \pm 0.034$ | $0.431 \pm 0.020$ | $0.444 \pm 0.057$ | $0.478 \pm 0.032$ |
| GVAE[†] | $0.554 \pm 0.031$ | $0.815 \pm 0.008$ | $0.894 \pm 0.011$ | $0.894 \pm 0.055$ | $0.936 \pm 0.008$ |
| MLVAE | $0.336 \pm 0.016$ | $0.404 \pm 0.013$ | $0.431 \pm 0.038$ | $0.466 \pm 0.052$ | $0.432 \pm 0.023$ |
| MLVAE[†] | $0.550 \pm 0.025$ | $0.824 \pm 0.021$ | $0.878 \pm 0.014$ | $0.919 \pm 0.034$ | $0.925 \pm 0.024$ |
| Shu | $0.208 \pm 0.052$ | $0.273 \pm 0.012$ | $0.331 \pm 0.007$ | $0.317 \pm 0.025$ | $0.310 \pm 0.038$ |
| Shu[†] | $0.207 \pm 0.048$ | $0.399 \pm 0.072$ | $0.462 \pm 0.024$ | $0.433 \pm 0.023$ | $0.444 \pm 0.013$ |
| *Symbolic vector-tokened compositional representations* | | | | | |
| VCT | $0.440 \pm 0.033$ | $\mathbf{0.932 \pm 0.062}$ | $0.769 \pm 0.087$ | $0.695 \pm 0.049$ | $0.641 \pm 0.039$ |
| VCT[†] | $0.432 \pm 0.043$ | $0.723 \pm 0.065$ | $0.731 \pm 0.017$ | $0.479 \pm 0.023$ | $0.458 \pm 0.040$ |
| COMET | $0.369 \pm 0.077$ | $0.683 \pm 0.037$ | $0.848 \pm 0.049$ | $0.501 \pm 0.251$ | $\mathbf{0.984 \pm 0.006}$ |
| COMET[†] | $0.348 \pm 0.069$ | $0.545 \pm 0.197$ | $0.829 \pm 0.051$ | $0.856 \pm 0.256$ | $0.532 \pm 0.348$ |
| *Fully continuous compositional representations* | | | | | |
| Ours | $\mathbf{0.804 \pm 0.016}$ | *0.864 ± 0.011* | $\mathbf{0.952 \pm 0.008}$ | $0.884 \pm 0.012$ | $0.869 \pm 0.024$ |

## C.5 Downstream Performance

We present our full suite of experimental results that empirically demonstrate the utility of our Soft TPR representation from the perspective of downstream models, by considering the sample efficiency, and low-sample regime performance of downstream models on the tasks of FoV regression, and abstract visual reasoning.

### C.5.1 Sample Efficiency Results

For sample efficiency, as mentioned in Section 5.2, and in line with [25], we compute a ratio-based metric obtained by dividing the performance of the downstream model when trained using a restricted number of 100, 250, 500, 1,000 and 10,000 samples, by its performance when trained using all samples (19,104, 480,000, 1,036,800 and 100,000 samples for the tasks of regression on the Cars3D, Shapes3D, MPI3D datasets, and the abstract visual reasoning task respectively). As this metric is dependent on the performance of downstream models when trained using all samples, we do not compute this metric for representation learners where the corresponding downstream models achieve an $R^2$ score of less than $0.5$ for regression, as this may produce sample efficiency scores with very little semantic meaning (e.g. a model that achieves a sample efficiency score of 0.9 when its final $R^2$ score is 0.1). As a result, we remove Shu from Shapes3D sample efficiency calculations, and COMET and Shu from the Cars3D sample efficiency calculations.

As many models for the abstract visual reasoning task have low classification accuracies on the held-out test set following training with the maximal number of 100,000 samples, we do not compute sample efficiencies for this task, and instead refer readers to results in Section C.5.2 for the raw classification accuracies associated with each model.

Note that for all box plots, we follow standard convention, and display the median in each box with a solid line, where the box shows the quartiles of the corresponding values, and the whiskers extend to 1.5 times the interquartile range. We again, use the same legend, where grey denotes SlowVAE, orange denotes AdaGVAE-k, green denotes GVAE, red denotes MLVAE, purple denotes Shu, pink denotes VCT, brown denotes COMET, and blue denotes our model, Soft TPR Autoencoder.

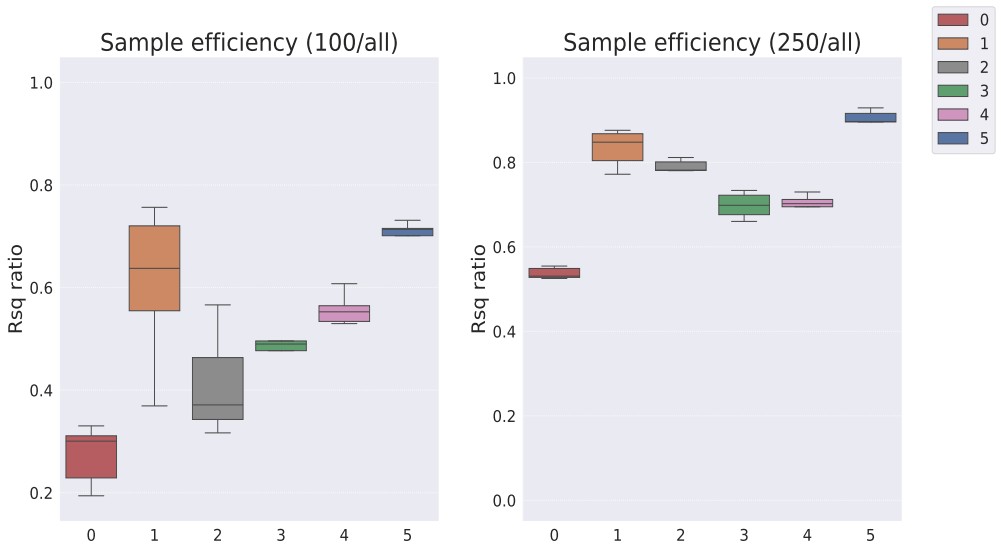

Figure 23: Downstream regression model sample efficiency on the Cars3D dataset (original setting).

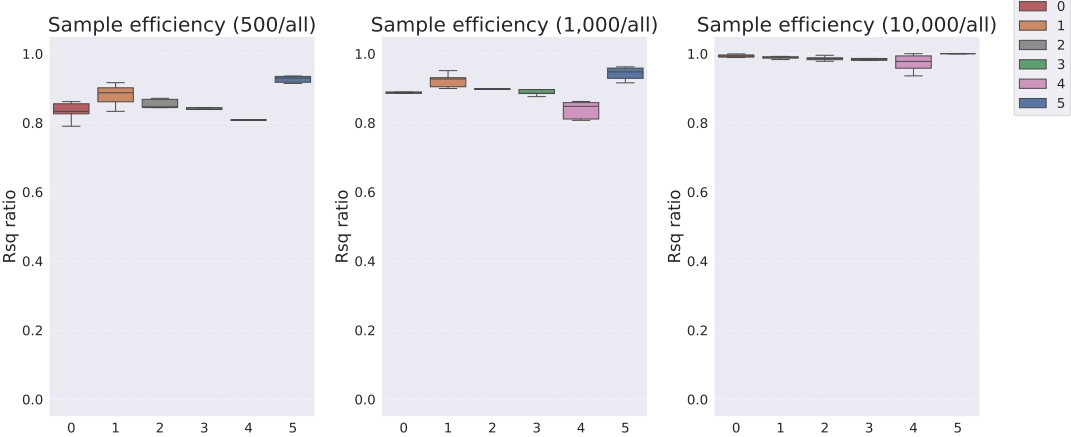

Figure 24: Downstream regression model sample efficiency on the Cars3D dataset (original setting).

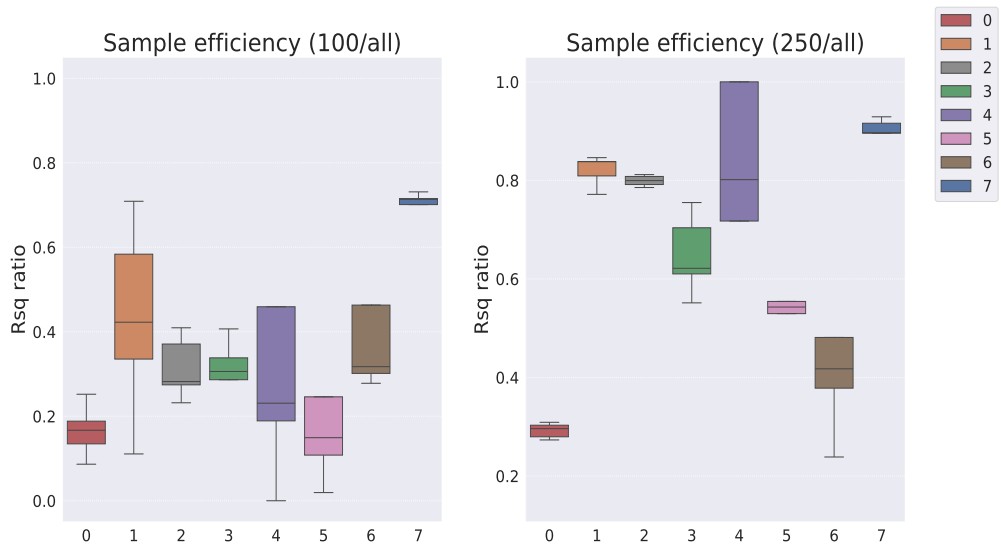

Figure 25: Downstream regression model sample efficiency on the Cars3D dataset (dimensionality-controlled setting).

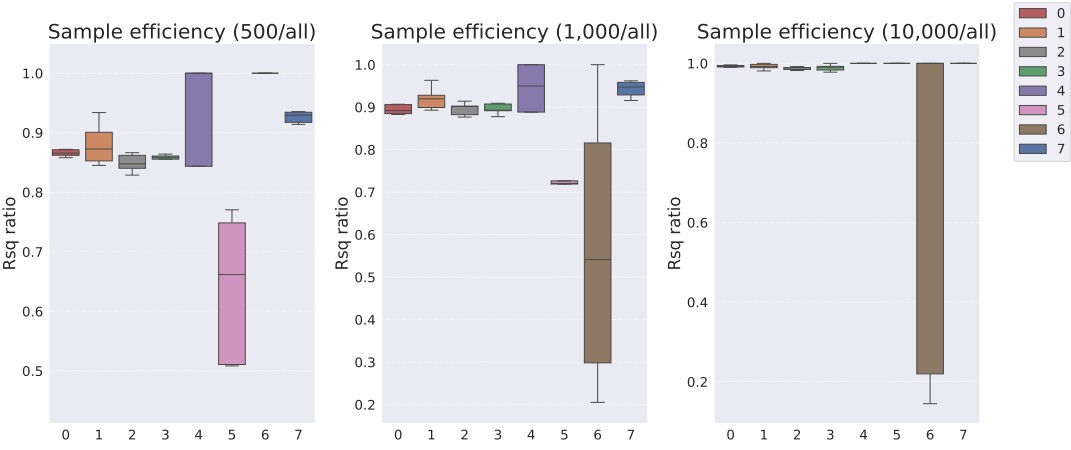

Figure 26: Downstream regression model sample efficiency on the Cars3D dataset (dimensionality-controlled setting).

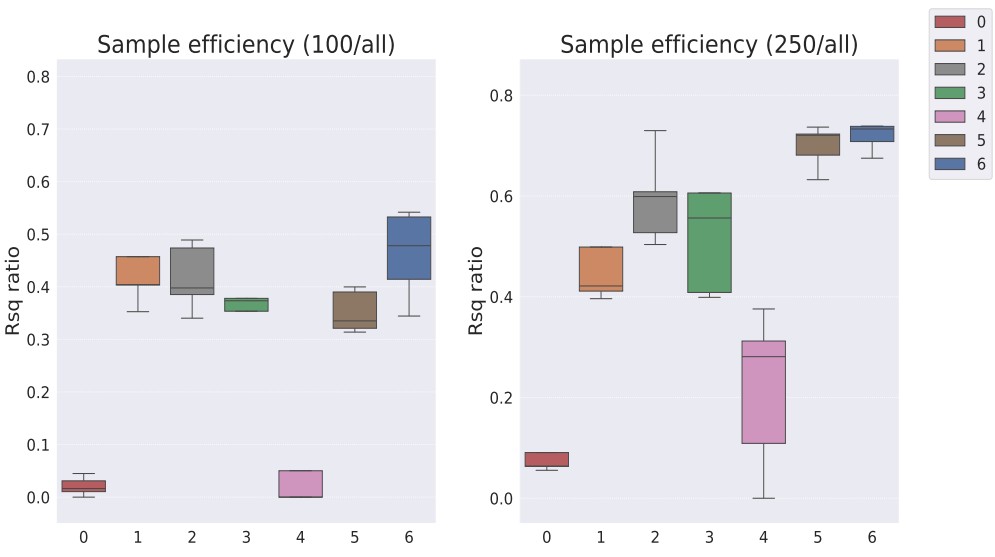

Figure 27: Downstream regression model sample efficiency on the Shapes3D dataset (original setting).

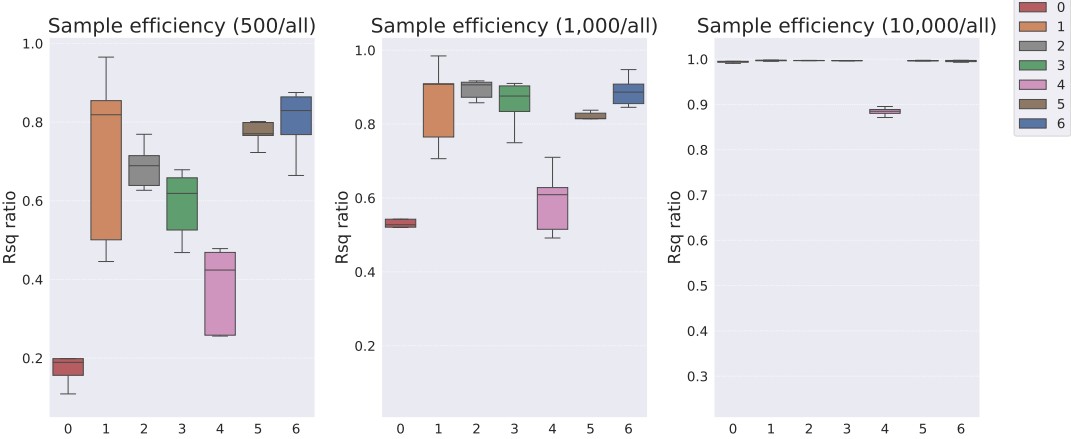

Figure 28: Downstream regression model sample efficiency on the Shapes3D dataset (original setting).

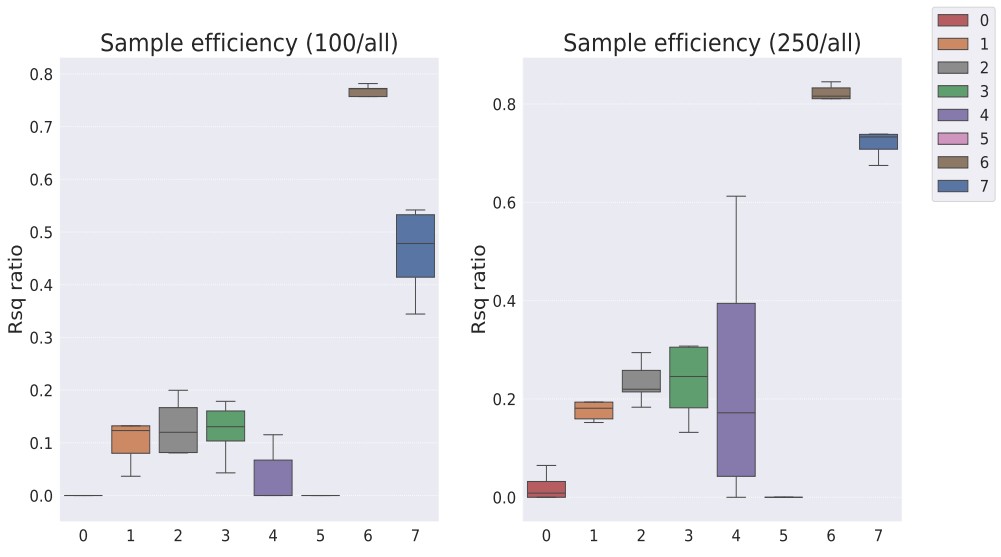

Figure 29: Downstream regression model sample efficiency on the Shapes3D dataset (dimensionality-controlled setting).

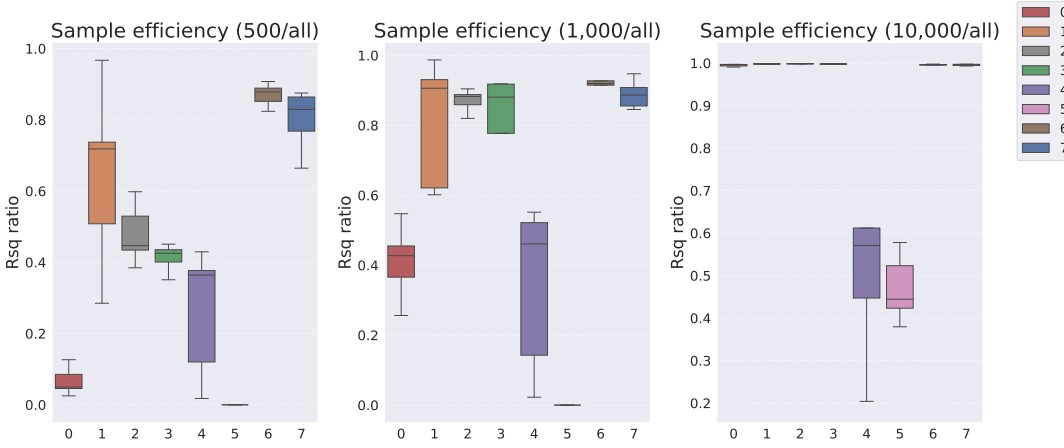

Figure 30: Downstream regression model sample efficiency on the Shapes3D dataset (dimensionality-controlled setting).

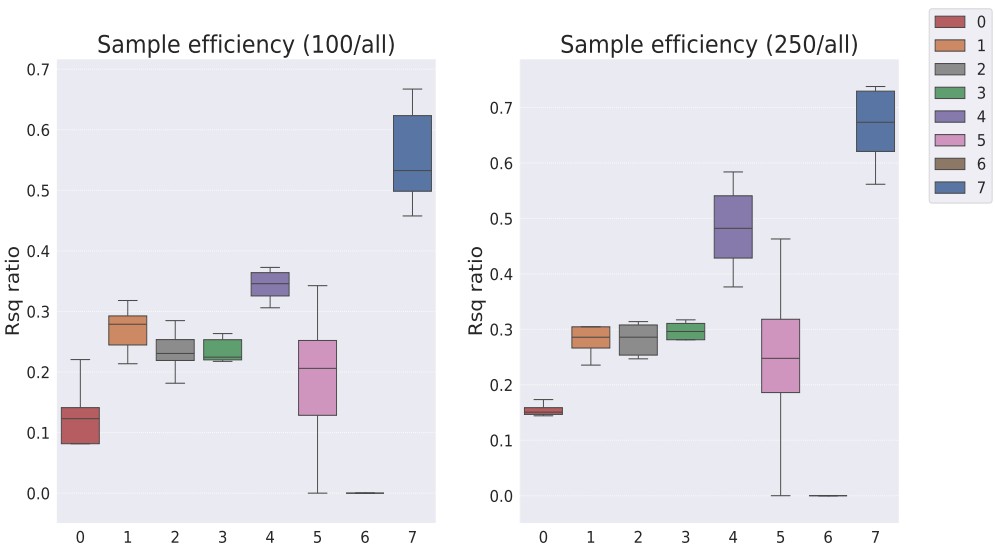

Figure 31: Downstream regression model sample efficiency on the MPI3D dataset (original setting).

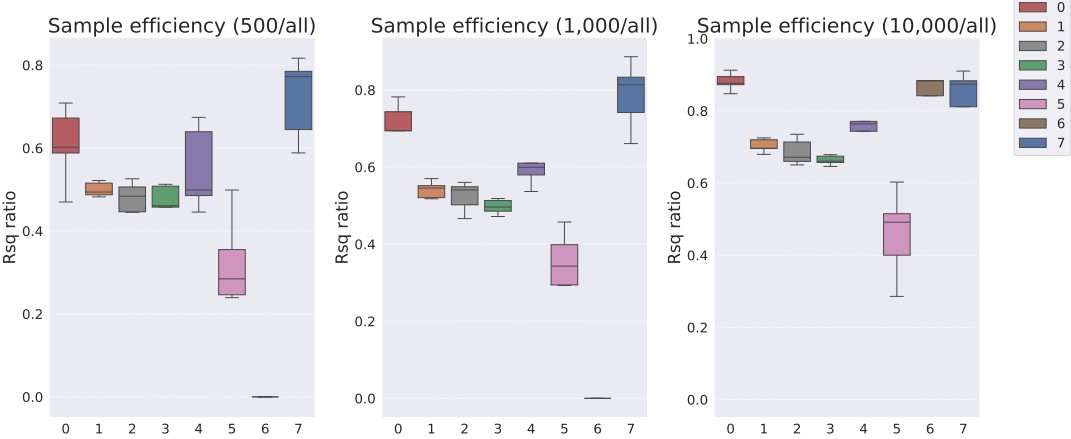

Figure 32: Downstream regression model sample efficiency on the MPI3D dataset (original setting).

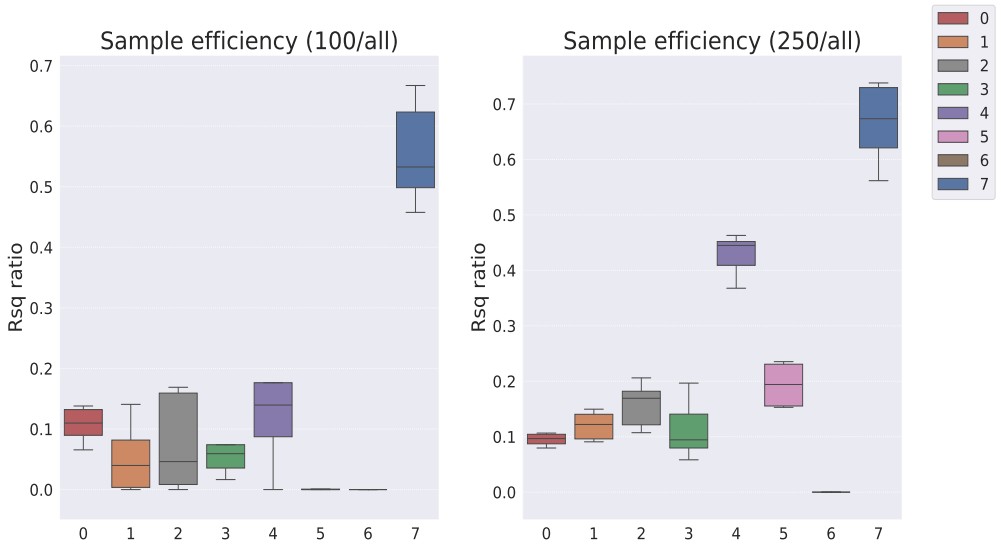

Figure 33: Downstream regression model sample efficiency on the MPI3D dataset (dimensionality-controlled setting).

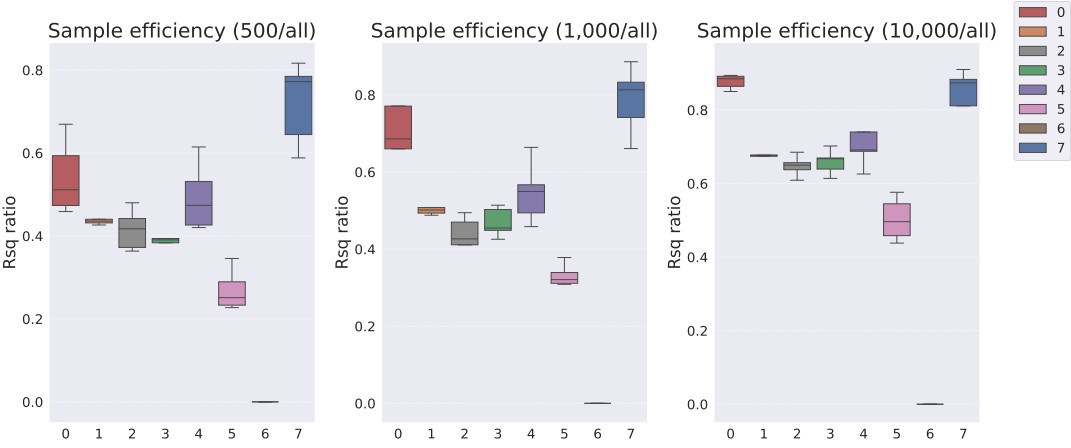

Figure 34: Downstream regression model sample efficiency on the MPI3D dataset (dimensionality-controlled setting).

Table 34: Downstream regression model sample efficiency on the Cars3D dataset

| Models | $R^2$ ratio | | | | |
|---|---|---|---|---|---|
| | $10^2$/all samples | $2.5 \times 10^2$/all samples | $5 \times 10^3$/all samples | $10^4$/all samples | $10^5$/all samples |
| Symbolic scalar-tokened compositional representations | | | | | |
| Slow-VAE | $0.273 \pm 0.052$ | $0.537 \pm 0.012$ | $0.833 \pm 0.025$ | $0.882 \pm 0.018$ | *$0.994 \pm 0.004$* |
| Slow-VAE[†] | $0.166 \pm 0.055$ | $0.292 \pm 0.014$ | $0.866 \pm 0.005$ | $0.895 \pm 0.010$ | $0.993 \pm 0.002$ |
| Ada-GVAE-k | $0.610 \pm 0.131$ | $0.834 \pm 0.040$ | *$0.881 \pm 0.026$* | *$0.922 \pm 0.019$* | $0.989 \pm 0.005$ |
| Ada-GVAE-k[†] | $0.452 \pm 0.156$ | $0.821 \pm 0.028$ | $0.879 \pm 0.029$ | $0.920 \pm 0.021$ | $0.992 \pm 0.007$ |
| GVAE | $0.412 \pm 0.092$ | $0.791 \pm 0.013$ | $0.855 \pm 0.012$ | $0.900 \pm 0.008$ | $0.986 \pm 0.006$ |
| GVAE[†] | $0.314 \pm 0.066$ | *$0.799 \pm 0.010$* | $0.849 \pm 0.014$ | $0.895 \pm 0.014$ | $0.987 \pm 0.004$ |
| MLVAE | $0.477 \pm 0.069$ | $0.698 \pm 0.027$ | $0.844 \pm 0.015$ | $0.893 \pm 0.016$ | $0.984 \pm 0.010$ |
| MLVAE[†] | $0.305 \pm 0.071$ | $0.648 \pm 0.072$ | $0.854 \pm 0.013$ | $0.896 \pm 0.012$ | $0.989 \pm 0.008$ |
| Symbolic vector-tokened compositional representations | | | | | |
| VCT | $0.558 \pm 0.028$ | $0.694 \pm 0.034$ | $0.808 \pm 0.013$ | $0.837 \pm 0.023$ | $0.973 \pm 0.024$ |
| VCT[†] | $0.197 \pm 0.151$ | $0.640 \pm 0.113$ | $0.640 \pm 0.113$ | $0.725 \pm 0.038$ | **$1.000 \pm 0.000$** (=) |
| Fully continuous compositional representations | | | | | |
| Ours | **$0.705 \pm 0.023$** | **$0.889 \pm 0.042$** | **$0.926 \pm 0.009$** | **$0.942 \pm 0.018$** | **$1.000 \pm 0.000$** (=) |

Table 35: Downstream regression model sample efficiency on the Shapes3D dataset

| Models | $R^2$ ratio | | | | |
|---|---|---|---|---|---|
| | $10^2$/all samples | $2.5 \times 10^2$/all samples | $5 \times 10^3$/all samples | $10^4$/all samples | $10^5$/all samples |
| Symbolic scalar-tokened compositional representations | | | | | |
| Slow-VAE | $0.021 \pm 0.016$ | $0.085 \pm 0.035$ | $0.183 \pm 0.051$ | $0.563 \pm 0.070$ | $0.994 \pm 0.002$ |
| Slow-VAE[†] | $0.003 \pm 0.008$ | $0.021 \pm 0.025$ | $0.066 \pm 0.036$ | $0.419 \pm 0.107$ | $0.995 \pm 0.002$ |
| Ada-GVAE-k | *$0.480 \pm 0.155$* | $0.510 \pm 0.160$ | $0.717 \pm 0.206$ | $0.855 \pm 0.103$ | $0.997 \pm 0.001$ |
| Ada-GVAE-k[†] | $0.133 \pm 0.088$ | $0.199 \pm 0.058$ | $0.643 \pm 0.231$ | $0.809 \pm 0.164$ | *$0.998 \pm 0.000$* (=) |
| GVAE | $0.417 \pm 0.056$ | $0.594 \pm 0.079$ | $0.688 \pm 0.052$ | $0.893 \pm 0.024$ | $0.997 \pm 0.000$ |
| GVAE[†] | $0.13 \pm 0.047$ | $0.234 \pm 0.038$ | $0.479 \pm 0.076$ | $0.871 \pm 0.029$ | *$0.998 \pm 0.000$* (=) |
| MLVAE | $0.371 \pm 0.041$ | $0.515 \pm 0.093$ | $0.590 \pm 0.080$ | $0.855 \pm 0.059$ | $0.997 \pm 0.000$ |
| MLVAE[†] | $0.123 \pm 0.048$ | $0.235 \pm 0.069$ | $0.413 \pm 0.035$ | $0.808 \pm 0.141$ | **$0.999 \pm 0.000$** |
| Symbolic vector-tokened compositional representations | | | | | |
| VCT | $0.020 \pm 0.025$ | $0.216 \pm 0.139$ | $0.377 \pm 0.100$ | $0.591 \pm 0.079$ | $0.884 \pm 0.008$ |
| VCT[†] | $0.000 \pm 0.000$ | $0.000 \pm 0.000$ | $0.000 \pm 0.000$ | $0.009 \pm 0.018$ | $0.470 \pm 0.071$ |
| COMET | $0.352 \pm 0.036$ | $0.699 \pm 0.038$ | $0.772 \pm 0.028$ | $0.812 \pm 0.025$ | $0.997 \pm 0.001$ |
| COMET[†] | **$0.752 \pm 0.039$** | **$0.810 \pm 0.034$** | **$0.870 \pm 0.029$** | **$0.916 \pm 0.031$** | $0.996 \pm 0.001$ |
| Fully continuous compositional representations | | | | | |
| Ours | $0.464 \pm 0.071$ | *$0.730 \pm 0.038$* | *$0.804 \pm 0.075$* | *$0.889 \pm 0.035$* | $0.996 \pm 0.001$ |

Table 36: Downstream regression model sample efficiency on the MPI3D dataset

| Models | $R^2$ ratio | | | | |
|---|---|---|---|---|---|
| | $10^2$/all samples | $2.5 \times 10^2$/all samples | $5 \times 10^3$/all samples | $10^4$/all samples | $10^5$/all samples |
| Symbolic scalar-tokened compositional representations | | | | | |
| Slow-VAE | $0.130 \pm 0.051$ | $0.155 \pm 0.011$ | *$0.608 \pm 0.082$* | *$0.692 \pm 0.081$* | *$0.881 \pm 0.022$* |
| Slow-VAE[†] | $0.107 \pm 0.027$ | $0.095 \pm 0.011$ | $0.541 \pm 0.079$ | $0.668 \pm 0.117$ | $0.877 \pm 0.017$ |
| Ada-GVAE-k | *$0.270 \pm 0.037$* | $0.279 \pm 0.026$ | $0.500 \pm 0.016$ | $0.541 \pm 0.020$ | $0.703 \pm 0.017$ |
| Ada-GVAE-k[†] | $0.053 \pm 0.053$ | $0.120 \pm 0.023$ | $0.442 \pm 0.018$ | $0.504 \pm 0.015$ | $0.680 \pm 0.012$ |
| GVAE | $0.234 \pm 0.035$ | $0.282 \pm 0.027$ | $0.481 \pm 0.032$ | $0.524 \pm 0.035$ | $0.686 \pm 0.033$ |
| GVAE[†] | $0.077 \pm 0.073$ | $0.157 \pm 0.037$ | $0.415 \pm 0.043$ | $0.443 \pm 0.034$ | $0.648 \pm 0.025$ |
| MLVAE | $0.236 \pm 0.019$ | $0.288 \pm 0.030$ | $0.462 \pm 0.051$ | $0.497 \pm 0.017$ | $0.663 \pm 0.012$ |
| MLVAE[†] | $0.065 \pm 0.042$ | $0.114 \pm 0.049$ | $0.387 \pm 0.024$ | $0.469 \pm 0.034$ | $0.659 \pm 0.030$ |
| Shu | $0.343 \pm 0.024$ | *$0.482 \pm 0.075$* | $0.549 \pm 0.091$ | $0.601 \pm 0.047$ | $0.750 \pm 0.058$ |
| Shu[†] | $0.143 \pm 0.103$ | $0.427 \pm 0.035$ | $0.493 \pm 0.073$ | $0.547 \pm 0.070$ | $0.714 \pm 0.067$ |
| Symbolic vector-tokened compositional representations | | | | | |
| VCT | $0.189 \pm 0.107$ | $0.246 \pm 0.137$ | $0.294 \pm 0.145$ | $0.312 \pm 0.14$ | $0.418 \pm 0.180$ |
| VCT[†] | $0.039 \pm 0.088$ | $0.168 \pm 0.082$ | $0.230 \pm 0.110$ | $0.279 \pm 0.127$ | $0.502 \pm 0.052$ |
| COMET | $0.000 \pm 0.000$ | $0.000 \pm 0.000$ | $0.000 \pm 0.000$ | $0.000 \pm 0.000$ | $0.823 \pm 0.139$ |
| COMET[†] | $0.000 \pm 0.000$ | $0.000 \pm 0.000$ | $0.000 \pm 0.000$ | $0.000 \pm 0.000$ | $0.187 \pm 0.374$ |
| Fully continuous compositional representations | | | | | |
| Ours | $\mathbf{0.556 \pm 0.078}$ | $\mathbf{0.665 \pm 0.067}$ | $\mathbf{0.721 \pm 0.089}$ | $\mathbf{0.787 \pm 0.078}$ | $\mathbf{0.858 \pm 0.040}$ |

### C.5.2 Low Sample Regime Results

To evaluate the utility of our Soft TPR representation from the perspective of downstream models, we additionally evaluate the raw performance of downstream models as a function of the number of samples they have been trained on (again considering 100, 250, 500, 1,000, 10,000 and all samples). We find that our Soft TPR representation contributes to a substantial performance boost in the downstream model's performance in a low-sample regime where the downstream model has been trained with 100, 250, 500, and 1,000 samples. We present our full suite of experimental results, and highlight the particular performance differentials conferred by our representational form in the low-sample regime.

**FoV Regression**

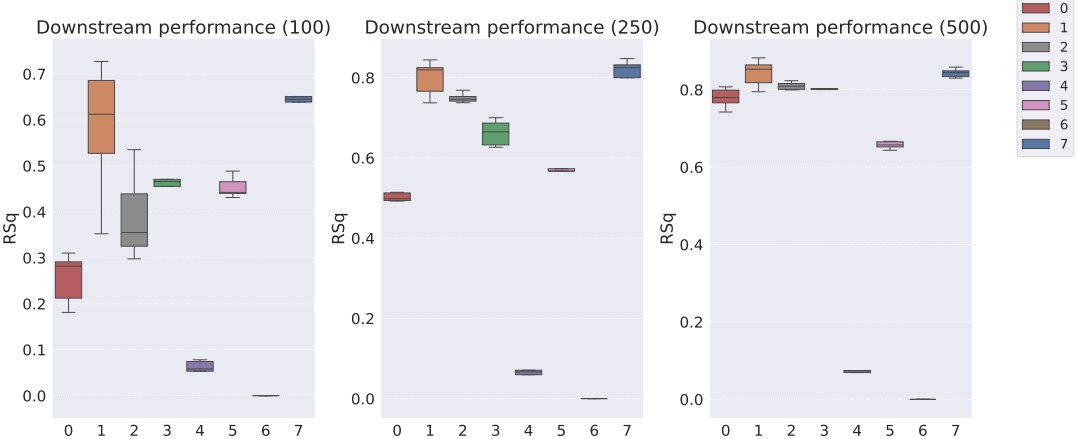

Figure 35: Downstream regression model $R^2$ scores on the Cars3D dataset (original setting).

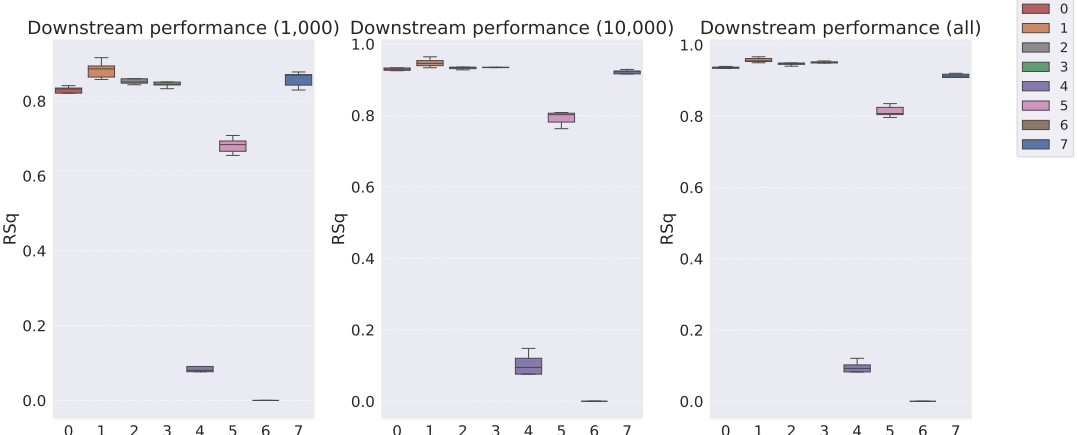

Figure 36: Downstream regression model $R^2$ scores on the Cars3D dataset (original setting).

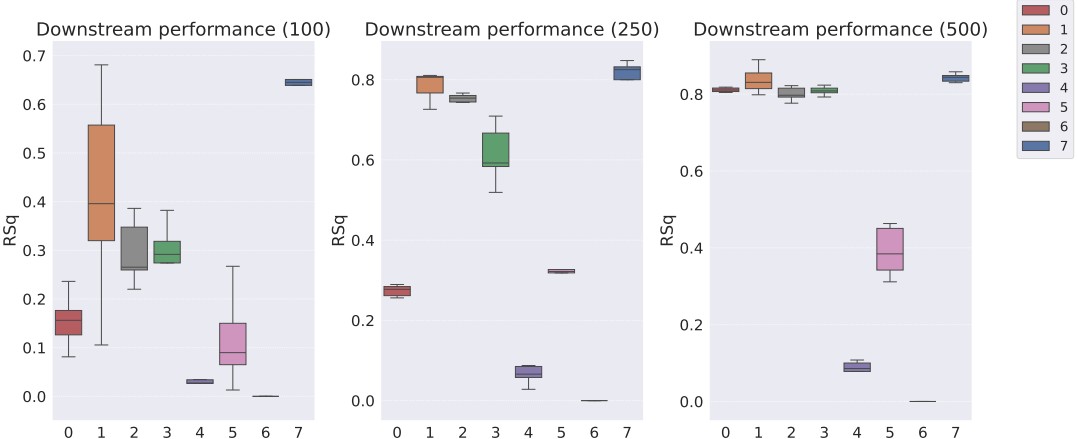

Figure 37: Downstream regression model $R^2$ scores on the Cars3D dataset (dimensionality-controlled setting).

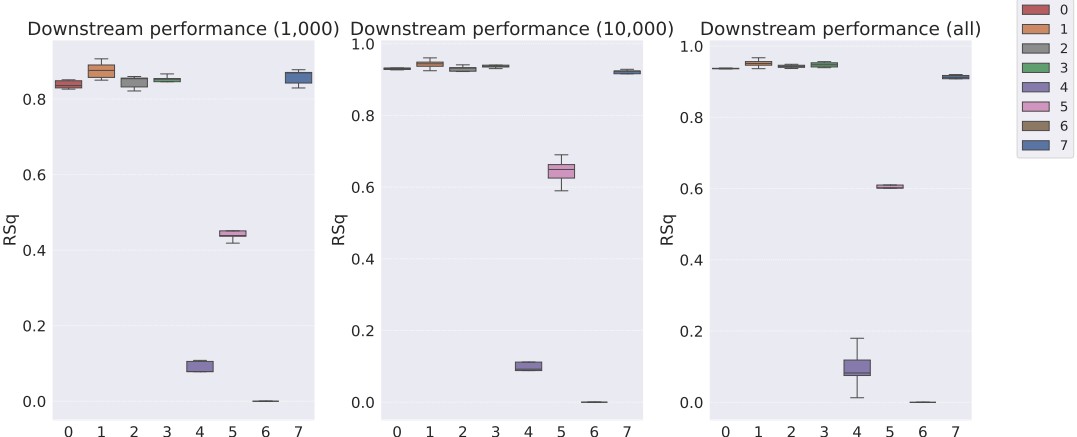

Figure 38: Downstream regression model $R^2$ scores on the Cars3D dataset (dimensionality-controlled setting).

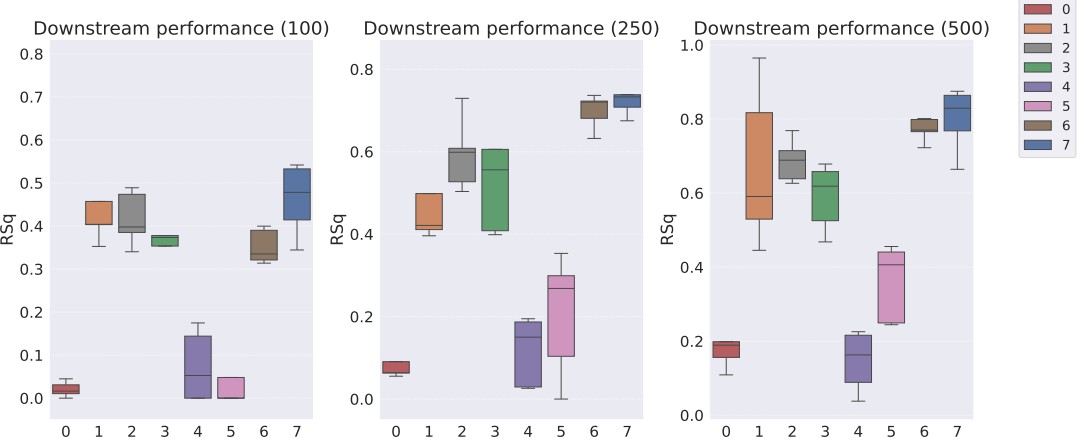

Figure 39: Downstream regression model $R^2$ scores on the Shapes3D dataset (original setting).

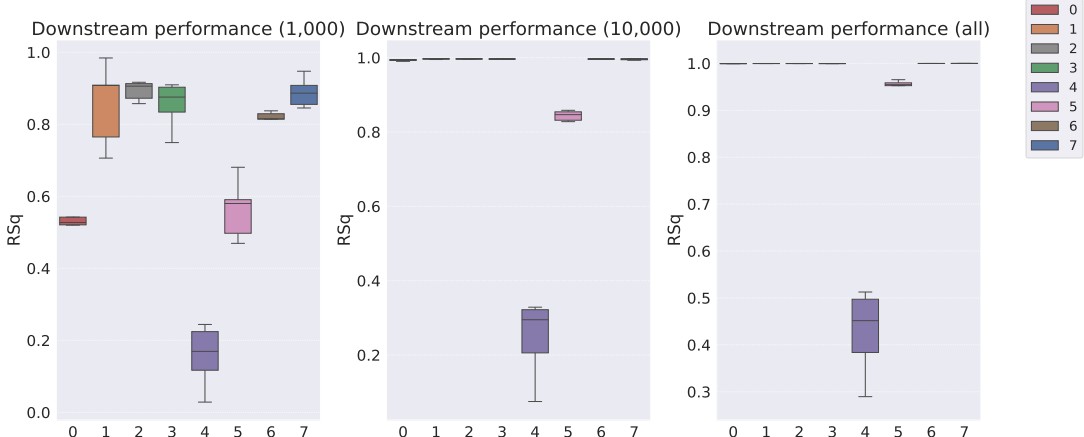

Figure 40: Downstream regression model $R^2$ scores on the Shapes3D dataset (original setting).

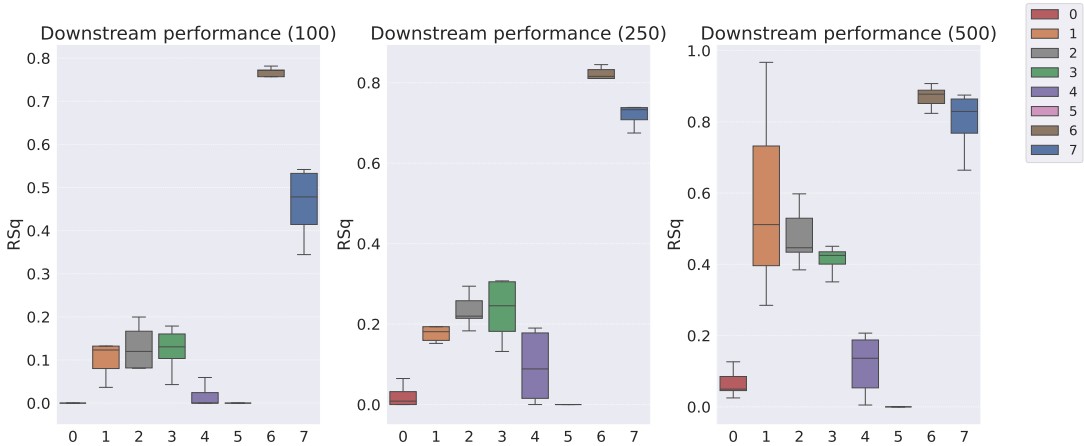

Figure 41: Downstream regression model $R^2$ scores on the Shapes3D dataset (dimensionality-controlled setting).

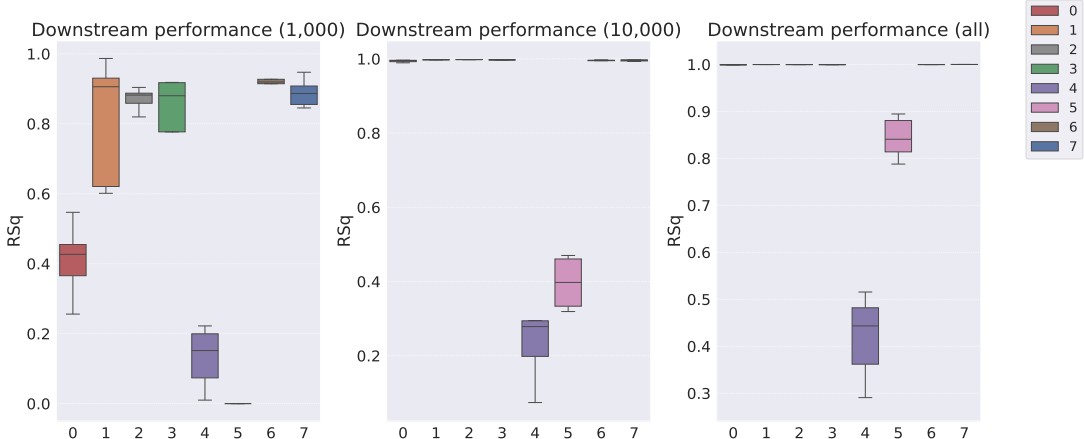

Figure 42: Downstream regression model $R^2$ scores on the Shapes3D dataset (dimensionality-controlled setting).

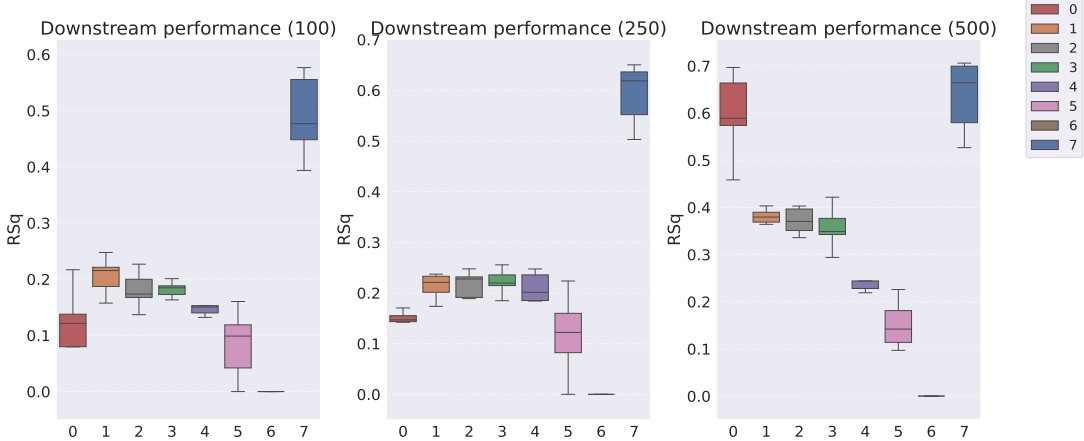

Figure 43: Downstream regression model $R^2$ scores on the MPI3D dataset (original setting).

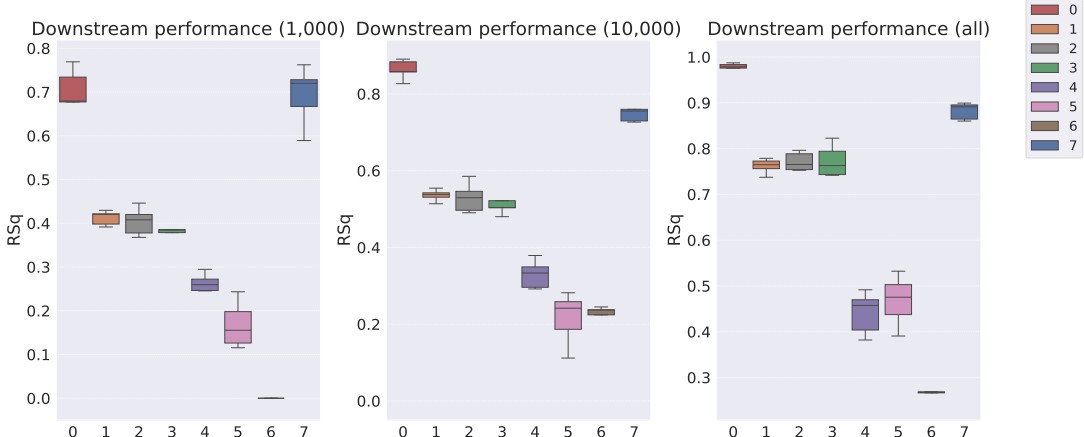

Figure 44: Downstream regression model $R^2$ scores on the MPI3D dataset (original setting).

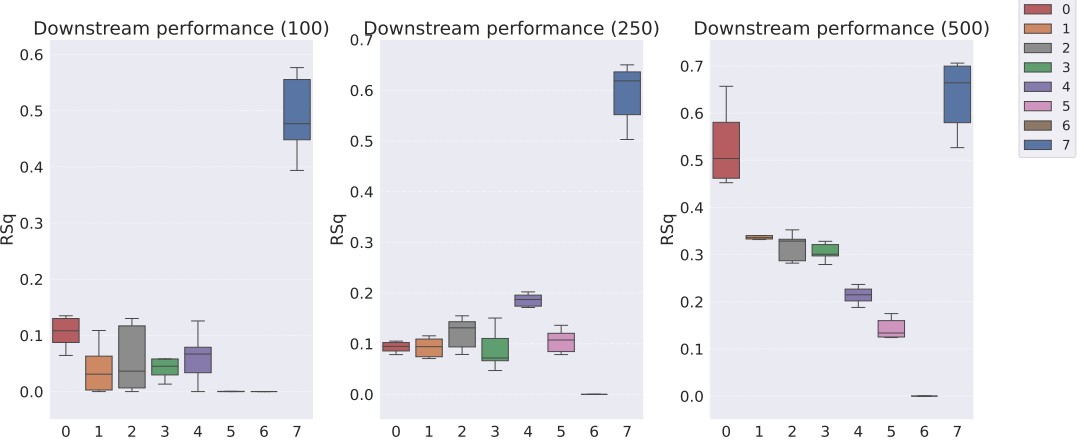

Figure 45: Downstream regression model $R^2$ scores on the MPI3D dataset (dimensionality-controlled setting).

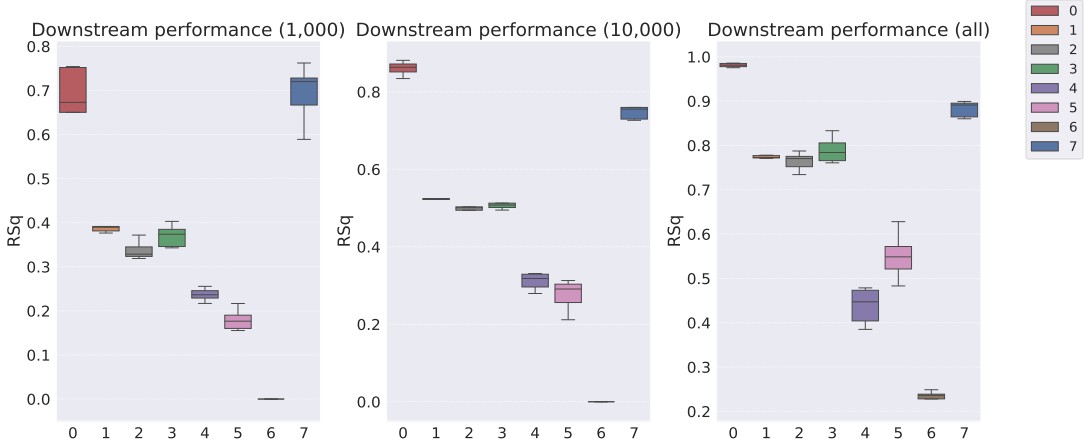

Figure 46: Downstream regression model $R^2$ scores on the MPI3D dataset (dimensionality-controlled setting).

Table 37: Downstream regression model performance on the Cars3D dataset

| Models | $R^2$ score | | | | | |
|---|---|---|---|---|---|---|
| | $10^2$ samples | $2.5 \times 10^2$ samples | $5 \times 10^3$ samples | $10^4$ samples | $10^5$ samples | all samples |
| Symbolic scalar-tokened compositional representations | | | | | | |
| Slow-VAE | $0.255 \pm 0.05$ | $0.502 \pm 0.009$ | $0.779 \pm 0.023$ | $0.825 \pm 0.016$ | $0.929 \pm 0.003$ | $0.935 \pm 0.004$ |
| Slow-VAE$^\dagger$ | $0.155 \pm 0.052$ | $0.274 \pm 0.013$ | $0.811 \pm 0.005$ | $0.838 \pm 0.010$ | $0.930 \pm 0.002$ | $0.935 \pm 0.004$ |
| Ada-GVAE-k | *$0.584 \pm 0.125$* | *$0.798 \pm 0.040$* | *$0.843 \pm 0.028$* | **$0.883 \pm 0.020$** | **$0.947 \pm 0.009$** | **$0.957 \pm 0.005$** (=) |
| Ada-GVAE-k$^\dagger$ | $0.430 \pm 0.149$ | $0.784 \pm 0.033$ | $0.836 \pm 0.028$ | *$0.875 \pm 0.018$* | *$0.944 \pm 0.010$* | **$0.957 \pm 0.005$** (=) |
| GVAE | $0.390 \pm 0.087$ | $0.750 \pm 0.011$ | $0.810 \pm 0.009$ | $0.852 \pm 0.006$ | $0.934 \pm 0.006$ | *$0.947 \pm 0.005$* (=) |
| GVAE$^\dagger$ | $0.296 \pm 0.061$ | $0.754 \pm 0.009$ | $0.801 \pm 0.016$ | $0.845 \pm 0.015$ | $0.931 \pm 0.007$ | *$0.947 \pm 0.005$* (=) |
| MLVAE | $0.452 \pm 0.063$ | $0.662 \pm 0.029$ | $0.800 \pm 0.018$ | $0.847 \pm 0.010$ | $0.933 \pm 0.007$ | $0.948 \pm 0.007$ |
| MLVAE$^\dagger$ | $0.289 \pm 0.066$ | $0.614 \pm 0.067$ | $0.809 \pm 0.011$ | $0.849 \pm 0.014$ | $0.938 \pm 0.006$ | $0.948 \pm 0.007$ |
| Shu | $0.044 \pm 0.045$ | $0.055 \pm 0.044$ | $0.061 \pm 0.047$ | $0.067 \pm 0.050$ | $0.085 \pm 0.056$ | $0.075 \pm 0.051$ |
| Shu$^\dagger$ | $0.010 \pm 0.042$ | $0.065 \pm 0.022$ | $0.079 \pm 0.031$ | $0.079 \pm 0.030$ | $0.101 \pm 0.046$ | $0.075 \pm 0.051$ |
| Symbolic vector-tokened compositional representations | | | | | | |
| VCT | $0.454 \pm 0.021$ | $0.565 \pm 0.033$ | $0.657 \pm 0.009$ | $0.681 \pm 0.019$ | $0.792 \pm 0.018$ | $0.813 \pm 0.014$ |
| VCT$^\dagger$ | $0.117 \pm 0.087$ | $0.340 \pm 0.073$ | $0.390 \pm 0.059$ | $0.444 \pm 0.019$ | $0.643 \pm 0.034$ | $0.813 \pm 0.014$ |
| COMET | $-0.058 \pm 0.027$ | $-0.098 \pm 0.054$ | $-0.074 \pm 0.036$ | $-0.147 \pm 0.101$ | $-0.036 \pm 0.028$ | $-0.035 \pm 0.023$ |
| COMET$^\dagger$ | $-0.379 \pm 0.303$ | $-0.258 \pm 0.085$ | $-0.966 \pm 0.470$ | $-0.306 \pm 0.103$ | $-0.739 \pm 0.751$ | $-0.035 \pm 0.023$ |
| Fully continuous compositional representations | | | | | | |
| Ours | **$0.642 \pm 0.023$** | **$0.809 \pm 0.037$** | **$0.843 \pm 0.010$** | $0.858 \pm 0.019$ | $0.917 \pm 0.013$ | $0.910 \pm 0.010$ |

Table 38: Downstream regression model performance on the Shapes3D dataset

| Models | $R^2$ score | | | | | |
|---|---|---|---|---|---|---|
| | $10^2$ samples | $2.5 \times 10^2$ samples | $5 \times 10^3$ samples | $10^4$ samples | $10^5$ samples | all samples |
| Symbolic scalar-tokened compositional representations | | | | | | |
| Slow-VAE | $0.019 \pm 0.017$ | $0.085 \pm 0.035$ | $0.183 \pm 0.051$ | $0.562 \pm 0.070$ | $0.993 \pm 0.002$ | **$1.000 \pm 0.000$** (=) |
| Slow-VAE$^\dagger$ | $-0.030 \pm 0.034$ | $0.018 \pm 0.028$ | $0.066 \pm 0.036$ | $0.418 \pm 0.107$ | $0.994 \pm 0.003$ | *$0.999 \pm 0.000$* |
| Ada-GVAE-k | $0.480 \pm 0.155$ | $0.510 \pm 0.160$ | $0.666 \pm 0.170$ | $0.854 \pm 0.103$ | *$0.997 \pm 0.001$* (=) | **$1.000 \pm 0.000$** (=) |
| Ada-GVAE-k$^\dagger$ | $0.133 \pm 0.088$ | $0.199 \pm 0.058$ | $0.563 \pm 0.209$ | $0.809 \pm 0.164$ | *$0.997 \pm 0.001$* (=) | **$1.000 \pm 0.000$** (=) |
| GVAE | $0.417 \pm 0.056$ | $0.593 \pm 0.079$ | $0.687 \pm 0.052$ | *$0.893 \pm 0.024$* | *$0.997 \pm 0.000$* (=) | **$1.000 \pm 0.000$** (=) |
| GVAE$^\dagger$ | $0.130 \pm 0.047$ | $0.234 \pm 0.038$ | $0.478 \pm 0.076$ | $0.870 \pm 0.029$ | **$0.998 \pm 0.000$** | **$1.000 \pm 0.000$** (=) |
| MLVAE | $0.370 \pm 0.041$ | $0.515 \pm 0.093$ | $0.590 \pm 0.080$ | $0.854 \pm 0.059$ | $0.996 \pm 0.000$ | **$1.000 \pm 0.000$** (=) |
| MLVAE$^\dagger$ | $0.123 \pm 0.048$ | $0.234 \pm 0.069$ | $0.412 \pm 0.035$ | $0.807 \pm 0.141$ | *$0.997 \pm 0.000$* (=) | **$1.000 \pm 0.000$** (=) |
| Shu | $0.062 \pm 0.086$ | $0.117 \pm 0.075$ | $0.146 \pm 0.073$ | $0.157 \pm 0.078$ | $0.245 \pm 0.096$ | $0.427 \pm 0.082$ |
| Shu$^\dagger$ | $-0.029 \pm 0.076$ | $0.093 \pm 0.08$ | $0.118 \pm 0.077$ | $0.131 \pm 0.079$ | $0.228 \pm 0.085$ | $0.419 \pm 0.082$ |
| Symbolic vector-tokened compositional representations | | | | | | |
| VCT | $-0.201 \pm 0.289$ | $0.202 \pm 0.136$ | $0.359 \pm 0.093$ | $0.564 \pm 0.075$ | $0.844 \pm 0.012$ | $0.954 \pm 0.008$ |
| VCT$^\dagger$ | $-4.183 \pm 1.679$ | $-1.464 \pm 0.492$ | $-0.552 \pm 0.307$ | $-0.205 \pm 0.134$ | $0.396 \pm 0.063$ | $0.843 \pm 0.040$ |
| COMET | $0.352 \pm 0.036$ | $0.699 \pm 0.038$ | $0.772 \pm 0.029$ | $0.812 \pm 0.025$ | $0.996 \pm 0.001$ | **$1.000 \pm 0.000$** (=) |
| COMET$^\dagger$ | **$0.751 \pm 0.039$** | **$0.870 \pm 0.029$** | **$0.996 \pm 0.001$** | **$0.915 \pm 0.031$** | $0.996 \pm 0.001$ | **$1.000 \pm 0.000$** (=) |
| Fully continuous compositional representations | | | | | | |
| Ours | *$0.464 \pm 0.071$* | *$0.730 \pm 0.038$* | *$0.804 \pm 0.075$* | $0.889 \pm 0.035$ | $0.995 \pm 0.002$ | **$1.000 \pm 0.000$** (=) |

Table 39: Downstream regression model performance on the MPI3D dataset

| Models | $R^2$ score | | | | | |
|---|---|---|---|---|---|---|
| | $10^2$ samples | $2.5 \times 10^2$ samples | $5 \times 10^3$ samples | $10^4$ samples | $10^5$ samples | all samples |
| Symbolic scalar-tokened compositional representations | | | | | | |
| Slow-VAE | $0.127 \pm 0.059$ | $0.152 \pm 0.011$ | $0.596 \pm 0.083$ | $0.678 \pm 0.082$ | $\mathbf{0.863 \pm 0.023}$ | $0.980 \pm 0.005$ |
| Slow-VAE$^\dagger$ | $0.105 \pm 0.026$ | $0.093 \pm 0.011$ | $0.531 \pm 0.077$ | $0.655 \pm 0.113$ | $0.860 \pm 0.016$ | $\mathbf{0.981 \pm 0.004}$ |
| Ada-GVAE-k | $0.206 \pm 0.031$ | $0.213 \pm 0.023$ | $0.381 \pm 0.014$ | $0.412 \pm 0.015$ | $0.536 \pm 0.013$ | $0.762 \pm 0.014$ |
| Ada-GVAE-k$^\dagger$ | $0.037 \pm 0.045$ | $0.093 \pm 0.018$ | $0.342 \pm 0.014$ | $0.390 \pm 0.012$ | $0.527 \pm 0.010$ | $0.774 \pm 0.003$ |
| GVAE | $0.181 \pm 0.030$ | $0.217 \pm 0.023$ | $0.371 \pm 0.026$ | $0.404 \pm 0.028$ | $0.530 \pm 0.035$ | $0.771 \pm 0.018$ |
| GVAE$^\dagger$ | $0.056 \pm 0.057$ | $0.120 \pm 0.029$ | $0.316 \pm 0.027$ | $0.338 \pm 0.019$ | $0.495 \pm 0.020$ | $0.764 \pm 0.019$ |
| MLVAE | $0.182 \pm 0.013$ | $0.222 \pm 0.024$ | $0.357 \pm 0.042$ | $0.384 \pm 0.023$ | $0.513 \pm 0.025$ | $0.773 \pm 0.031$ |
| MLVAE$^\dagger$ | $0.051 \pm 0.032$ | $0.089 \pm 0.037$ | $0.305 \pm 0.018$ | $0.370 \pm 0.023$ | $0.520 \pm 0.033$ | $0.790 \pm 0.027$ |
| Shu | $0.151 \pm 0.016$ | $0.211 \pm 0.026$ | $0.238 \pm 0.019$ | $0.264 \pm 0.018$ | $0.330 \pm 0.033$ | $0.441 \pm 0.041$ |
| Shu$^\dagger$ | $0.057 \pm 0.048$ | $0.186 \pm 0.012$ | $0.214 \pm 0.017$ | $0.237 \pm 0.013$ | $0.311 \pm 0.020$ | $0.438 \pm 0.037$ |
| Symbolic vector-tokened compositional representations | | | | | | |
| VCT | $0.047 \pm 0.139$ | $0.110 \pm 0.087$ | $0.124 \pm 0.104$ | $0.105 \pm 0.175$ | $0.164 \pm 0.183$ | $0.455 \pm 0.071$ |
| VCT$^\dagger$ | $-0.005 \pm 0.058$ | $0.084 \pm 0.062$ | $0.106 \pm 0.092$ | $0.153 \pm 0.072$ | $0.276 \pm 0.036$ | $0.550 \pm 0.046$ |
| COMET | $-0.051 \pm 0.015$ | $-0.042 \pm 0.018$ | $-0.037 \pm 0.012$ | $-0.053 \pm 0.024$ | $0.218 \pm 0.036$ | $0.266 \pm 0.004$ |
| COMET$^\dagger$ | $-0.118 \pm 0.045$ | $-0.089 \pm 0.050$ | $-0.074 \pm 0.039$ | $-0.179 \pm 0.098$ | $0.027 \pm 0.099$ | $0.236 \pm 0.008$ |
| Fully continuous compositional representations | | | | | | |
| Ours | $\mathbf{0.490 \pm 0.068}$ | $\mathbf{0.594 \pm 0.056}$ | $\mathbf{0.635 \pm 0.070}$ | $\mathbf{0.693 \pm 0.060}$ | $0.757 \pm 0.03$ | $0.882 \pm 0.016$ |

## b) Abstract Visual Reasoning

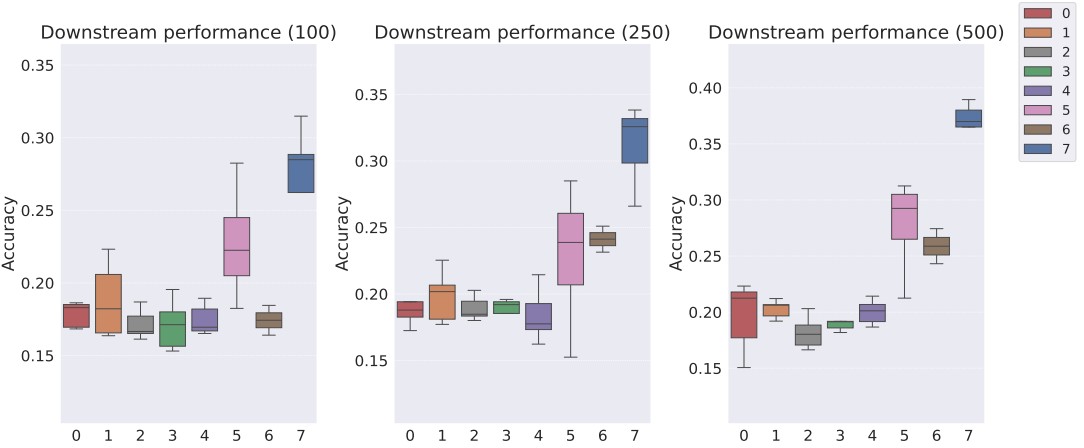

Figure 47: Downstream WReN model classification accuracy on the abstract visual reasoning dataset (original setting).

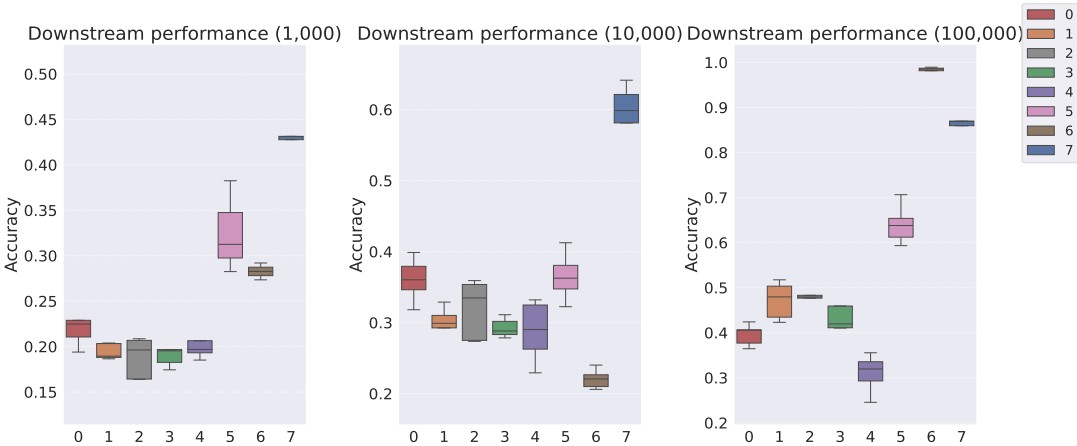

Figure 48: Downstream WReN model classification accuracy on the abstract visual reasoning dataset (original setting).

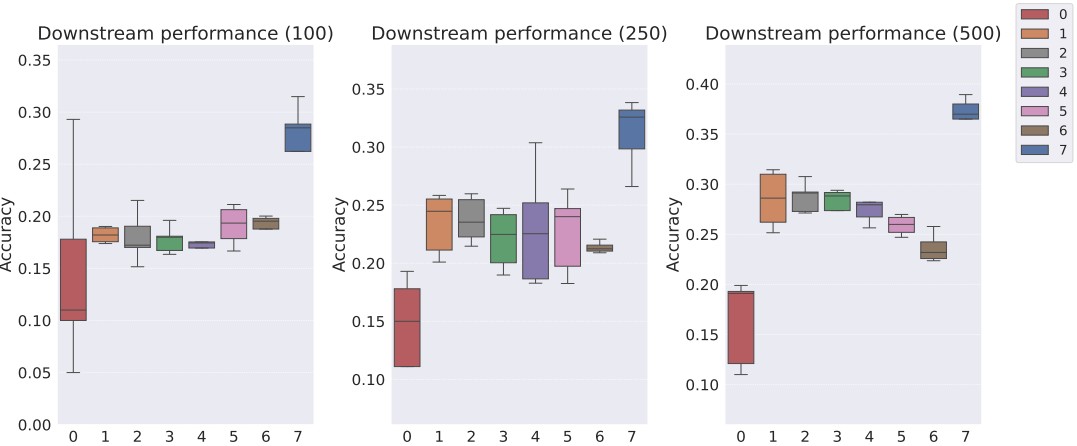

Figure 49: Downstream WReN model classification accuracy on the abstract visual reasoning dataset (dimensionality-controlled setting).

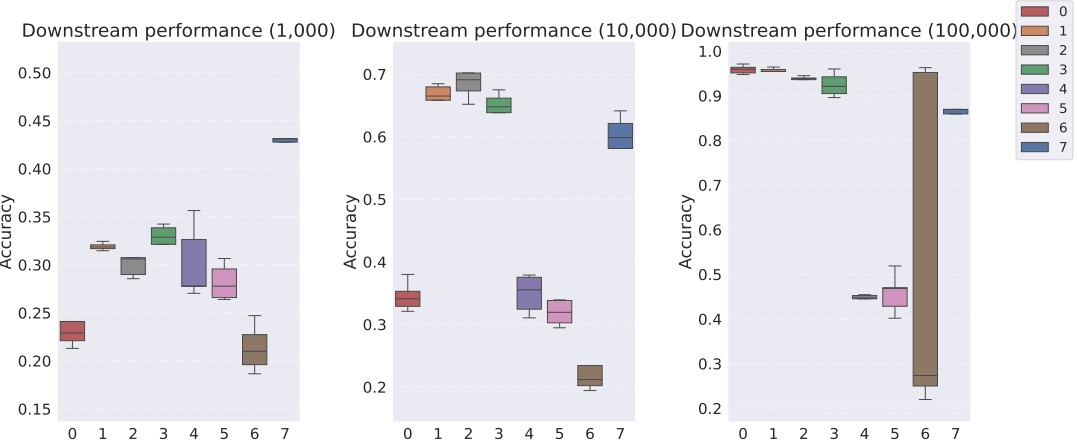

Figure 50: Downstream WReN model classification accuracy on the abstract visual reasoning dataset (dimensionality-controlled setting).

Table 40: Downstream WReN model performance on the abstract visual reasoning dataset

| Models | Classification accuracy | | | | | |
|---|---|---|---|---|---|---|
| | $10^2$ samples | $2.5 \times 10^2$ samples | $5 \times 10^2$ samples | $10^3$ samples | $10^4$ samples | $10^5$ samples |
| Symbolic scalar-tokened compositional representations | | | | | | |
| Slow-VAE | $0.178 \pm 0.008$ | $0.196 \pm 0.024$ | $0.196 \pm 0.028$ | $0.228 \pm 0.030$ | $0.361 \pm 0.028$ | $0.396 \pm 0.022$ |
| Slow-VAE[†] | $0.146 \pm 0.084$ | $0.149 \pm 0.034$ | $0.163 \pm 0.039$ | $0.237 \pm 0.023$ | $0.345 \pm 0.021$ | *0.959 ± 0.009* |
| Ada-GVAE-k | $0.188 \pm 0.023$ | $0.198 \pm 0.018$ | $0.203 \pm 0.007$ | $0.194 \pm 0.008$ | $0.295 \pm 0.028$ | $0.472 \pm 0.037$ |
| Ada-GVAE-k[†] | $0.182 \pm 0.007$ | *0.234 ± 0.024* | *0.285 ± 0.025* | $0.319 \pm 0.004$ | *0.662 ± 0.023* | $0.954 \pm 0.009$ |
| GVAE | $0.171 \pm 0.009$ | $0.189 \pm 0.008$ | $0.182 \pm 0.013$ | $0.188 \pm 0.020$ | $0.319 \pm 0.038$ | $0.478 \pm 0.032$ |
| GVAE[†] | $0.180 \pm 0.022$ | $0.237 \pm 0.018$ | $0.287 \pm 0.014$ | $0.306 \pm 0.020$ | **0.684 ± 0.020** | $0.936 \pm 0.008$ |
| MLVAE | $0.171 \pm 0.016$ | $0.188 \pm 0.009$ | $0.193 \pm 0.012$ | $0.194 \pm 0.015$ | $0.293 \pm 0.012$ | $0.432 \pm 0.023$ |
| MLVAE[†] | $0.177 \pm 0.012$ | $0.221 \pm 0.023$ | $0.277 \pm 0.023$ | $0.325 \pm 0.017$ | $0.644 \pm 0.026$ | $0.925 \pm 0.024$ |
| Shu | $0.175 \pm 0.009$ | $0.184 \pm 0.018$ | $0.200 \pm 0.010$ | $0.202 \pm 0.014$ | $0.288 \pm 0.038$ | $0.310 \pm 0.038$ |
| Shu[†] | $0.177 \pm 0.014$ | $0.230 \pm 0.045$ | $0.285 \pm 0.028$ | $0.302 \pm 0.034$ | $0.444 \pm 0.013$ | $0.444 \pm 0.013$ |
| Symbolic vector-tokened compositional representations | | | | | | |
| VCT | *0.228 ± 0.036* | $0.229 \pm 0.049$ | $0.277 \pm 0.039$ | *0.326 ± 0.042* | $0.371 \pm 0.040$ | $0.641 \pm 0.039$ |
| VCT[†] | $0.191 \pm 0.017$ | $0.226 \pm 0.031$ | $0.259 \pm 0.009$ | $0.282 \pm 0.017$ | $0.319 \pm 0.018$ | $0.458 \pm 0.040$ |
| COMET | $0.174 \pm 0.010$ | $0.241 \pm 0.010$ | $0.259 \pm 0.016$ | $0.283 \pm 0.009$ | $0.221 \pm 0.012$ | **0.983 ± 0.006** |
| COMET[†] | $0.190 \pm 0.012$ | $0.214 \pm 0.004$ | $0.236 \pm 0.013$ | $0.213 \pm 0.023$ | $0.532 \pm 0.348$ | $0.532 \pm 0.348$ |
| Fully continuous compositional representations | | | | | | |
| Ours | **0.273 ± 0.033** | **0.312 ± 0.027** | **0.360 ± 0.033** | **0.412 ± 0.066** | $0.560 \pm 0.103$ | $0.869 \pm 0.024$ |

## C.6 Ablation Experiments

We perform the following ablation studies:

- **Foundational Model Components**: We evaluate the impact of *Relaxed Representational Constraints* (RRC) and *Distributed Encoding* (DE) on disentanglement performance, as presented in Table 43. Previous work [30, 33] has established a positive correlation between disentanglement and both downstream sample efficiency and overall performance. Therefore, we do not separately assess the effects of these foundational components on downstream tasks.

- **Auxiliary Model Components**: We investigate the influence of additional model components, as detailed in Appendix C.6.2.

- **Effect of Soft TPR on Downstream Tasks**: Separately, we examine whether Soft TPR offers advantages over its quantised, explicit counterpart, $\psi_{tpr}^*$, in terms of downstream model performance.

### C.6.1 Distributed Encoding Ablation

To perform an ablation study on distributed encoding, we set the role embedding matrix $M_{\xi_R}$ to the identity matrix. This modification results in maximally sparse role-filler bindings of the form $\xi_F(f_{m(i)}) \otimes e_i$, specifically:

$$\xi_F(f_{m(i)}) \otimes \xi_R(r_i) = \begin{bmatrix} 0 & \cdots & 0 & \xi_F(f_{m(i)}) & 0 & \cdots & 0 \end{bmatrix}$$

In this setup, the embedded filler $\xi_F(f_{m(i)})$ occupies the $i$-th column of the resulting matrix. Consequently, the TPRs generated by the TPR decoder become degenerate, effectively forming a concatenation of filler embeddings:

$$\psi_{tpr}(x) \cong \left( \xi_F(f_{m(1)})^\top, \ldots, \xi_F(f_{m(N_R)})^\top \right)^\top$$

This configuration confines each filler's information to specific dimensions within the representation, thereby limiting the model's ability to leverage distributed information across multiple dimensions.

Our experiments reveal moderate reductions in disentanglement performance and increases in loss compared to the distributed encoding scenario. Considering the rotational isomorphism between an identity matrix $M_{\xi_R}$ and a (semi-)orthogonal $M_{\xi_R}$, it is somewhat surprising to observe these differences under identical conditions (e.g., initialisation and random seeds). While future work

*should* aim to learn linearly independent (but not necessarily orthogonal) $M_{\xi_R}$, which would more convincingly account for such results by embedding each filler in *overlapping* role subspaces and eliminating the rotational isomorphism with the degenerate TPR case, this behaviour warrants further analysis and investigation. One potential avenue of investigation is examining whether the sparsity of binding embeddings $\xi_F(f_{m(i)}) \otimes \xi_R(r_i)$ affects 'neuron' specialisation (i.e., the tendency of each row of the weight matrix to attend only to certain filler-specific information).

Table 41: Effect of DE on disentanglement (MPI3D dataset).

| Property | FactorVAE score | DCI score | MIG score | BetaVAE score |
|---|---|---|---|---|
| - DE | $0.901 \pm 0.102$ | $0.704 \pm 0.135$ | $0.530 \pm 0.087$ | $0.991 \pm 0.021$ |
| + DE (Full) | $0.949 \pm 0.032$ | $0.828 \pm 0.015$ | $0.620 \pm 0.067$ | $1.000 \pm 0.000$ |

### C.6.2 Additional Ablations

We additionally examine the importance of the following properties of our model in producing explicitly compositional Soft TPR representations: 1) the presence of weak supervision, by setting $\lambda_1 = \lambda_2 = 0$ in Equation 7, 2) the explicit dependency between the quantised filler embeddings and the decoder output, by instead using the Soft TPR to reconstruct the input image, and 3) the semi-orthogonality of the role embedding matrix, $M_{\xi_R}$ by removing this constraint in the random initialisation of $M_{\xi_R}$, with the results of these ablations illustrated in Table 42.

Table 42: Effect of model properties on disentanglement performance (MPI3D dataset).

| Property | DCI Score |
|---|---|
| - Weak supervision | $0.225 \pm 0.034$ |
| - Explicit filler dependency | $0.718 \pm 0.051$ |
| - Semi orthogonality | $0.756 \pm 0.039$ |
| Full | $0.828 \pm 0.015$ |

### C.6.3 Soft TPR vs TPR

To examine the effect of the Soft TPR on downstream tasks, for each fully trained Soft TPR representation learner, we extract both the Soft TPR, $z$, and the Soft TPR's corresponding quantised TPR, $\psi_{tpr}^*$, and investigate downstream performance using both types of representation. In all plots, we use the same legend, denoting the explicit TPR as yellow (0) and the Soft TPR as blue (1). Across all considered cases: i.e., 1) convergence rate of representation learning (as measured by the downstream model's ability to effectively leverage representations produced at different stages of training), 2) sample efficiency of downstream models, and 3) raw performance of downstream models in the low sample regime, the Soft TPR confers differential performance boosts compared to the explicit TPR.

**Convergence Rate of Representation Learning**

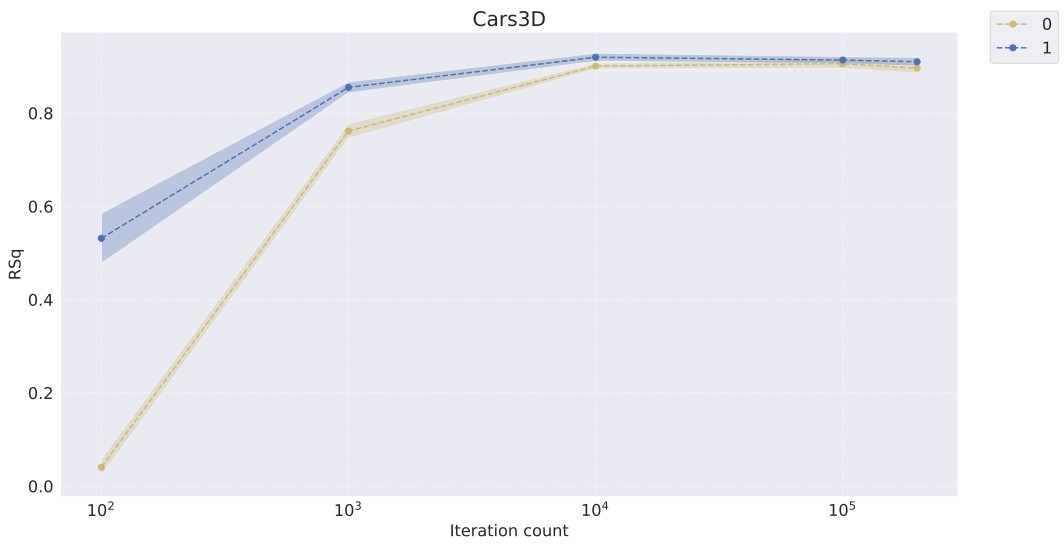

Figure 51: Convergence of Soft TPR (0) vs TPR (1) as measured by FoV regression on the Cars3D dataset

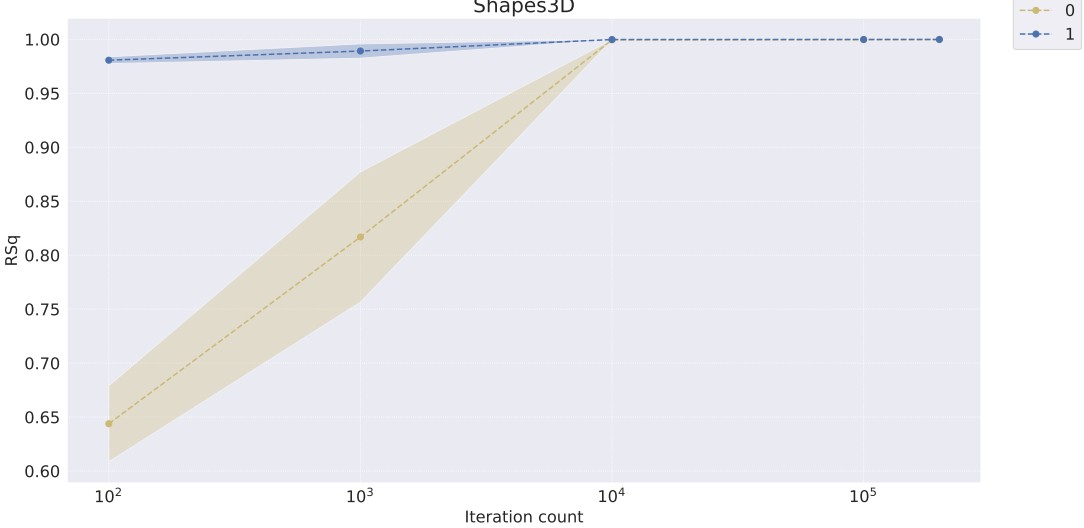

Figure 52: Convergence of Soft TPR (0) vs TPR (1) as measured by FoV regression on the Shapes3D dataset

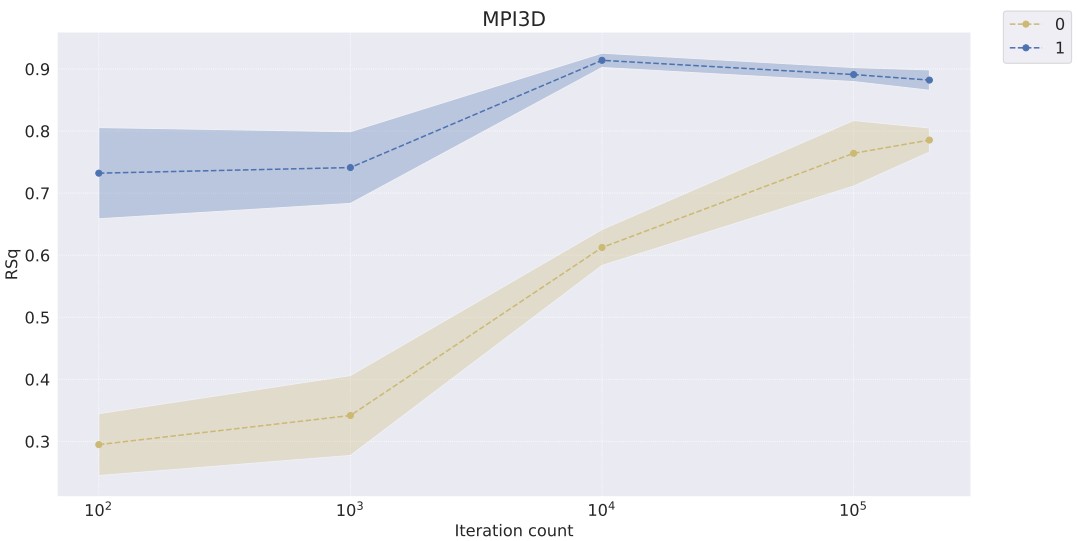

Figure 53: Convergence of Soft TPR (0) vs TPR (1) as measured by FoV regression on the MPI3D dataset

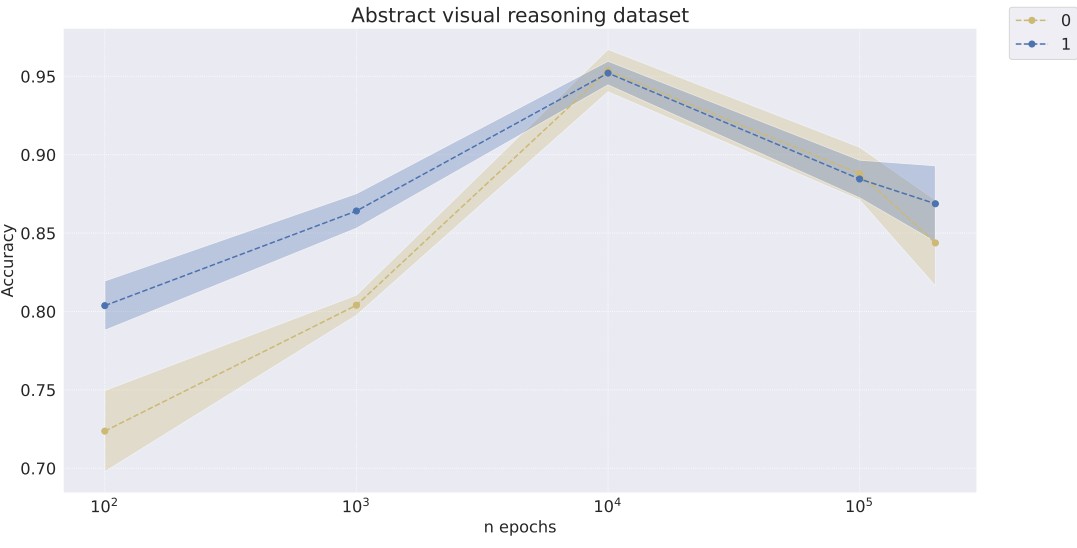

Figure 54: Convergence of Soft TPR (0) vs TPR (1) as measured by classification performance on the abstract visual reasoning dataset

**Sample Efficiency of Downstream Models**

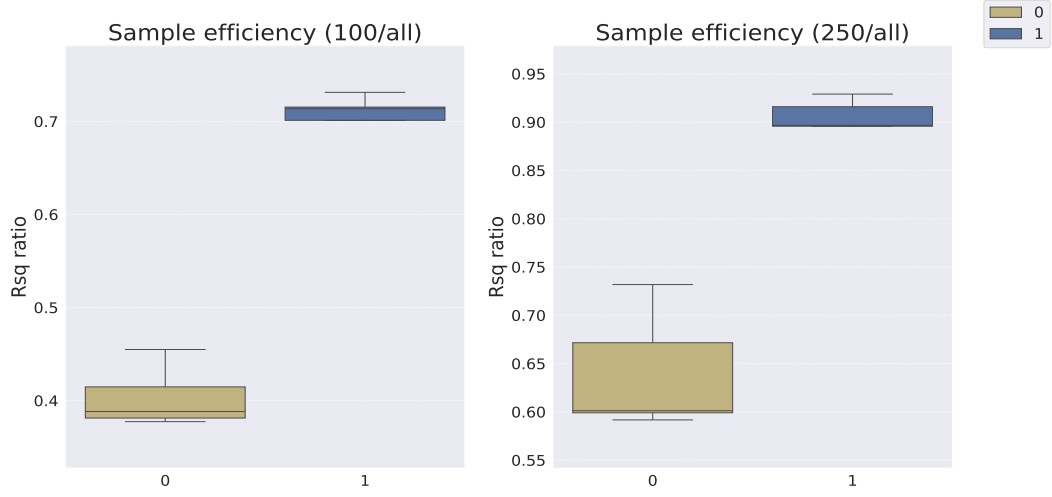

Figure 55: Downstream regression model sample efficiency on the Cars3D dataset using TPR representations (0) vs Soft TPR representations (1)

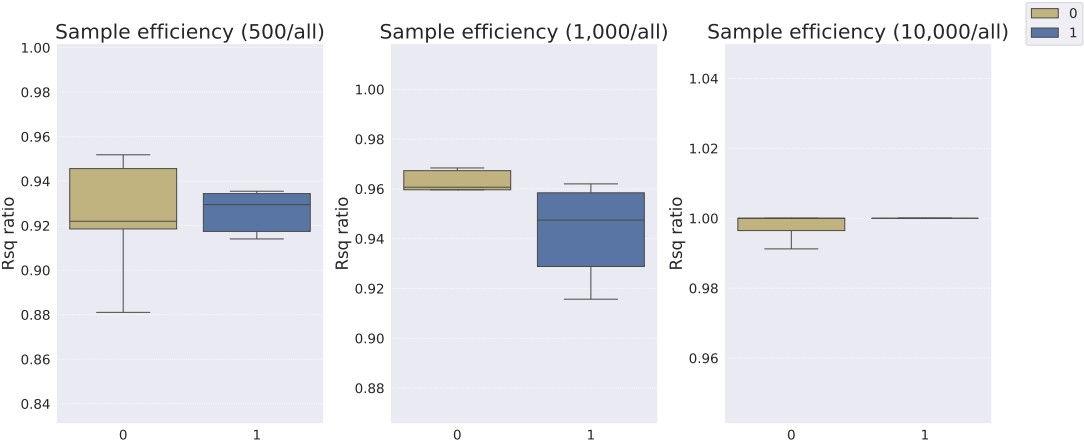

Figure 56: Downstream regression model sample efficiency on the Cars3D dataset using TPR representations (0) vs Soft TPR representations (1)

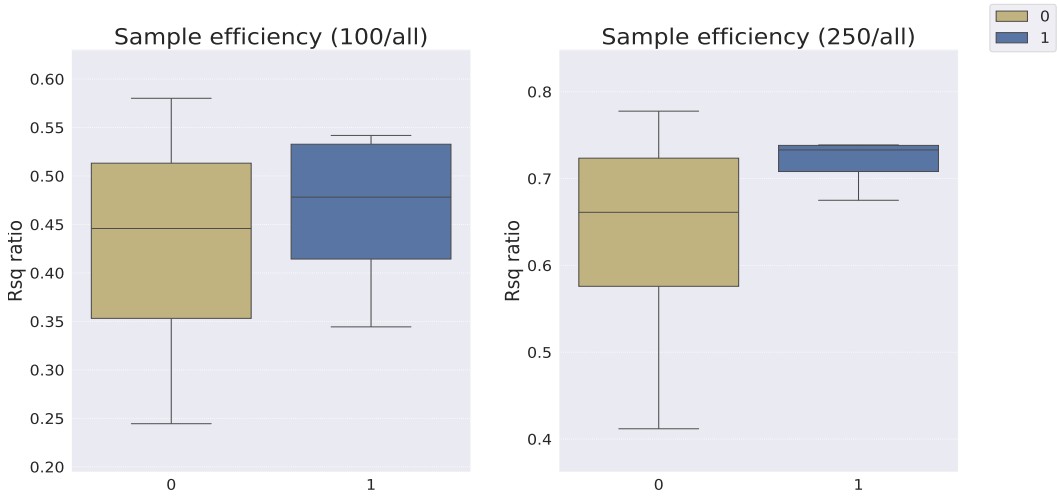

Figure 57: Downstream regression model sample efficiency on the Shapes3D dataset using TPR representations (0) vs Soft TPR representations (1)

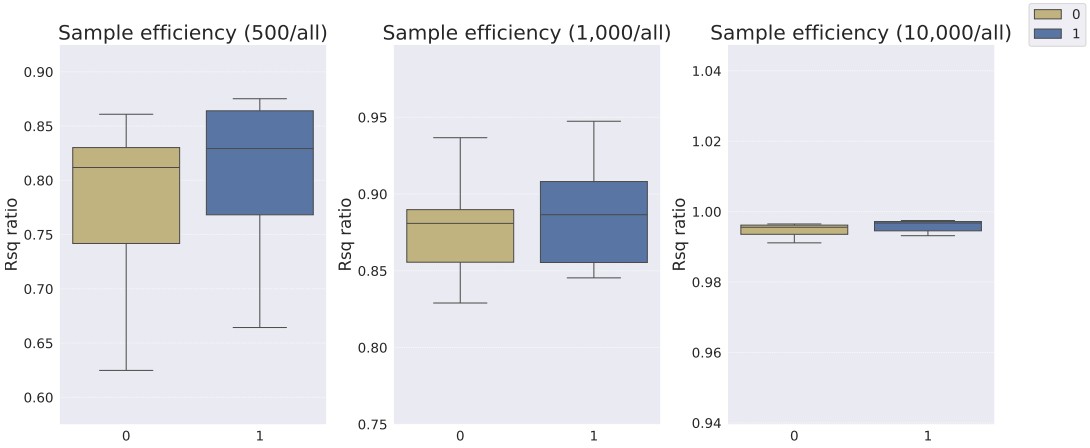

Figure 58: Downstream regression model sample efficiency on the Shapes3D dataset using TPR representations (0) vs Soft TPR representations (1)

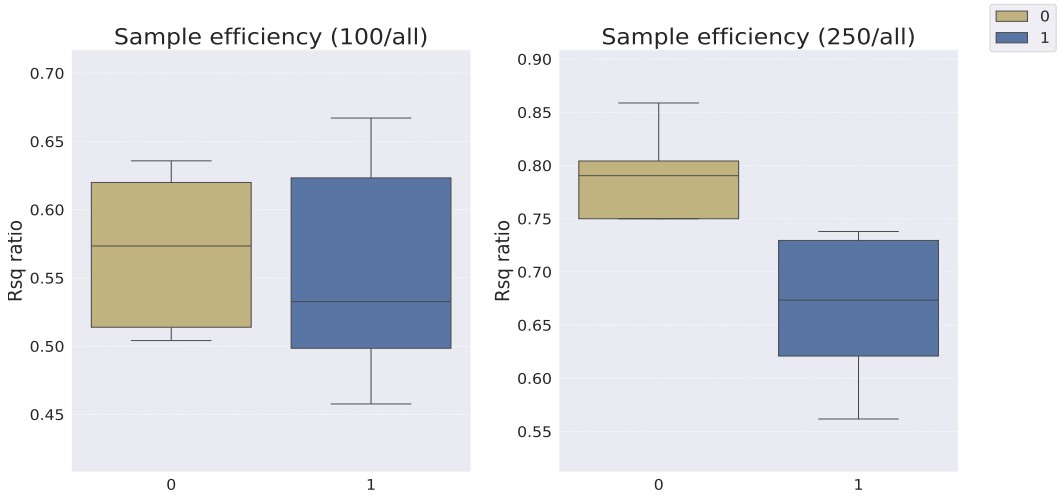

Figure 59: Downstream regression model sample efficiency on the MPI3D dataset using TPR representations (0) vs Soft TPR representations (1)

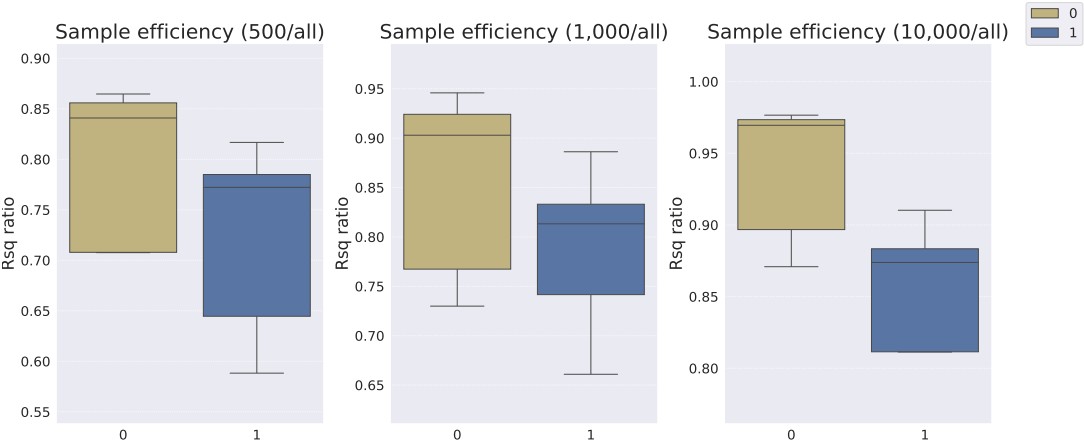

Figure 60: Downstream regression model sample efficiency on the MPI3D dataset using TPR representations (0) vs Soft TPR representations (1)

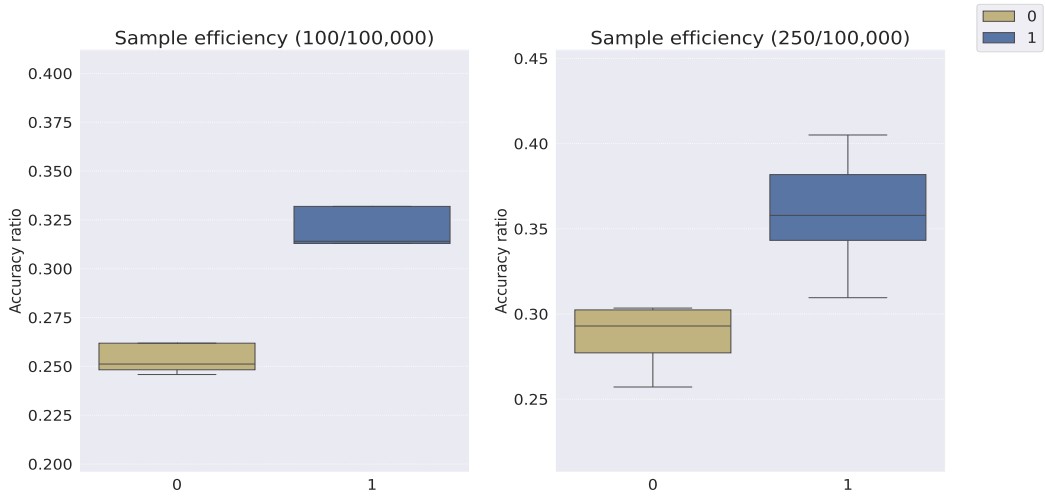

Figure 61: Downstream WReN model sample efficiency on the abstract visual reasoning dataset using TPR representations (0) vs Soft TPR representations (1)

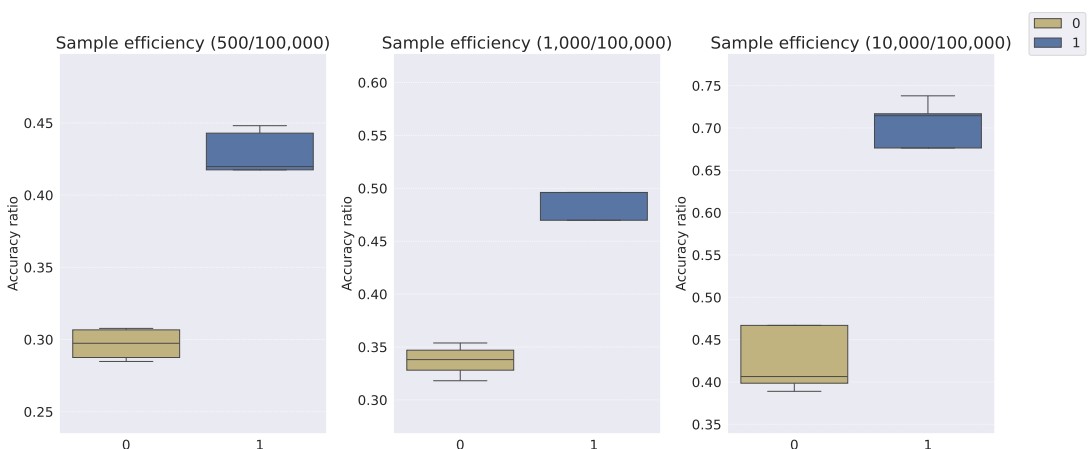

Figure 62: Downstream WReN model sample efficiency on the abstract visual reasoning dataset using TPR representations (0) vs Soft TPR representations (1)

**Low Sample-Regime Performance of Downstream Models**

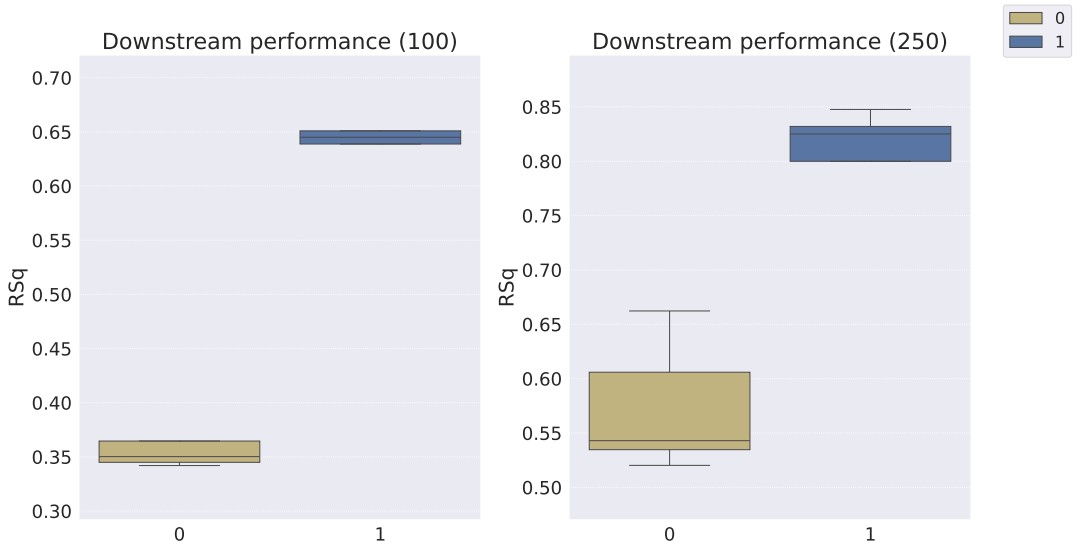

Figure 63: Downstream regression model $R^2$ scores on the Cars3D dataset using TPR representations (0) vs Soft TPR representations (1)

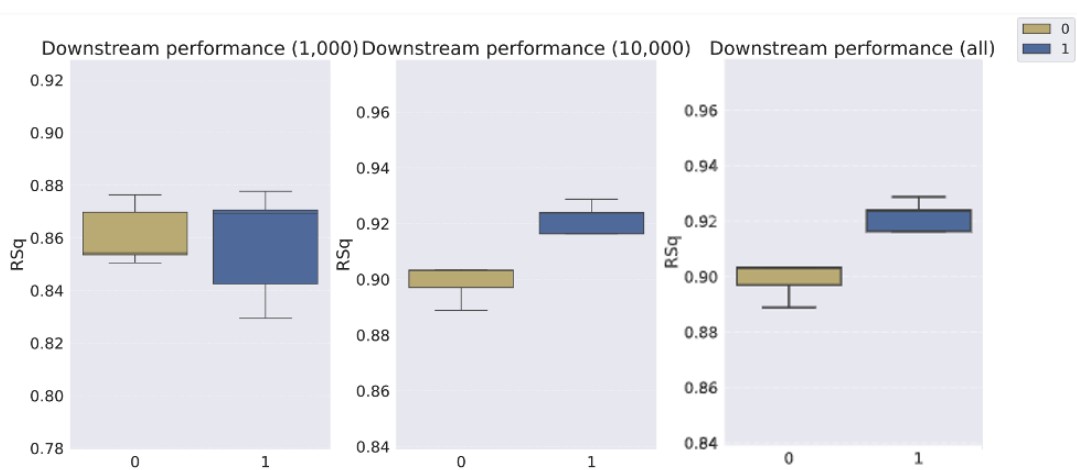

Figure 64: Downstream regression model $R^2$ scores on the Cars3D dataset using TPR representations (0) vs Soft TPR representations (1)

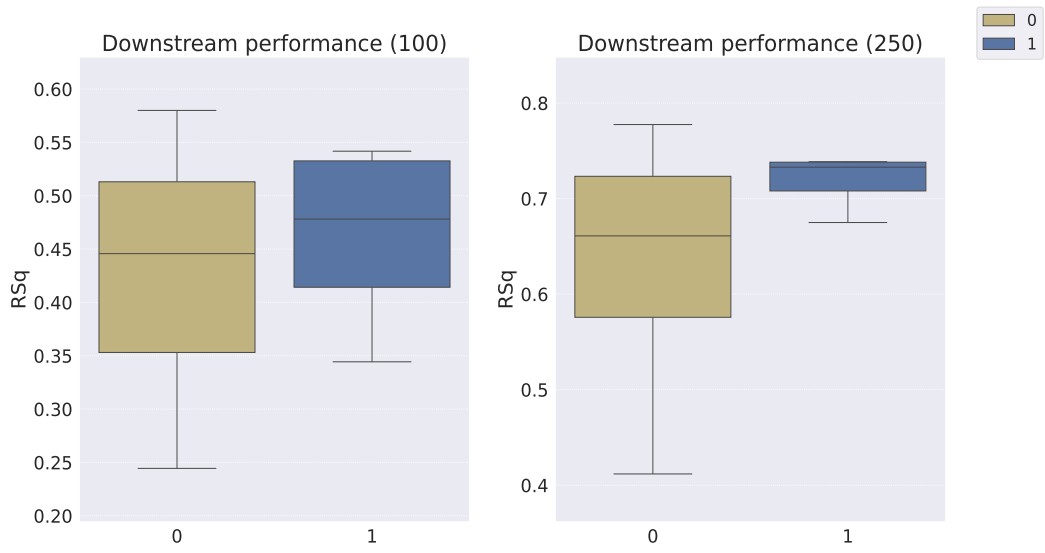

Figure 65: Downstream regression model $R^2$ scores on the Shapes3D dataset using TPR representations (0) vs Soft TPR representations (1)

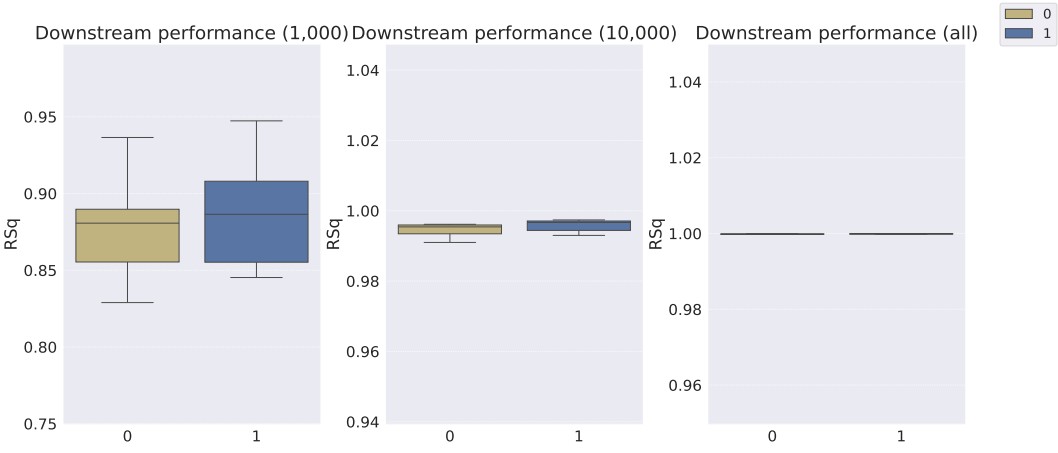

Figure 66: Downstream regression model $R^2$ scores on the Shapes3D dataset using TPR representations (0) vs Soft TPR representations (1)

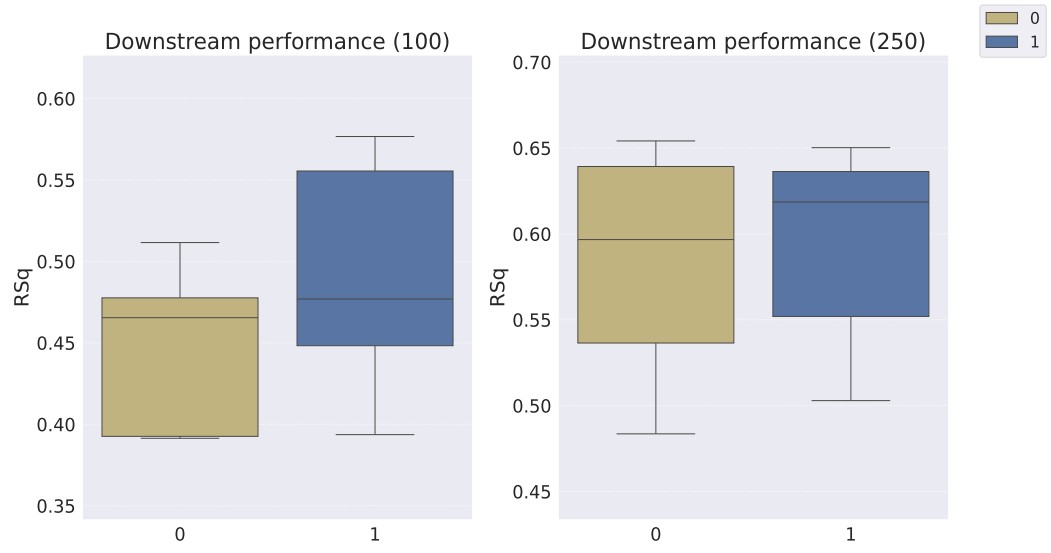

Figure 67: Downstream regression model $R^2$ scores on the MPI3D dataset using TPR representations (0) vs Soft TPR representations (1)

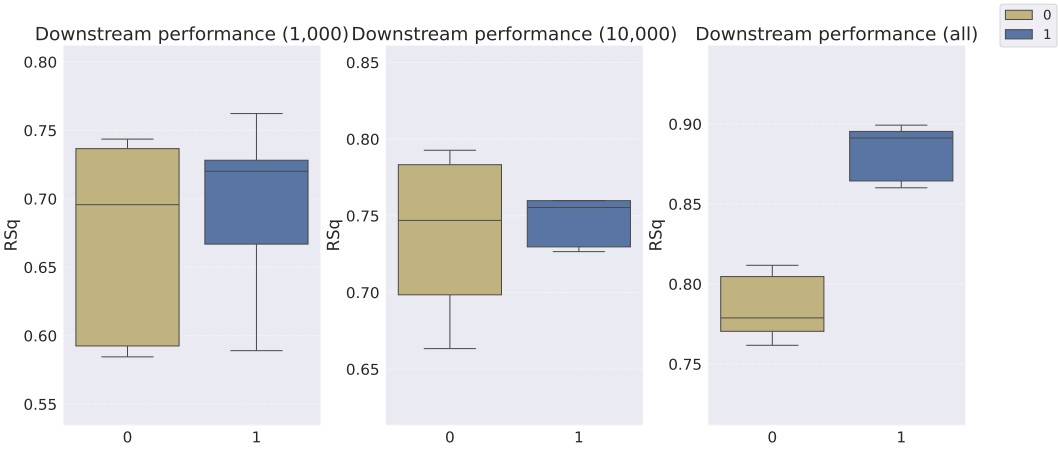

Figure 68: Downstream regression model $R^2$ scores on the MPI3D dataset using TPR representations (0) vs Soft TPR representations (1)

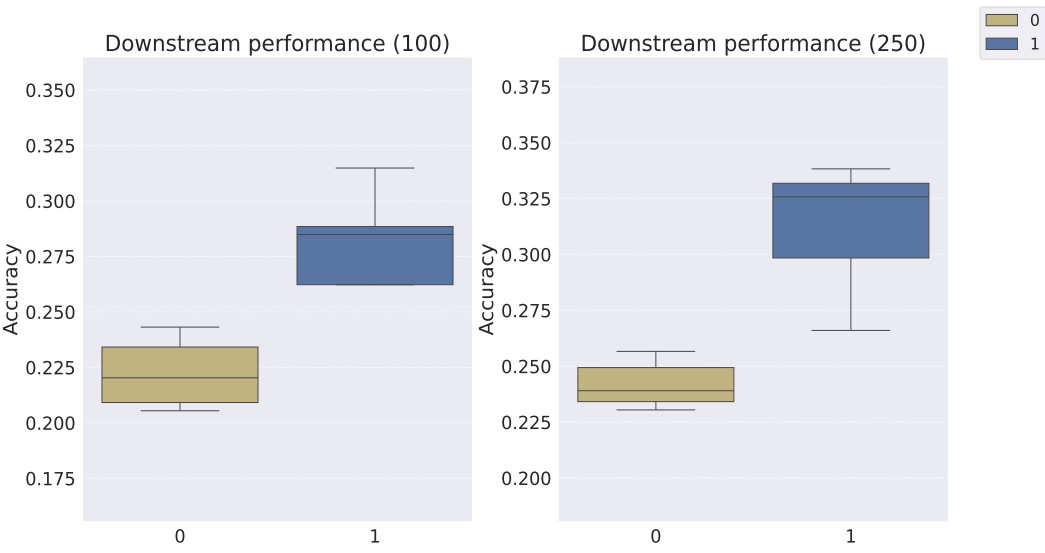

Figure 69: Downstream WReN model classification accuracy on the abstract visual reasoning dataset using TPR representations (0) vs Soft TPR representations (1)

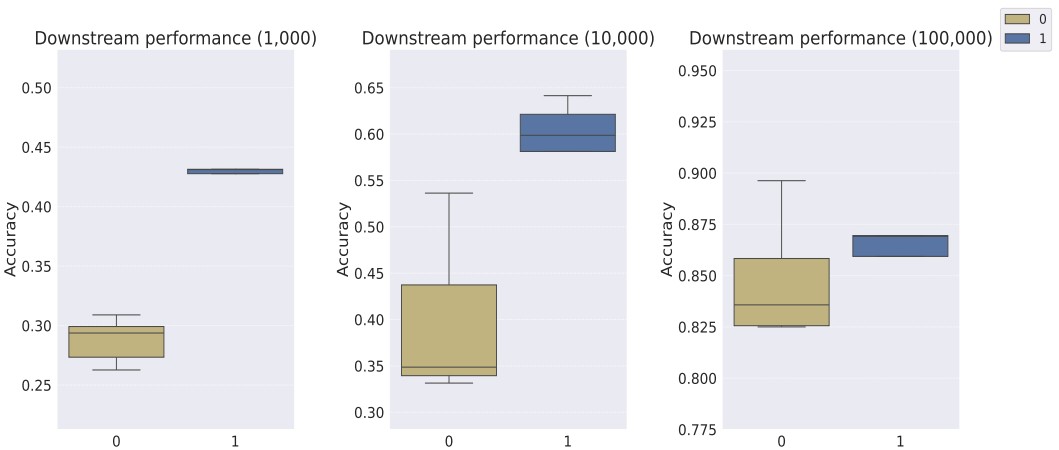

Figure 70: Downstream WReN model classification accuracy on the abstract visual reasoning dataset using TPR representations (0) vs Soft TPR representations (1)

### C.6.4 Robustness to Hyperparameter Choices

We perform an additional experiment to empiricially verify that our model is robust to different hyperparameter choices. For the MPI3D dataset, which disentanglement models experience the greatest difficulty in producing disentangled representations for (see Table 1), we randomly pick another set of hyperparameters, shown in Table 43, from the top 5 models based on the MSE loss criterion. For this randomly chosen hyperparameter configuration, we evaluate the disentanglement of the representations on the MPI3D dataset produced by the resulting model.

Table 43: Hyperparameter values of ablation setting

| Hyperparameter | MPI3D |
|---|---|
| | Architectural hyperparameters |
| $D_R$ | 16 |
| $N_R$ *(fixed)* | 10 |
| $D_F$ | 64 |
| $N_F$ | 50 |
| | Loss function hyperparameters |
| $\lambda_1$ | 0.0000 |
| $\lambda_2$ | 0.25378 |
| $\beta$ *(fixed)* | 0.5 |

Table 44: Disentanglement metric scores on the MPI3D dataset

| Hyperparameter configuration | FactorVAE score | DCI score | BetaVAE score | MIG score |
|---|---|---|---|---|
| Original | $0.949 \pm 0.032$ | $0.828 \pm 0.015$ | $1.000 \pm 0.000$ | $0.620 \pm 0.067$ |
| Ablation | $0.911 \pm 0.029$ | $0.798 \pm 0.031$ | $1.000 \pm 0.000$ | $0.590 \pm 0.047$ |

# D  Limitations and Future Work

## D.1  Extension to Linguistic Domains

Applying the Soft TPR to the TPR's typical domain of language is an intriguing future direction, especially as language can deviate from strict algebraic compositionality – for instance, idiomatic expressions such as 'spill the beans' cannot be understood as a function of their constituents alone. Soft TPR's more flexible specification allows it to capture approximate forms of compositionality precluded from the TPR's strict algebraic definition (Eq 1), thereby potentially providing the Soft TPR the ability to better handle the nuance and complexity of language.

To adapt our framework to language, we replace our Conv/Deconv encoder/decoders with simple RNNs, retrain our TPR decoder, and remove the semi-supervised loss, using Eq 6 as the full loss. Preliminary results in Table 45 on the BaBI [7] dataset are compared with TPR baselines from AID [51]. Our Soft TPR Autoencoder does not presently surpass AID, but notable points include:

1. Our use of simpler RRN-based encoders and an MLP-based downstream network, unlike the more sophisticated architectures of [51].

2. Soft TPR retains its performance improvement above the corresponding explicit TPR it can be quantised into.

3. The smaller gap between systematic vs non-systematic dataset splits in our model compared to TPR-RNN (+AID) and FWM.

4. We train our representation learner using self-supervision (reconstruction loss) alone, only employing supervision on the downstream prediction network, while the baselines employ strong supervision and end-to-end training to produce representations.

Table 45: Mean word error rate [%] on the sys-bAbI task [51]

| Model | w/o sys diff | w/ sys diff | Gap |
|---|---|---|---|
| TPR-RNN | $0.79 \pm 0.16$ | $8.74 \pm 3.74$ | 7.95 |
| TPR-RNN + AID | *$0.69 \pm 0.08$* | $5.61 \pm 1.78$ | 4.92 |
| FWM | $0.79 \pm 0.14$ | *$2.85 \pm 1.61$* | 2.06 |
| FWM + AID | **$0.45 \pm 0.16$** | **$1.21 \pm 0.66$** | **0.76** |
| Ours (TPR) | $4.54 \pm 0.61$ | $6.51 \pm 1.23$ | 1.97 |
| Ours (Soft TPR) | $2.79 \pm 0.24$ | $3.64 \pm 0.69$ | *0.85* |

## D.2 Need for Weak Supervision

To produce a compositional representation, $\psi(x) = C(\psi_1(a_1), \ldots, \psi_n(a_n))$ (as in Section 3.1), each representational constituent, $\psi_i(a_i)$, must map 1-1 to a data constituent, $a_i$. Without supervision (i.e., access only to observational data, $\{x_i\}$), this is challenging, because the data constituents $\{a_i\}$ underlying each object, $x$, are unknown and cannot be identified.

We can frame the above intuition in a more mathematically rigorous way, using the generative framework. Essentially, as formally proved in [25], it is impossible to identify the true distribution for the data constituents (generative factors), $p(a)$, using observational data, $\{x_i\}$, alone as there are infinitely many bijective functions $f : \text{supp}(a) \rightarrow a$ such that: 1) $a$ and $f(a)$ are completely entangled (i.e. non-diagonal Jacobian) *and* 2) the marginal distributions of $a$ and $f(a)$ are identical (meaning the marginal distributions of the observations are also identical, i.e., $\int p(x|a)p(a)da = \int p(x|f(a))p(f(a))da$). Thus, without inductive biases, it is impossible to infer the data constituents $\{a_i\}$ of any observation, $x$, from observational data $\{x_i\}$ alone.

To combat this non-identifiability result, in line with [13, 22, 31, 33, 35], we use weak supervision, presenting the model with data pairs $(x, x')$ where $x$ and $x'$ differ in a subset of FoVs, e.g. $\beta(x) = \{\text{shape/cube}, \text{colour/}purple, \text{size/large}\}$, $\beta(x') = \{\text{shape/cube}, \text{colour/}cyan, \text{size/large}\}$, where the FoV *types* corresponding to the different FoV values are *known* to the model. Note that this weak supervision is minimal, only providing the model to access to the differing FoV *types* in an index set, $I$, (i.e., $I := \{\text{colour}\}$) and not *any* of the FoV values (i.e., $\{\text{cube}, \text{purple}, \text{cyan}, \text{large}\}$ are all not known by the model).

Some possible future extensions to reduce this level of weak supervision, or alternative forms of weak supervision include:

1. **Embodied Learning**: In the visual domain, some roles, e.g. object position correspond to affordances. Embodied agents may be able to reduce the need for explicit supervision by collecting $(x, x')$ and $I$ through interaction with their environment.

2. **Pretrained Filler Embeddings**: Initialising the filler embedding matrix, $M_{\xi_F}$ with embeddings learnt by a pre-trained vision model could impart knowledge of domain-agnostic fillers (e.g., colours, shapes), reducing the need to explicitly provide $(x, x')$ and $I$ to the model.

3. **Segmentation Masks**: Segmentation masks for each object may potentially reduce the need to explicitly provide $(x, x')$ and $I$ to the model.

## D.3 Downstream Utility

Our investigation of downstream utility centers on two selected tasks – FoV regression/classification and abstract visual reasoning, which aligns with the standard framework for assessing the quality and downstream utility of compositional representations [26, 30, 33, 40, 44, 46]. While existing work [30, 33] demonstrates that explicitly compositional representations enhance downstream sample efficiency compared to non-compositional representations, a result we improve upon (C.5.1), the broader utility of compositional representations remains a topic of ongoing exploration [21, 40, 42, 44].

Theoretical perspectives [1, 2] argue that explicitly compositional representations are fundamental in the production of productive, systematic, and inferentially coherent thought – 3 key properties characterising human cognition. Investigating how explicitly compositional representations can yield empirical benefits across these dimensions represents an essential avenue for future research. Although preliminary studies [40, 42, 44] do not find strong evidence that explicitly compositional representations improve compositional generalisation (a key aspect of systematicity), [44] suggests that this finding is because compositional representations are necessary, but not sufficient to induce systematicity; an explicitly compositional processing approach [6] is also required.

Future work could extend our theoretical framework with the hope of producing empirical results consistent with the theoretical arguments of [1, 2]. In particular, as our unbinding module is designed to provably and efficiently recover structured role-filler constituents from the Soft TPR, it may be possible to exploit this module to systematically reconfigure roles and fillers from existing representations to create representations of novel combinations of role-filler bindings (i.e., novel

compositional data). This type of ability could prove extremely beneficial in areas such as concept learning and compositional generalisation.

## D.4 Dimensionality

In this subsection, we denote the dimensions of the role and filler embedding spaces as $D_F$ and $D_R$ respectively. We also denote the number of fillers as $N_F$ and the number of roles as $N_R$.

The Soft TPR belongs to $V_F \otimes V_R$, which is a $D_F \times D_R$ dimensional space, which grows multiplicatively in $D_F, D_R$. Several factors, however, mitigate scalability concerns in light of this fact:

1. **Independence of Embedding Space Dimensionality**: We note that the dimensionality of the role and filler embedding spaces ($D_R, D_F$ respectively) are properties of the corresponding embedding functions ($\xi_R : R \to V_R$ and $\xi_F : F \to V_F$) and thus, can be fixed independently of the number of roles, $N_R$, the number of fillers, $N_F$, or the number of total role-filler bindings (which we denote by $n$) within a domain. Thus, it is possible to fix the Soft TPR's dimensionality ($D_F \times D_R$) to be smaller than $N_F \times N_R$ (the number of roles/FoV types multipled by the number of fillers/FoV tokens) or $n$ (the number of bindings), which all may be large in complex visual domains. As illustrated in Table 46, the dimensionality of the TPR is smaller than $N_F \times N_R$ in both the Shapes3D and MPI3D domains.

2. **Relaxing Orthogonality**: While $D_F$ can be set a priori with no regard to $N_R, N_F$ or $n$, we require $D_R \geq N_R$ for semi-orthogonality of the role-embedding matrix $M_{\xi_R}$, which guarantees faithful (see proof 2 of A.2) and computationally efficient (see 'Unbinding' heading of B.4.1) recoverability of constituents. It is, however, possible to relax this constraint (i.e., to have $D_R < N_R$) to further reduce dimensionality. In this case, semi-orthogonality of $M_{\xi_R}$ is impossible and hence the recoverability of constituents cannot be guaranteed, however, there are some less stringent guarantees on the outcome of unbinding that can still be derived (see p291 of [3] for more details).

We also more explicitly compare the dimensionality of the Soft TPR with baselines in Table 47. Scalar-tokened symbolic representations have a low dimensionality of 10 ($N_R$) at the expense of representational expressivity (each representational constituent $\psi_i(a_i)$ is scalar-valued). In contrast, Soft TPR has vector-valued representational constituents (i.e. $\approx \xi_F(f_{m(i)}) \otimes \xi_R(r_i)$), similar to VCT and COMET. When compared to these models, the Soft TPR has significantly lower dimensionality compared to VCT and is comparable with COMET.

Table 46: Comparison of multiplicative dimensionality

| Parameter | Dataset | | |
|---|---|---|---|
| | Cars3D | Shapes3D | MPI3D |
| $D_R$ | 12 | 16 | 12 |
| $N_R$ | 10 | 10 | 10 |
| $D_F$ | 128 | 32 | 32 |
| $N_F$ | 106 | 57 | 50 |
| $D_F \cdot D_R$ (*TPR dimension*) | 1536 | **512** | **384** |
| $N_F \cdot N_R$ | **1060** | 570 | 500 |

Table 47: Comparison of dimensionality of representations

| Model | Cars3D | Shapes3D | MPI3D |
|---|---|---|---|
| | Representational dimension | | |
| Symbolic scalar-tokened compositional representations | | | |
| SlowVAE | 10 | 10 | 10 |
| Ada-GVAE-k | 10 | 10 | 10 |
| GVAE | 10 | 10 | 10 |
| ML-VAE | 10 | 10 | 10 |
| Shu | 10 | 10 | 10 |
| Symbolic vector-tokened compositional representations | | | |
| VCT | 5120 | 5120 | 5120 |
| COMET | **640** | *640* | *640* |
| Fully continuous compositional representations | | | |
| Ours (Soft TPR) | *1536* | **512** | **384** |

## D.5 Computational Cost

In the Soft TPR Autoencoder, the expensive tensor product operation is employed to generate $\psi_{tpr}^*$. Given the computational cost of the tensor product, we more concretely examine the computational

cost of training the Soft TPR Autoencoder by computing the FLOPs for a single forward pass on a batch size of 16 using the open-source implementation of fvcore https://github.com/facebookresearch/fvcore/tree/main/docs, visible in Table 48. This data demonstrates that, despite the tensor product's computational cost, the mathematically-informed derivation of our model allows it to obtain compositional representations with vector-valued representational constituents at a significantly lower cost compared to relevant, vector-tokened baselines (2 orders of magnitude less than VCT, and 4 orders of magnitude less than COMET).

Future work could explore the use of tensor contraction techniques to reduce computational expense. For instance [29] uses a Hadamard product based tensor product compression technique. This reduces computational cost from $n^2$ (tensor product of 2 vectors) to $n$ (Hadamard product), but comprises the theoretical guarantees on constituent recoverability. We believe developing tensor contraction techniques within the TPR framework is an important direction for future research, to ensure efficient TPR-based representations with provable recoverability of constituents.

Table 48: Comparison of FLOPs required for a forward pass of batch size 16

| Model | Cars3D | Shapes3D | MPI3D |
|---|---|---|---|
| | Representational dimension | | |
| Symbolic scalar-tokened compositional representations | | | |
| SlowVAE | $1.47 \times 10^8$ | $1.47 \times 10^8$ | $1.47 \times 10^8$ |
| Ada-GVAE-k | $1.47 \times 10^8$ | $1.47 \times 10^8$ | $1.47 \times 10^8$ |
| GVAE | $1.47 \times 10^8$ | $1.47 \times 10^8$ | $1.47 \times 10^8$ |
| ML-VAE | $1.47 \times 10^8$ | $1.47 \times 10^8$ | $1.47 \times 10^8$ |
| Shu | $1.45 \times 10^8$ | $1.45 \times 10^8$ | $1.45 \times 10^8$ |
| Symbolic vector-tokened compositional representations | | | |
| VCT | $2.53 \times 10^{11}$ | $2.53 \times 10^{11}$ | $2.53 \times 10^{11}$ |
| COMET | $5.12 \times 10^{13}$ | $5.12 \times 10^{13}$ | $5.12 \times 10^{13}$ |
| Fully continuous compositional representations | | | |
| Ours (Soft TPR) | $3.21 \times 10^9$ | $2.93 \times 10^9$ | $2.89 \times 10^9$ |

