# OpenReview forum: "Fully Distributed, Flexible Compositional Visual Representations via Soft Tensor Products"
_NeurIPS.cc/2024/Conference — NeurIPS 2024 poster_

### Official Review · Reviewer_xKnt · 2024-06-26

**Soundness:** 4
**Presentation:** 4
**Contribution:** 3
**Rating:** 7
**Confidence:** 3

**Summary:**

The authors propose the use of the tensor product to model the interactions between object properties and their values, in contrast to the usual concatenation-based fusion for compositional representations. Extensive experiments on a large variety of image datasets are performed, where performance gains are often significantly higher than those of the baselines considered.

**Strengths:**

- Modeling “multiplicative interactions” has proven to be a powerful mechanism for deep learning [1], but is largely neglected in modern times. This paper proposes an interesting way of fusing compositional representations of object properties and their values through the tensor product (and subsequent summation). Beyond the TPR’s success in this paper, I imagine the insights here through the proposed model form could help the design of more expressive deep learning architectures in future work more generally.
- The experiments are exhaustive, the technical exposition clear and crisp, and for the most part, every architectural design and decision is justified in great detail. Overall, the paper is remarkably digestible given the technical sophistication, yet offers many insights.

---

- [1]: Jayakumar, Siddhant M. et al. “Multiplicative Interactions and Where to Find Them.” *International Conference on Learning Representations* (2020).

**Weaknesses:**

# [W1] Multiplicatively large dimensionality of the TPR representations & issues scaling to larger settings

My main criticism of the paper is the resultant TPR's dimensionality. In particular, the soft TPR representations live in a $D_f\cdot D_r$ dimensional space. Due to the use of the Kronecker product, the dimension of the TPR representations grows multiplicatively with the two terms.

Whilst the datasets studied in the paper are relatively simple and only 10 factors of variation are modeled, it seems prudent to acknowledge that the TPR size could grow prohibitively large for increased values of $D_f, D_r$ for more complex datasets outside of controlled settings, where significantly larger number of FoVs exist. In particular, the NeurIPS checklist states: `The authors should discuss the computational efficiency of the proposed algorithms`.

For example, I see from [L1008] that $N_R:=10$, and $N_F$ is as high as $106$ for Cars3D. Even in this regime with a very small number of roles, the TPR representations are (presumably?) larger than the concatenation-based representations of the baselines. A comparison of FLOPs/multiply-adds needed for the compared methods would be appreciated to better understand the methods’ drawbacks through the use of the (often computationally expensive) tensor product.

**Questions:**

**[Q1]** As expanded upon in [L933], the “role” embedding matrix $M_{\xi_R}$ is initialized with $n$ orthogonal columns but not trained. We are told that it is “intuitive” that role embeddings do not need to convey semantic information, but I think it would be interesting to see an ablation study on this.

In particular: if it is the case that the embedding matrix does not need to learn semantic representations, why not make the simpler choice of initializing $M_{\xi_R}$ as a (truncated) identity matrix (this shares the same property of semi-orthogonality)? The decision not to make this parameter learnable is peculiar, and I would be interested in seeing its impact explored experimentally (for example, through the DCI score as in Table. 7).

**Limitations:**

Limitations are discussed well throughout (and in further detail throughout the appendix), but an extra discussion of the potential drawbacks of increased computational costs should be made.

---

> ### Author Rebuttal · Authors · 2024-08-07
>
> We thank the reviewer for their comprehensive and insightful review.
>
> **W1: Scalability Issues - Multiplicative Dimensionality of (Soft) TPR**
>
> As noted, the Soft TPR lives in a $D_{F} \cdot D_{R}$ dimensional space, which grows multiplicatively. However, several factors mitigate scalability concerns:
>
> 1.	Independent Dimensionality: The dimensions of the role and filler embedding spaces ($D_{R}$ and $D_{F}$) can be fixed independently of the number of roles, $N_{R}$, and the number of fillers, $N_{F}$. Thus, Soft TPR’s dimensionality ($D_{F} \cdot D_{R}$) can be smaller than $N_{F} \cdot N_{R}$, the number of roles (FoV types) multiplied by the number of fillers (FoV tokens), which may be large in complex visual domains. In this rebuttal, Table* 5 is added to illustrate this, with the dimensionality of the TPR being smaller than $N_{F} \cdot N_{R}$ in both the Shapes3D and MPI3D datasets.
>
> 2.	Relaxing Orthogonality: While $D_{R} \geq N_{R}$ is needed for semi-orthogonality of the role embedding matrix, $M_{\psi_{R}}$, which guarantees faithful (L841-858) and computationally efficient (L917-929) recoverability of constituents, it is possible to relax this constraint to further reduce dimensionality.  When $D_{R} < N_{R}$, semi-orthogonality of $M_{\psi_{R}}$, and thus faithful recoverability of constituents cannot be guaranteed, but some less stringent guarantees on unbinding outcomes can still be provided (see 291 of [1] for details).
>
> Table* 6 is added in this rebuttal to compare Soft TPR’s dimensionality with other baselines. Scalar-tokened symbolic representations have a low dimensionality of 10 ($N_{R}$) at the expense of expressivity (each representational constituent $\psi_{i}(a_{i})$ is scalar-valued). In contrast, Soft TPR has vector-valued constituents (i.e. approx $\psi_{F}(f_{m(i)}) \otimes \psi_{R}(r_{i})$), like VCT and COMET. When compared to these models, our Soft TPR has significantly lower dimensionality than VCT and is comparable to COMET.
>
> Despite these mitigations, we acknowledge that the tensor product is computationally expensive. To more concretely address this concern, Table* 7 is added to indicate the FLOPs for a single forward pass on a batch size 16 using  fvcore [F]. This data demonstrates that, despite the tensor product's computational cost, the mathematically-informed derivation of our model allows it to obtain compositional representations with vector-valued constituents at a significantly lower cost than relevant baselines (2 orders magnitude less than VCT and 4 orders magnitude less than COMET).
>
> Future work could explore the use of tensor contraction techniques [G,H] to reduce computational expense. For instance, [I] uses a Hadamard product based tensor product compression technique. This reduces computational cost from $nm$ (tensor product of 2 vectors) to $n$ (Hadamard product), but compromises the theoretical guarantees of constituent recoverability. We believe developing tensor contraction techniques within the TPR framework is an important direction for future research, to ensure efficient TPR-based representations with provable recoverability of constituents.
>
>
> **Q1: Non-learnability of role embedding matrix**
>
> Semi-orthogonality of the role embedding matrix, $M_{\psi_{R}}$ ensures that the unbinding vector, $u_{i}$ (defined as the $i$-th column of $M_{\psi_{R}}$’s left inverse (L841-850) corresponds to the $i$-th column of $M_{\psi_{R}}$ (L925-927). The Unbinding Module leverages this to efficiently unbind soft fillers from $z$, using columns of $M_{\psi_{R}}$ as unbinding vectors instead of performing costly matrix inversion. If $M_{\psi_{R}}$ is not semi-orthogonal, the Unbinding Module cannot reliably produce soft fillers from $z$, as its columns are not true unbinding vectors.
>
> Extracting the soft fillers from the Soft TPR via unbinding ensures that the TPR decoder produces $\psi_{tpr}^{\*}$ as per Eq 5. Minimising $||z-\psi_{tpr}^{\*}||_{2}^{2}$ (term 1 in Eq 6) encourages the encoder, $E$, to produce outputs that are continuous relaxations of explicit TPRs.
>
> In response to the reviewer's suggestion, we add Table* 8 in this rebuttal with results of the ablation where $M_{\psi_{R}}$ is learnable. This retains a competitive edge over disentanglement baselines, but substantially lower FactorVAE and DCI scores empirically demonstrate the importance of fixing $M_{\psi_{R}}$ for effective Soft TPR Autoencoder performance. This fixation leverages the unique mathematical properties of the Soft TPR/TPR framework, which are essential for effectively learning the representation.
>
> We thank the reviewer for the suggestion to use a truncated identity matrix. Results in Table*8 show that a truncated identity matrix retains an edge over learnable $M_{\psi_{R}}$, again highlighting the need for semi-orthogonality. However, the lower disentanglement results suggest potential benefits of random initialisation. Unbinding with columns of a truncated identity will force soft fillers to be axis-aligned ($\tilde{f}\_{m(i)}:= zu_{i}$), potentially limiting the flexibility and richness of learned representations. Additionally, a more varied set of weights in $M_{\psi_{R}}$ produced by random initialisation helps to break symmetry in subsequent layers, and injects noise, which can improve overall learning dynamics [J, K].
>
> [F]: Fvcore https://github.com/facebookresearch/fvcore/tree/main/docs
>
> [G]: Kossaifi et al. "Tensor Contraction Layers for Parsimonious Deep Nets." CVPR 2017.
>
> [H]: Sharir et al. "Neural Tensor Contractions and the Expressive Power of Deep Neural Quantum States." Physical Review 2022.
>
> [I]: Schlag et al. "Enhancing the Transformer with Explicit Relational Encoding for Math Problem Solving." 2019.
>
> [J]: Glorot and Bengio. “Understanding the difficulty of training deep forward neural networks”. AISTATS 2010.
>
> [K] Noh et al. “Regularising Deep Neural Networks by Noise: Its Interpretation and Optimization.” Neurips 2017.

---

> > ### Comment · Reviewer_xKnt · 2024-08-08
> >
> > Thanks to the authors for their thorough response.
> >
> > The rebuttal was a pleasure to read; the extra experiments and ablations are insightful, and the reviewers make a good point that—despite the dimensionality of the space growing multiplicatively—the independent dimensionality of the two factors means this is not always too problematic.
> >
> > The “weaknesses” raised in my initial review were only very minor in the first place—the authors have acknowledged the multiplicative growth of the space, and I see no critical reasons why the paper should not be accepted.

---

> ### Author Response · Authors · 2024-08-09
>
> Dear Reviewer xKnt,
>
> We are glad you found our additional experiments and ablations insightful. Your feedback on our approach, especially regarding the dimensionality, is greatly valued. Thank you again for your time and thoughtful review.
>
> Best,
>
> Submission14814 Authors.

---

### Official Review · Reviewer_93av · 2024-07-07

**Soundness:** 3
**Presentation:** 3
**Contribution:** 3
**Rating:** 7
**Confidence:** 4

**Summary:**

This work explores compositional representations -- considered a crucial capability underlying intelligent human behaviour in deep learning systems. It argues that there is an incompatibility between discrete symbolic compositional representations—e.g. as obtained through traditional disentanglement approaches—and the continuous vector spaces underlying deep learning systems. To address this, the authors introduce a novel continuous compositional representation that builds on the Tensor Product Representation (TPR) approach, akin to a soft approximation to the TPR. They introduce a method for learning soft TPRs with weakly supervised data called the Soft TPR Autoencoder, and apply it to visual representation learning. This model demonstrates state of the art disentanglement for representation learning and improved sample efficiency for downstream models using those representations.

**Strengths:**

- importance: compositional representation learning, and compositional generalisation and sample efficiency is an important and relatively under-explored area in deep learning research. By focusing on an approach for representation that aims to be more compatible with deep learning this work makes important contributions to this area.
- novelty: the method presented is interesting & new but builds on the well established TPR framework.
- clarity: the method is explained well and in enough detail to understand and reproduce it.
- evaluation: the paper provides a thorough evaluation of the model, including comparisons with the relevant baselines, convergence rates, and performance in low sample regimes. The better performance on downstream tasks w.r.t. number of training examples is particularly interesting and supports some of the motivations for compositional representation.
 - interested to see where this work might develop in future work exploring hierarchical compositional representations.

**Weaknesses:**

- motivation for the approach & model could be unpacked a bit more (see questions)
- other domains: the authors focus on applying the work to the visual data, outside the typical domain of TPR models. In general, a strong feature, enabling them to tackle the messier world of complex visual data (and weaker super vision) and compare the model to the many disentanglement approaches that have previously been applied in the visual domain. However, it could be interesting to see how the _Soft_ TPR approach compares to traditional TPR in its typical domain of language.
 - no high impact is shown for downstream utility. the improvements seen for downstream tasks are certainly interesting, particularly in the low data regime, however only two tasks are explored. are there any more good downstream tasks to evaluate the utility of their learned representations? could there even be more speculation as to where future work might really leverage what can be learned with this approach?
 - need for supervision (albeit weakly): could the work be extended to use different forms of supervision or less/no supervision. noting that their ablation showed the importance of the supervision, perhaps the authors could explore more variants on this ablation, and speculate on extensions of the model that could reduce the need for supervision?

**Questions:**

A couple of questions, both on the motivations of the approach in the paper and how it could be improved. In both cases, these are more questions focused on the conceptual motivations and understanding than what the results show:
 - In terms of motivating the approach, can you be more specific about how representations obtained with traditional disentanglement approaches (typically continuously-valued themselves) could create an incompatibility with the continuous vector space of deep learning systems, and lead to suboptimal performance for the representation learning and downstream models? Any hypothetical or concrete examples (and/or references discussing this point)?
- Relative to TPR, what is the relaxation in Soft TPR supposed to be helping with, and why? The manifold example is given, e.g. in Figure 1c, but seems quite vague. Also discussed is that the data being explained maybe only approximately compositional, but why would TPR fail with this and why would we specifically expect soft TPR to help? Could this be unpacked more, perhaps with some concrete examples.

---

> ### Author Rebuttal · Authors · 2024-08-07
>
> We appreciate the reviewer’s detailed and thoughtful feedback.
>
> **W1, Q1, Q2: Motivation for approach and model**
>
> Thank you for the detailed questions on the motivation. We clarify these points in the General Response.
>
> **W2: Limited Domains**
>
> Applying Soft TPR to language is an intriguing future direction, especially as language can deviate from strict algebraic compositionality. E.g., idiomatic expressions like ‘spill the beans’ cannot be understood as a function of their constituents alone. Traditional TPR, relying on strict algebraic compositionality (Eq 1) may be less effective in such cases. Soft TPR's more flexible, approximately compositional structure might better handle such complexities.
>
> To adapt our framework to language, we replace our Conv/Deconv encoder/decoders with simple RNNs, retain our TPR decoder and remove the semi-supervised loss, using Eq 6 as the full loss. Preliminary results (Table* 4 in this rebuttal) on the BaBI dataset [A] are compared with TPR baselines from AID [36]. Our Soft TPR Autoencoder does not yet surpass AID, but notable points include:
>
>   1) Our use of simpler RRN-based encoders and an MLP-based downstream network, unlike the more sophisticated architectures of [36]
>
>   2) Soft TPR retains its performance improvement above the corresponding explicit TPR it can be quantised into
>
>   3) The smaller gap between systematic vs non-systematic dataset splits in our model compared to TPR-RNN (+AID) and FWM
>
>   4) We train our representation learner using self-supervision (reconstruction loss) alone, only employing supervision on the downstream prediction network, while the baselines employ strong supervision and end-to-end training to produce representations
>
>
> **W3: Downstream Utility**
>
> Our focus on the 2 selected tasks (FoV regression/classification and abstract visual reasoning) aligns with the standard framework for assessing the quality and downstream utility of compositional representations [19,23,26,32,B,C]. While explicitly compositional representations enhance downstream sample efficiency over non-compositional representations [23,26], which we improve upon (C.5.1), their broader utility remains an open question [15,32,33,C].
>
> Theoretical arguments [D,E] posit that explicitly compositional representations are crucial for productive, systematic, and inferentially coherent thought/processing – 3 key properties of human cognition. Using compositional representations to produce empirical benefits along these dimensions is a crucial direction of future research. While preliminary results [32,33,C] do not find strong evidence that compositional representations enhance compositional generalisation (a form of systematicity), [C] suggests that compositional representations are necessary, but not sufficient to promote systematicity; an explicitly compositional form of processing is also required.
>
> We believe our theoretical formalism offers a unified framework for future research to achieve empirical results aligned with [D,E]. In particular, as our unbinding module provably recovers structured role-filler constituents from the Soft TPR (L189-197, L841-858), our model enables the systematic rearrangement of a representation’s roles/fillers to form novel composite representations. This potentially advances the development of compositional processing methods as suggested by [C] and holds potential in the domains such as concept learning and compositional generalisation.
>
> We will add some discussion in the paper regarding this point.
>
> **W4: Need for Weak Supervision**
>
> To produce a compositional representation $\psi(x) = C(\psi_{1}(a_{1}), \ldots, \psi_{n}(a_{n}))$ (L116), each representational constituent $\psi_{i}(a_{i})$ must map directly to a data constituent, $a_{i}$. Without supervision, this is challenging because the data constituents are unknown.
>
> In generative models, [19] showed that unsupervised disentanglement is theoretically impossible for essentially this reason: without supervision, there is no mechanism to match representational constituents with ground truth generative factors (role/filler bindings). Although our non-generative model avoids this impossibility result, the proof’s intuition is compelling. Thus, we use weak supervision [7,16,24,26,28] by presenting the model with data pairs $(x, x')$ differing in a subset of FoVs, indexed by set $I$. This weak supervision is minimal, only providing knowledge of the changing FoV types (e.g. ‘floor colour’) and not the FoV tokens (i.e., the specific floor colours in $x, x’$).
>
> As suggested by the reviewer, Table* 3 is added in the rebuttal to present additional variants on the supervision ablation from the perspective of disentanglement performance (please refer to Eq 7 for definitions of $\lambda_{1}, \lambda_{2}$).
>
> We outline some possible extensions to reduce weak supervision:
>
>  1. In the visual domain, some roles e.g. ‘object position' correspond to affordances.  Embodied agents may be able to reduce the need for explicit supervision by collecting $(x, x'), I$ through interaction.
>
>  2. Initialising the filler embedding matrix, $M_{\psi_{F}}$ with embeddings learnt by a pre-trained vision model could impart knowledge of domain-agnostic fillers (e.g. colours, shapes), reducing the need for weak supervision.
>
>  3.  Segmentation masks for each constituent may potentially reduce the need for weak supervision.
>
>
> [A] Weston et al. 2015. Towards ai-complete question answering: A set of prerequisite toy tasks.
>
> [B] Yang et al. 2022. Towards Building a Group-based Unsupervised Representation Disentanglement Framework. ICLR.
>
> [C] Montero et al. 2022. Lost in Latent Space: Examining Failures of Disentangled Models at Combinatorial Generalisation. NeurIPS.
>
> [D] Fodor. 1975. The Language of Thought: A Theory of Mental Representation.
>
> [E] Chomsky. 1957. Syntactic Structures.

---

> > ### Comment · Reviewer_93av · 2024-08-12
> >
> > Thanks for the detailed and clear follow-up, many of concerns have been addressed and I have raised my score.

---

> ### Author Response · Authors · 2024-08-13
>
> Dear Reviewer 93av,
>
> We are glad our response addressed many of your concerns. Thank you again for your time and insightful review. We really appreciate your thoughtful feedback,
>
> Submission14814 Authors.

---

### Official Review · Reviewer_6VtP · 2024-07-12

**Soundness:** 3
**Presentation:** 3
**Contribution:** 2
**Rating:** 5
**Confidence:** 4

**Summary:**

This paper introduces a novel framework for representation learning known as Soft Tensor Product Representations (Soft TPR), aimed at capturing the compositional structure of data more effectively. The authors propose a continuously-valued compositional representation that contrasts with traditional symbolic methods. The paper makes several key contributions in the realm of representation learning, including the conceptualization of Soft TPR, the Soft TPR Autoencoder, and the demonstration of the benefits of this representation for both representation learning and downstream models.

**Strengths:**

The idea of Soft TPR brings a fresh perspective to the field of representation learning, offering a new way to represent compositional structures in a continuous manner. It provides a thorough theoretical exploration of Soft TPR, complete with detailed mathematical proofs and framework extensions. The paper clearly articulates the differences between Soft TPR and existing methods, holding significant potential for enhancing model interpretability and improving robustness to covariate shift.

**Weaknesses:**

1. No MPI scores for disentanglement (in Table 1) are reported.
2. The construction of the model's loss function involves a multitude of hyperparameters, suggesting that the model might require a complex and intricate tuning process. Compared to other models like VCT, the process of adjusting parameters to achieve optimal results could involve additional complexity and such a requirement might pose challenges in terms of computational resources and time.
3. The reason why *Soft TPR is better than TPR is not fully explored.

**Questions:**

1. what is the dissentanglment performance of TPR?  The paper only report the downstream results of TPR in Table 6, is it possible to report the disentangment performance of TPR in Table 1?
2. how many pairs are provided for the weakly supervised? For example, in Table 4, the numbers are the total samples for training, or the labeled sample pairs?
3. VCT uses attention-based operations (cross-attention and self-attention) to extract concept vectors, and reconstruct images from the concept tokens, what is the difference between those attention-based operation and the unbinding module  or TPR constructor in this paper?

**Limitations:**

I don't see any negative societal impact.

---

> ### Author Rebuttal · Authors · 2024-08-07
>
> We thank the reviewer for their detailed and thoughtful review.
>
> **W1**: To our best knowledge, MPI score is not established in disentanglement learning literature [19,23-26,31,32,34,38].
>
> Consistent with VCT [34], we evaluate disentanglement using 3 datasets (Cars3D, Shapes3D, MPI3D) and 4 metrics (FactorVAE, DCI, MIG, BetaVAE) commonly used in disentanglement studies [19,23-26,29,31-34]. Table 1 in our paper compares our method with SoTA approaches using FactorVAE and DCI scores. For MIG and BetaVAE results, please see Table 17 of C.3.3.
>
> To avoid confusion, we will clarify in Table 1’s caption that additional results are in the supplementary.
>
> **W2**: Our loss function in Eq 7 has 2 tuneable hyperparameters, $\lambda_{1}$ and $\lambda_{2}$ in the weakly supervised loss (we set $\beta$ in $\mathcal{L}_{u}$ of Eq 6 to 0.5 (L1006)).
>
> The use of hyperparameters for loss functions is common in compositional representation learning. Canonical VAE disentanglement methods [4,8,9,11,18] and our baselines [31,26,29] each have 1-2 loss hyperparameters.
>
> VCT’s lack of tuneable loss hyperparameters is a strength but VCT is far more computationally expensive to train compared to our model.
>
> To illustrate this point in this rebuttal, Table* 7 is added for comparing FLOPs. Table* 1 is added to show the total time for tuning our model on an RTX 4090 including 2 loss hyperparameters and 4 architectural hyperparameters (details in B.3). Table* 2 is added for the cost of tuning our model relative to the total time required for training a single VCT model on a V100 GPU for 200,000 iterations, excluding the 48-144 hours for training VCT’s VQ-VAE image tokeniser.
>
> Furthermore, ablation experiments (C.6.2) demonstrate our model’s robustness to both architectural and loss hyperparameters indicating multiple viable hyperparameter points that ease the tuning process.
>
>
> **W3**: Please refer to our detailed explanation of the differences between Soft TPR and TPR in Section 2 of our General Response.
>
> **Q1**: The disentanglement performance of the Soft TPR and TPR are identical. To establish why, we outline our method of extending existing disentanglement evaluation procedures to the continuous instantiation of compositional structure of the Soft TPR/TPR (please refer to C.3.2 for additional details).
>
> Existing disentanglement metrics assume a symbolic representation of compositional structure, requiring a vector $v$ with a dimension $N_{R}$ matching the number of roles (FoV types). Each dimension of $v$ is populated with a scalar-valued filler (FoV token). To produce $v$ from the Soft TPR we follow the procedure in C.3.2:
>
> 1) Quantised fillers $\{\psi_{F}(f_{m(i)})\}$ are extracted from $z$ via Unbinding and Quantisation modules.
> 2) Each dimension of $v$ is populated with the index of the quantised filler for role $i$ (i.e. $v_{i} = m(i)$).
>
> As the explicit TPR, $\psi_{tpr}^{*}$ is built from the Soft TPR’s quantised fillers (Eq 5), the disentanglement scores for both are identical.
>
> We use the quantised fillers of the Soft TPR to construct $v$, not its soft fillers $\{\tilde{f}_{m(i)}\}$ for 2 reasons:
>
> 1) There is no natural way of quantising soft fillers into scalars other than using PCA, which incurs significant information loss.
>
> 2) Quantised fillers align with the intent of disentanglement metrics, which assess perfect (not approximate) compositionality.
>
> The Soft TPR, $z$, does not enhance disentanglement compared to the explicit TPR, $\psi_{tpr}^{*}$. However, it is important to note our motivation for continuously relaxing the TPR into the Soft TPR is to not to more explicitly represent compositional structure, but rather, to further align TPR-based structure with continuous vector spaces (details in Section 2 of General Response). This enhances downstream performance and representation learner convergence compared to the TPR (C.6.1). Our model is also able to always recover precise, algebraic compositional structure (with form $\sum_{i} \psi_{F}(f_{m(i)}) \otimes \psi_{R}(r_{i})$) from the Soft TPR, $z$, by quantising it using the TPR decoder. Our model can thus leverage the duality of the Soft TPR’s (approximately) compositional structure.
>
> **Q2**: The term refers to the number of individual data points, not pairs (i.e., 100 samples=50 pairs). We will add a footnote for clarification.
>
> **Q3**: The key differences are as follows:
>
> 1) Motivation: VCT applies attention to generate concept/image tokens, whereas our framework uses the 3 modules of the TPR decoder to generate a highly specific element $\psi_{tpr}^{\*}$ as per Eq 5. VCT uses these concept tokens to produce a symbolic form of compositional structure, whereas we use $\psi_{tpr}^{\*}$ to minimise $||z - \psi_{tpr}^{\*}||^{2}_{2}$ (term 1 in Eq 6), which encourages the Encoder, $E$, to produce a Soft TPR, a continuous representation of (approximately) compositional structure.
>
> 2) Continuous vs Symbolic Representation: VCT represents compositional structure symbolically, through direct sums, with each constituent $c_{i}$ embedded in a separate subspace. Our method embeds all constituents in the same vector space and additively superimposes them. This continuous form of compositional structure produces empirical benefits (C.3.3, C.4.2, C.5).
>
> 3) Constituent Structure: VCT’s unstructured constituents (concept tokens) do not guarantee systematic relations between constituents with the same filler/role. Soft TPR, however, with its structured (soft) role-filler bindings ensures systematic relations between constituents with the same filler/role, resulting in a disentanglement improvement.
>
> 4) Mathematical Basis: Our modules are mathematically designed to recover soft fillers, quantised fillers, and explicit TPRs (Eqs 3-5). Attention, as employed by VCT, cannot theoretically guarantee the production of these structures.

---

> > ### Comment · Reviewer_6VtP · 2024-08-12
> >
> > Thanks for the response. Most of my concerns are addressed. I have raised my score.

---

> ### Author Response · Authors · 2024-08-12
>
> Dear Reviewer 6VtP,
>
> We are glad that our response addressed most of your concerns. Thank you again for your time and thoughtful feedback regarding our approach.
>
> Best,
>
> Submission14814 Authors.

---

### Author Rebuttal · Authors · 2024-08-07

(Please note all rebuttal tables are included in the 1-page PDF)

**1) Incompatibility between disentangled representations and deep learning’s continuous vector spaces (Reviewer 93av Q1)**

Traditional disentanglement methods, although producing continuously-valued representations $\psi_{d}(x)$, use a symbolic, direct-sum approach to represent compositional structure (L37-40,122-124). Specifically, each FoV/data constituent $a_{i}$ is represented by a single dimension/contiguous subset of dimensions, $\psi_{i}(a_{i})$, which are concatenated together to form $\psi_{d}(x) = \psi_{1}(a_{1}) \oplus \ldots \oplus \psi_{n}(a_{n})$

Here, each representational component $\psi_{i}(a_{i})$ representing a FoV is embedded into a separate, independent subspace $V_{i}$ within the overall vector space. For example, consider 2 features, colour and shape, with embeddings $\psi_{col}(purple) = (1\\;0)^{T}, \psi_{sh}(square) = (1\\;0)^{T}$.

A disentangled representation is: $\psi_{d}(purple\\;square) = \psi_{col}(purple) \oplus \psi_{sh}(square) = \begin{bmatrix} 1 \\\ 0 \end{bmatrix} \oplus \begin{bmatrix} 1 \\\ 0 \end{bmatrix} = \begin{bmatrix} 1 \\\ 0 \\\ 1 \\\ 0 \end{bmatrix}$.

Here, colour and shape are allocated to distinct subspaces of the representational space $\mathbb{R}^{4}$ (i.e. dims 1&2 for colour,  dims 3&4 for shape). This discrete allocation mirrors symbolic systems where discrete symbols occupy separate spaces in the representation, although $\psi_{d}(x)$ is continuously-valued.

Deep learning systems rely on gradient-based optimisation. Disentanglement’s symbolic method of representing constituency structure (i.e., allocating constituents to discrete subspaces of $\mathbb{R}^{4}$) complicates this process, as discrete subspace boundaries (i.e. between dims 1&2, dims 3&4) must be managed for each constituent. This complicates gradient-based processes, as changes must navigate abrupt transitions between subspaces rather than smooth, continuous alterations.

Additionally, this symbolic method of representing constituency structure requires subspace alignment to ensure the overall representation in the larger vector space, $V_{F}$ is semantically meaningful (e.g. the colour subspace having significantly larger magnitudes than others). Guaranteeing this may be challenging.

In contrast, TPR-based representations combine features in a unified vector space (e.g. $\psi_{col}(purple) \otimes \psi_{sh}(square) = (1\\;0)^{T} \otimes (1\\;0)^{T} \cong (1\\;0\\;0\\;0)^{T}$), meaning that constituents (e.g., colour, shape) are integrated into the same vector space and cannot be separated into discrete parts of the representational space, $\mathbb{R}^{4}$. This continuous approach of representing constituency structure avoids the issues of discrete subspace boundaries and subspace alignment, facilitating smooth gradient updates and more effective learning to enhance representation learner (C.3.3, C.4.2) and downstream (C.5) performance.

**2) Why the relaxation of the TPR into the Soft TPR (Reviewer 6VtP W3, 93aV Q2)**

 For concreteness, consider the set of roles and fillers $R = \\{object\\;colour, object \\; shape\\}, F = \\{purple, green, square\\}$. We define role embedding $\psi_{R}: R \rightarrow \mathbb{R}^{2}$ and filler embedding $\psi_{F}: F \rightarrow \mathbb{R}^{3}$ functions as follows: $\psi_{R}(object\\;shape) = (1\\;0)^{T}, \psi_{R}(object\\;colour) = (0\\;1)^{T}$ and $\psi_{F}(purple) = (1\\;2\\;3)^{T}$,  $\psi_{F}(green) =  (2\\;2\\;3)^{T}$, $\psi_{F}(square) = (0\\;0\\;1)^{T}$

2.1. Discrete Mapping: The possible TPRs that can be produced are:

1.	$\psi_{tpr}(purple/object\\;colour, square/object\\;shape) = \psi_{F}(purple) \otimes \psi_{R}(object\\;colour) + \psi_{F}(square) \otimes \psi_{R}(object\\;shape)  \cong (0\\;0\\;1\\;1\\;2\\;3)^{T} $
2.	$\psi_{tpr}(green/object\\;colour, square/object\\;shape) = \psi_{F}(green) \otimes \psi_{R}(object\\;colour) + \psi_{F}(square) \otimes \psi_{R}(object\\;shape)  \cong (0\\;0\\;1\\;2\\;2\\;3)^{T}$

The possible TPRs form a 2 element subset, $T$, of the underlying vector space $ \mathbb{R}^{6}$. Relaxing TPRs to Soft TPRs (L191) gives a larger set of points, $T_{s} = \\{(0\\; 0\\;1\\;2\\;2\\;3)^{T} + \alpha, (0\\;0\\;1\\;1\\;2\\;3)^{T} + \alpha : |\alpha| < \epsilon\\}$, which includes points like $(-0.0001\\;0.0002\\;1\\;1\\;2.004\\;3.009)^{T}$ and others. As $T_{s}$ has strictly more points than $T$, there should be more functions e.g., parameterising the map from the observed data to $T_{s}$, the set of Soft TPRs, than $T$, the set of TPRs. This should make the Soft TPR representation potentially easier to learn/extract information from than the TPR, reflected in our empirical results in C.6.1.

2.2. Quasi-Compositional Structure: The TPR enforces a strict algebraic definition of compositionality (i.e. $\sum_{i}\psi_{F}(f_{m(i)}) \otimes \psi_{R}(r_{i})$). The relaxation of this constraint in Soft TPR (L191) enables it to represent structures that only approximately satisfy the TPR’s strict algebraic definition of compositionality (e.g., in French liaison consonants, where a weighted sum of multiple fillers, rather than a single filler, bind to a role [L])

2.3. Serial Construction: Building explicit TPRs requires tokening constituents (role-filler binding embeddings) before the compositional representation can be produced [13,17,22,27,30,36,37]. Soft TPRs allow the encoder $E$ produce any arbitrary element of $V_{F} \otimes V_{R}$ (in this case $\cong \mathbb{R}^{6}$) provided it is sufficiently close to a TPR. Thus, once the Soft TPR Autoencoder is trained, it is theoretically possible to remove the TPR Decoder, and exploit vector space continuity to create approximately compositional representations directly from the Encoder, $E$, without needing to token the constituents.

[L]: Smolensky and Goldrick. Gradient Symbolic Representations in Grammar: The case of French Liaison. 2016.

---

> ### Public Comment · ~Bethia_Sun1 · 2025-01-15
>
> Please note there is a small error in the author rebuttal above, under section *1) Incompatibility between disentangled representations and deep learning’s continuous vector spaces*, where the tensor product is mistakenly taken over $\psi_{col}(purple) \otimes  \psi_{sh}(square)$ to produce the representation, instead of computing $\xi_{F}(purple) \otimes \xi_{R}(col) + \xi_{F}(square) \otimes \xi_{R}(sh)$, but the conclusion remains exactly the same. (Note that $\psi_{col}(purple)= \xi_{F}(purple) \otimes \xi_{R}(col)$ and $\psi_{sh}(square)= \xi_{F}(square) \otimes \xi_{F}(sh)$ should *both* be 4-dimensional, *not* 2-dimensional).

---

### Decision · Program_Chairs · 2024-09-25

**Decision:**

Accept (poster)

**Comment:**

This paper proposes Soft Tensor Product Representations (Soft TPR) and the Soft TPR Autoencoder, a novel continuous compositional representation framework designed to bridge the gap between discrete symbolic representations and continuous vector spaces.

It tackles a crucial problem addressing a major limitation of current deep learning. Reviewers found the paper strong due to its thorough theoretical analysis, novelty of approach, clear explanation, and comprehensive experiments. Initial concerns, such as some experimental setting details, were also well addressed during the rebuttal. Therefore, I recommend acceptance.